# Mapping protein carboxymethylation sites provides insights into their role in proteostasis and cell proliferation

Simone Di Sanzo[1,7], Katrin Spengler [2,7], Anja Leheis [2], Joanna M. Kirkpatrick [1,6], Theresa L. Rändler [2], Tim Baldensperger[3], Therese Dau [1], Christian Henning [3], Luca Parca [4], Christian Marx[1], Zhao-Qi Wang[1,5], Marcus A. Glomb[3], Alessandro Ori [1,8 ✉] & Regine Heller [2,8 ✉]

Posttranslational mechanisms play a key role in modifying the abundance and function of cellular proteins. Among these, modification by advanced glycation end products has been shown to accumulate during aging and age-associated diseases but specific protein targets and functional consequences remain largely unexplored. Here, we devise a proteomic strategy to identify sites of carboxymethyllysine modification, one of the most abundant advanced glycation end products. We identify over 1000 sites of protein carboxymethylation in mouse and primary human cells treated with the glycating agent glyoxal. By using quantitative proteomics, we find that protein glycation triggers a proteotoxic response and indirectly affects the protein degradation machinery. In primary endothelial cells, we show that glyoxal induces cell cycle perturbation and that carboxymethyllysine modification reduces acetylation of tubulins and impairs microtubule dynamics. Our data demonstrate the relevance of carboxymethyllysine modification for cellular function and pinpoint specific protein networks that might become compromised during aging.

[1] Leibniz Institute on Aging – Fritz Lipmann Institute (FLI), 07745 Jena, Germany. [2] Institute of Molecular Cell Biology, Center for Molecular Biomedicine, Jena University Hospital, 07743 Jena, Germany. [3] Institute of Chemistry, Food Chemistry, Martin-Luther-University Halle-Wittenberg, 06120 Halle/ Saale, Germany. [4] Bioinformatics Unit, IRCCS Casa Sollievo della Sofferenza, S. Giovanni Rotondo, Italy. [5] Faculty of Biological Sciences, Friedrich-Schiller-University of Jena, Jena, Germany. [6] Present address: Proteomics Science Technology Platform, The Francis Crick Institute, MW1 1AT London, UK. [7] These authors contributed equally: Simone Di Sanzo, Katrin Spengler. [8] These authors jointly supervised this work: Alessandro Ori, Regine Heller. ✉email: alessandro.ori@leibniz-fli.de; regine.heller@med.uni-jena.de

Advanced glycation end products (AGEs) are generated via a non-enzymatic glycation of proteins initiated by the reaction of glucose, fructose or highly reactive dicarbonyls such as methylglyoxal or glyoxal (GO) with amino acids[1]. Glycation is enhanced when glucose levels are high, when dicarbonyls deriving from glycolytic intermediates or lipid peroxidation accumulate[2], or when dicarbonyl detoxification by glyoxalases such as glyoxalase 1 (GLO1) and protein deglycase DJ-1 is low[3]. AGE formation may alter the structure and function of the targeted proteins and, in addition, AGEs may elicit their effects as ligands of the pro-inflammatory receptor for advanced glycation end products (RAGE)[4]. One of the most abundant AGEs in vivo is N(6)-carboxymethyllysine (CML)[5–8], whose formation is triggered, among other routes, by GO[7] (Supplementary Fig. 1a). CML is known to be chemically stable, to accumulate in human tissues in diabetes, atherosclerosis, neurodegeneration and aging, and thus to be a biomarker of aging[8].

A causal relationship between the buildup of AGEs and aging or individual disease is still under debate, but first evidence was provided by studies, in which AGE levels were modulated via GLO1. Overexpression of GLO1 increases lifespan in *C. elegans*[9] and reduces endothelial dysfunction in diabetic mice[10], while knockdown of GLO1 mimics diabetic nephropathy in non-diabetic mice[11]. Whereas glycation of extracellular proteins such as hemoglobin HbA1c[12] or collagen[13] is well described, reports on glycation of intracellular proteins are still scarce. Examples of those are histones[6,14–16], mitochondrial proteins[17,18], the 20S proteasome[19], enzymes involved in energy production[20], small heat shock proteins[21] and the sodium channel Nav1.8[22]. Glycation is increasingly seen as driver of metabolic disease and aging, and may elicit specific effects by targeting signaling proteins[23,24]. In this context, glycation of the ryanodine receptor associated with mitochondrial damage[25], reversible inhibitory glycation of nuclear factor erythroid 2-related factor 2 (Nrf2)[26], and methylglyoxal-induced dimerization of Kelch-like ECH-associated protein 1 (KEAP1) with subsequent activation of the KEAP1/Nrf2 transcriptional program[27] have been reported. The latter may play a role in the upregulation of defense systems, such as GLO1 and the ubiquitin-proteasome system (UPS), and may be involved in the hormetic effect of methylglyoxal[28]. In contrast to earlier assumptions of irreversible AGE formation, recent studies suggested the possibility of deglycation via DJ-1 or protein arginine deiminase 4[15,16].

A global characterization of the targeted proteins and pathways, which may significantly contribute to manifestation of metabolic diseases and aging, is prerequisite to further understanding pathophysiological effects of AGEs, like CML. Here, we developed a proteomic workflow based on selective enrichment of CML-modified peptides coupled to mass spectrometry for identification and quantification. We applied this approach to two cellular models (mouse embryonic fibroblasts (MEF) and human umbilical vein endothelial cells (HUVEC)), and to organs of young and chronologically aged mice. This strategy allowed us to identify specific sites of carboxymethylation (CM) on proteins including CML and carboxymethylated protein N-termini (CM-Nterm). Our data reveal that CML accumulation in primary endothelial cells inhibits proliferation, which is due to both altered expression of cell cycle regulators, and glycation of tubulins associated with impaired microtubule dynamics.

## Results

**An enrichment strategy for the identification of CML-modified peptides.** Aiming to identify sites of CML modification on proteins, we developed a proteomic workflow employing an antibody-based enrichment of CML-modified peptides coupled to mass spectrometry (CMLpepIP). We applied CMLpepIP to MEF and HUVEC treated with different concentrations of GO, a cell-permeable dialdehyde (Fig. 1a). The uptake of GO in cells was validated in HUVEC, where intracellular levels increased ~12 fold after treatment with 1 mM GO for 24 h (Fig. 1b). In both cell types, GO treatment resulted in a dose-dependent increase of total CML levels, as verified by immunoblot (Fig. 1c and Supplementary Fig. 1b), and amino acid absolute quantification using liquid chromatography mass spectrometry (LC-MS) (Fig. 1d). We noted a sharp increase in CML levels at the respective highest concentrations of GO, likely due to a saturation of cell detoxifying systems.

To assess the efficiency of our enrichment protocol, we compared the fraction of peptide-spectrum matches (PSMs) assigned to carboxymethylated peptides between elutions from the enriched and control samples. When CMLpepIP was applied to GO-treated cells, we achieved up to seven times more PSM from CML-containing peptides as compared to controls. With a lower concentration of GO, the gain of modified peptides was reduced and, in any case, the fraction of PSM assigned to CML peptides never exceeded 5% (Fig. 1e). This indicates that CMLpepIP enables only partial enrichment of CML-modified peptides, especially in high complexity samples where CML peptides occur at low concentration.

Using CMLpepIP, we identified 1113 unique carboxymethylation sites in MEF and 307 in HUVEC (Supplementary Data 1), of which 1090 (MEF) and 300 (HUVEC) were CML sites and 23 (MEF) and 7 (HUVEC) were CM-Nterm sites. Overlapping sets of CM sites were detected across conditions and independent experiments (Supplementary Fig. 1c, d). We performed parallel reaction monitoring (PRM) measurements using isotopically labeled spike-in peptide standards to validate a subset of the identified CML sites (Supplementary Data 1). Co-elution of endogenous (light) and standard (heavy) peptides enabled quantitation relative to the spike-in standard across conditions that recapitulated the absolute levels of CML measured by LC-MS (Fig. 1f, g).

A subset of conserved CM sites was found to be modified in both MEF and HUVEC (Fig. 1h). These sites localized on proteins involved in translation, RNA splicing, protein refolding and cytoskeletal constituents (Fig. 1i and Supplementary Data 1). Since long-lived proteins are postulated to be the main targets of AGEs[29–31], we used published turnover data[32] and investigated the turnover distribution for modified and non-modified proteins in MEF and HUVEC. In both cell types, proteins with slower turnover were more likely to be affected by carboxymethylation, independently of their subcellular localization (Fig. 1j left, 1k left), indicating slow turnover as an important determinant of CM modification. Although we were able to detect CM sites across the entire dynamic range of identified proteins in both cell types (Supplementary Fig. 1e), we noted that protein abundance correlated with CM modifications, with high abundant proteins being more often detected as modified (Fig. 1j right, 1k right).

Taken together, CMLpepIP enabled to identify specific and conserved targets of CM-modification in two different primary culture systems. Slow turnover and high abundance appear to be key factors in driving protein carboxymethylation in cultured cells.

**Sites of protein carboxymethylation in mouse organs.** Next, we applied CMLpepIP to investigate protein targets of carboxymethylation in vivo. We analyzed CM sites in heart, kidney, and liver from young (3–4 months) and old C57BL/6J mice (26–33 months), and, in parallel, changes of protein abundance using quantitative MS (Fig. 2a). By combining data from young and old mice, we identified 198, 71 and 105 unique CML sites and

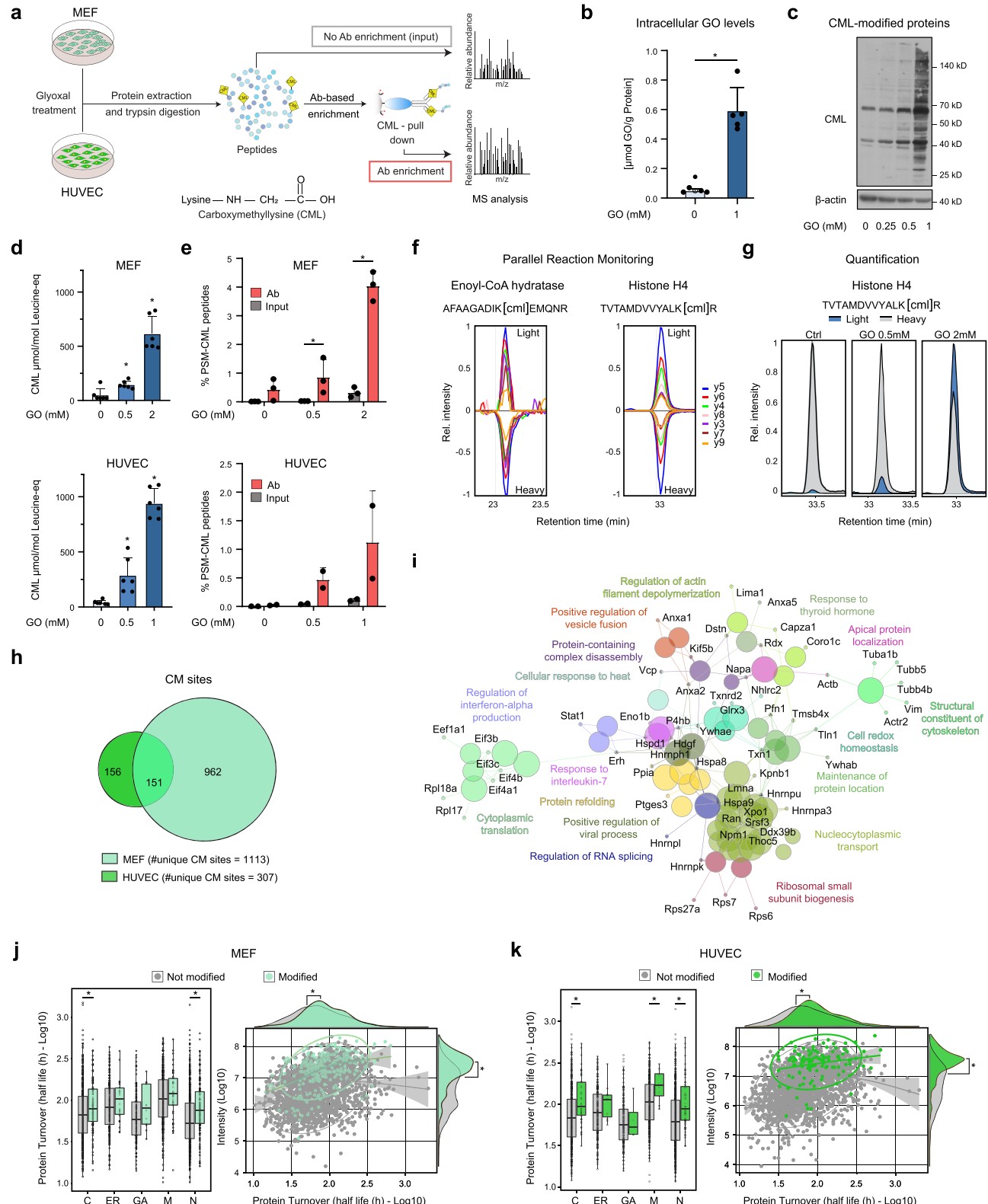

81, 116, and 88 CM-Nterm sites in heart, kidney, and liver, respectively (Supplementary Data 2). Using PRM with spike-in heavy peptides, we validated 16 out of 18 CML sites, for which we had successfully developed PRM assays (Supplementary Fig. 2a, b). Carboxymethylation appears to target proteins largely in an organ-specific manner (Fig. 2b). However, we identified a subset of proteins modified in all the organs tested. These included

mainly mitochondrial and nuclear proteins, such as histones (Supplementary Fig. 2c, d). The affected proteins participate in biological processes related to oxidative phosphorylation in all organs, myofibril assembly in the heart, and detoxification in the liver (Fig. 2c). Similar to what we observed in cultured cells (Fig. 1j, k), carboxymethylation in the heart occurred preferentially in proteins with slow turnover (Supplementary

**Fig. 1 Antibody-based enrichment of CML-modified peptides. a** Workflow for the identification of CML-modified peptides. Ab: antibody **b** Absolute quantification of intracellular GO levels in HUVEC after treatment with GO (1 mM, 24 h). $n = 5$, mean + SD, * $p < 0.01$, paired t-test, two tailed. **c** Immunoblot for CML-modified proteins from HUVEC treated with GO for 48 h. Densitometric quantification is shown in Supplementary Fig. 1b. $n = 3$. **d** Quantification of total CML levels in MEF and HUVEC treated with GO for 8 h or 48 h, respectively. $n = 6$, mean + SD, * $p < 0.05$ vs. control, one-way ANOVA using Geisser-Greenhouse correction. **e** Percentage of peptide-spectrum matches (PSM) containing CML modification among different conditions. $n = 3$ (MEF), $n = 2$ (HUVEC), mean + SD, * $p < 0.05$. Multiple t-test corrected using Holm–Šidák method, two tailed. **f** Validation of CML-modified peptides by parallel reaction monitoring (PRM) using heavy spike-in peptides. **g** Quantification of a CML-modified peptide from histone H4 by PRM in MEF treated with GO for 24 h. **h** Venn diagram showing the overlap between unique CM sites identified in MEF and HUVEC. **i** Enrichment of Gene Ontology biological processes among CM sites identified in both HUVEC and MEF. The enrichment was performed using the Cytoscape App ClueGO. **j, k** Left, protein turnover of CM-modified proteins grouped according to their subcellular localization, as defined by Gene Ontology annotation. Turnover data were taken from Mathieson et al.[32]. C: cytoplasm; ER: endoplasmic reticulum; GA: Golgi apparatus; M: mitochondria; N: nucleus. Right, scatterplot comparing protein abundance (intensities, averages of $n = 3$ (MEF), $n = 2$ (HUVEC)) and protein turnover of CM-modified proteins. Data were fitted with a generalized additive model and the shades represent 95% confidence intervals. * $p < 0.05$, Wilcoxon Rank Sum test with continuity correction, two-sided. Source data are provided as a Source Data file. Specific $p$ values are listed in Supplementary Data 6. Related to Supplementary Fig. 1 and Supplementary Data 1.

Fig. 2e). Analysis of CML sites on protein structures revealed that most of the modified residues are surface exposed and often located in alpha helixes (Fig. 2d). Consistent with the non-enzymatic nature of these modifications, we did not find any sequence motif shared among CML-modified peptides. Finally, since CML modification occurs on lysines and these residues are targets for other posttranslational modifications (PTM), we assessed the overlap of CML sites and other known PTM. We found that approximately 80% of the identified CML sites (260 out of 313) occur on residues that are known to be modified by other PTM, primarily ubiquitination and acetylation (Fig. 2e).

We next investigated age-related changes of protein abundance in the same tissues using Tandem Mass Tag (TMT)—10plex labeling (Fig. 2a). We used principal component analysis to confirm distinct proteome signatures between young and old mice for all the three organs (Supplementary Fig. 2f). By applying gene set enrichment analysis, we identified pathways affected by protein abundance changes in aged organs (Supplementary Fig. 2g and Supplementary Data 2). Among these, a pathway related to AGE signaling via RAGE including proteins such as NFκB1, MAPK9 and AKT1, was enriched with aging in both heart and kidney. In addition, we found a general trend for a reduced abundance of proteins involved in the GLO1 network in old mice in all organs tested (Fig. 2f). This was accompanied by a decrease of GLO1 activity in organ lysates from old mice (Fig. 2g). Notably, the decrease of enzyme activity was more pronounced than the reduction in protein levels, suggesting that additional mechanisms, e.g., limited availability of cofactors such as glutathione[33] might contribute to the observed decrease in GLO1 activity in old organs.

Inspired by this and by the increase of total CML levels in old organs[2,8,23] (Fig. 2h and Supplementary Fig. 2h), we tested whether we could detect an age-dependent elevation of CML modification at specific sites. Using PRM assays on total lysates, we demonstrated an age-dependent raise in the level of CML modification of histone H4 at position K92 that exceeded the abundance change observed at the protein level in heart and kidney (Fig. 2i). However, no age-related changes in the levels of two other CML sites, i.e., K263 of adenine nucleotide translocase type 1 and K154 of ATP synthase subunit γ, were observed (Fig. 2j).

Together, these data show that the protein targets of carboxymethylation are largely organ-specific, and that the levels of CML modification increase with aging in a site-specific manner, likely because of reduced levels and activity of dicarbonyl-detoxifying enzymes in old organs.

**CML modification targets the UPS**. To better understand the functional consequences of protein carboxymethylation, we performed experiments to characterize cellular responses to GO treatment. Since we observed that some of the CM targets are proteins

involved in protein quality control (Supplementary Fig. 3a), we hypothesized that accumulation of GO impairs proteostasis. To address this question, we applied a data-independent acquisition (DIA) method to monitor proteome-wide changes of protein abundance in MEF treated with different GO concentration for 8 h and 24 h (Fig. 3a and Supplementary Fig. 3b). Short treatment (8 h) with 0.5 mM and 2 mM GO induced only few significant alterations, however, a longer treatment (24 h) triggered a major proteome response that was dose-dependent (Fig. 3b).

At this time point, the abundance of CM-modified proteins was significantly increased (Fig. 3c), while KEGG pathways related to the UPS, mainly including ubiquitin-conjugating enzymes, were decreased (Fig. 3d and Supplementary Fig. 3c). Conversely, we found a significant increase of components of the 26S proteasome induced by GO treatment in both MEF and HUVEC (Fig. 3e left and Supplementary Fig. 3d). This was accompanied by an increased proteasomal activity in MEF after 0.5 mM GO but not after 2 mM (Fig. 3e right). In parallel, GO induced a dose-dependent accumulation of ubiquitinated proteins (Fig. 3f). We hypothesized that the lack of proteasome activity increase at high doses of GO might be due to a functional impairment of the proteasome itself. To test this, we evaluated the thermal stability of proteasomal proteins in MEF using thermal proteome profiling[34]. This analysis revealed that GO treatment negatively impacted the thermal stability of the proteasome, especially of components of the 19S regulatory particle (Fig. 3g and Supplementary Fig. 4a), while the overall thermal stability of proteins remained comparable across the tested conditions (Supplementary Fig. 4b, c and Supplementary Data 3). In addition, we identified a subset of proteasomal proteins and other members of the UPS, including ubiquitin itself, to be carboxymethylated both in cells treated with GO and in mouse organs (Supplementary Fig. 3a). To test whether carboxymethylation directly influences proteasome activity, we treated purified 26S proteasome with GO, and assessed chymotrypsin-like activity, the central proteasomal peptidase activity[35], using a native gel assay. We confirmed induction of CML modification by immunoblot (Fig. 3h middle). However, we were not able to detect any change of proteasome activity induced by GO treatment in vitro under the applied experimental conditions (Fig. 3h right).

Taken together these data describe a specific and dose-dependent proteome response to GO exposure. This is characterized by accumulation of CM-modified proteins and an induction of compensatory mechanisms including the UPS. However, the UPS may become saturated at higher GO concentration leading to the accumulation of CM-modified and ubiquitinated proteins.

**Glyoxal inhibits proliferation of primary human endothelial cells**. In endothelial cells, CM modifications induced by GO (1 mM, 48 h) occurred mainly in proteins of the nucleus, the

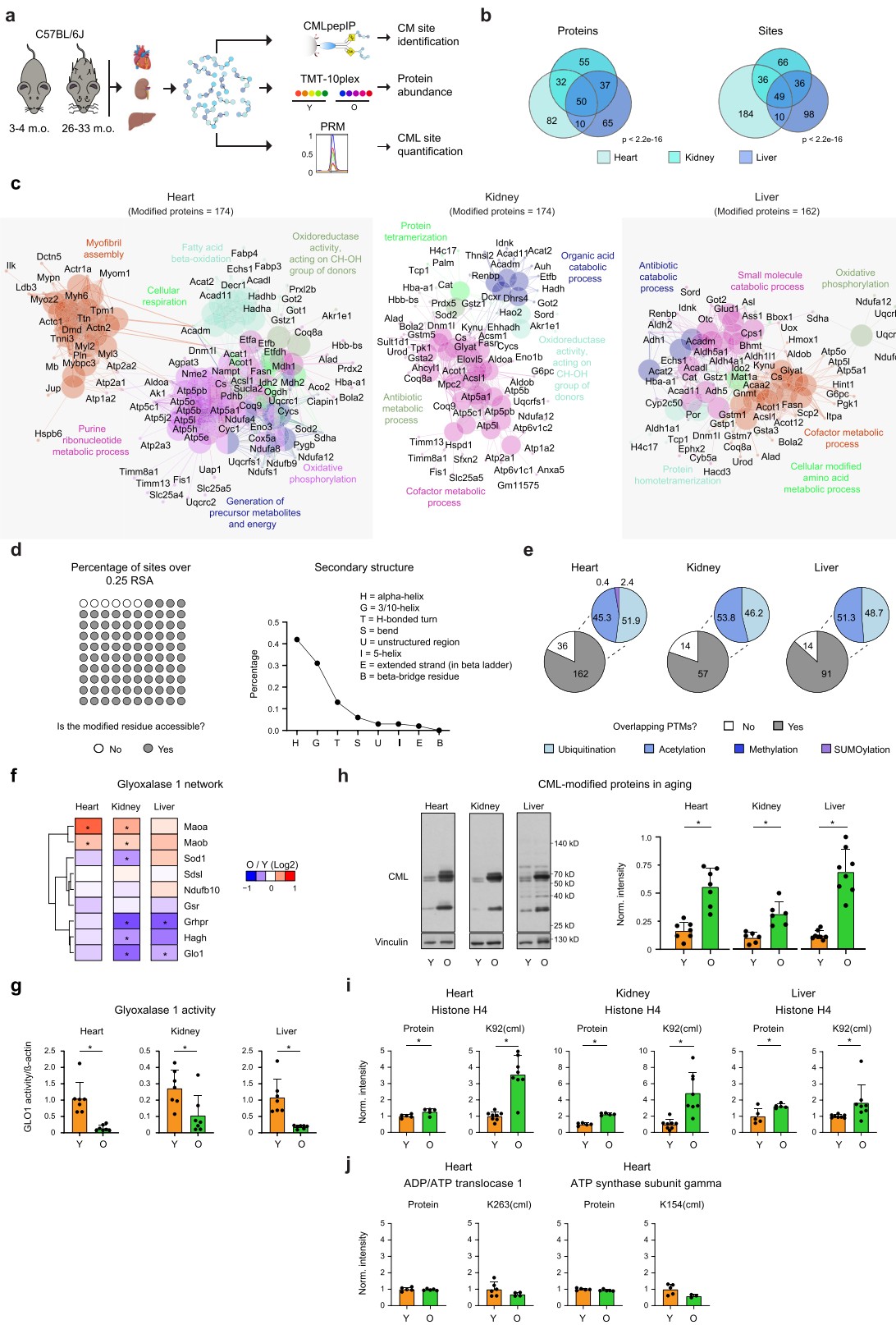

cytoskeleton and the mitochondria (Fig. 4a and Supplementary Data 4), while GO-initiated changes in protein abundance comprised an upregulation of stress response proteins, e.g., heme oxygenase 1, and a downregulation of processes related to proliferation and growth (Fig. 4b). This was accompanied by an inhibition of endothelial cell proliferation as demonstrated by reduced cell numbers and lower incorporation of BrdU in the

presence of serum or after stimulation with basic fibroblast growth factor (bFGF) (Fig. 4c, d). Furthermore, angiogenic sprouting in response to vascular endothelial growth factor (VEGF) was impaired (Fig. 4e). GO-treated cells did not show signs of apoptosis (Supplementary Fig. 5a).

Since mitochondrial proteins were found to be CML-modified (Fig. 4a and Supplementary Data 4), we first asked whether

**Fig. 2 The CM-modified proteome in mouse organs. a** Workflow for the analysis performed on mouse organs from young (Y) and old (O) mice. Image was created with BioRender.com. Tandem Mass Tag (TMT) was used for monitoring protein abundance, CMLpepIP for CM site identification and PRM for quantification. **b** Venn diagram of CM-modified proteins (left) and sites (right) identified in heart, kidney and liver from young and old mice combined. For each tissue, only unique entries were considered for the overlap. *P*-value was calculated by Fisher's Exact Test. **c** Enrichment of Gene Ontology biological processes among CM sites identified from heart, kidney and liver. Enrichment was performed using the Cytoscape App ClueGO. **d** Left, percentage of surface exposed CML sites. A 25% threshold for relative surface area (RSA) score was used to define sites as surface exposed. Right, percentage of CML sites located in different protein secondary structures. **e** Overlap between CML sites and other known PTMs. Gray scale pie charts indicate the number of overlapping CML sites and colored pie charts the overlapping PTM classes (shown as percentages). Known PTM were obtained from Minguez et al.[92] and Hornbeck et al.[93]. **f** Effect of aging on the abundance of proteins of the glyoxalase 1 (GLO1) network. The network was extracted from https://string-db.org using GLO1 as input. $n = 5$, * adj. $p < 0.05$, unpaired t-test, two tailed with Benjamini–Hochberg correction for multiple testing, as implemented in *limma*. **g.** GLO1 activity tested on heart, kidney and liver lysates from young (Y) and old (O) mice. $n = 7$, mean + SD, * $p < 0.05$, unpaired t-test, parametric, two tailed. **h** Immunoblot (left) and densitometry-based quantification (right) for CML-modified proteins in mouse organs from young and old mice. $n = 6$ (kidney, heart), $n = 8$ (liver), mean + SD, * $p < 0.01$, unpaired t-test, two tailed. **i, j** Normalized protein intensity of histone H4, ADP/ATP translocase type 1 and ATP synthase subunit γ from TMT experiment (protein—left panel) and intensity of CML-modified peptides from PRM (right panel). $n = 5$ for protein abundance, $n = 8$ for CML-modified peptides, mean + SD, * $p < 0.05$, unpaired t-test, two tailed. Source data are provided as a Source Data file. Specific p values are listed in Supplementary Data 6. Related to Supplementary Fig. 2 and Supplementary Data 2.

alterations in cellular energy production would underlie the observed inhibition of cell proliferation. Seahorse analyses revealed a decrease in basal respiration and mitochondrial ATP production (Fig. 4f and Supplementary Fig. 5b) as well as an upregulation of glycolysis in HUVEC treated with GO (1 mM, 48 h) (Fig. 4g and Supplementary Fig. 5c). The latter, together with an increased mitochondrial biogenesis (Fig. 4h), likely compensated the reduction of mitochondrial ATP production, since total cellular ATP was not altered (Fig. 4i). These data show that the cellular energy homeostasis was maintained despite the exposure of cells to dicarbonyl stress, and that reduced cell proliferation was not due to metabolic limitations.

**Glyoxal induces cell cycle arrest in endothelial cells**. To further clarify the mechanisms underlying inhibition of cell proliferation, we analyzed the effect of gloxal on cell cycle distribution. Applying a triple staining method, we found that treating endothelial cells with 1 mM GO for 24 h triggered an arrest in G0/G1 and G2 phases, while the percentage of cells in the S and M phases was significantly decreased (Fig. 5a). Some of these alterations, especially the decrease in mitosis, were already seen after 4 h of GO treatment, although at a lower degree. In accordance with the lower entry of cells into S and M phases, we observed a decrease of the cell cycle markers cyclin A (S phase) and phosphorylated H3 (S10) (p-H3 (S10), M phase) (Supplementary Fig. 6a). In addition, tracking cell proliferation dynamics revealed a slower proliferation in response to the high dose of GO (Fig. 5b).

We hypothesized that the observed cell cycle arrest induced by GO was due to both, specific glycation of proteins involved in cell cycle control and altered expression of cell cycle regulators (Fig. 5c and Supplementary Data 5). Interestingly, glycation was mostly observed on proteins controlling G2 and M phases, for instance, on tubulin α and β chains, while alteration of protein expression was mainly related to proteins involved in the regulation of S and G1 phases (Fig. 5c). As examples, 8 out of 10 subunits of the replicative helicase MCM2-7/GINS/CDC45 and several factors involved in lagging strand DNA synthesis such as subunits of DNA polymerases α and δ, FEN1 and DNA ligase 1 were downregulated, whereas the cyclin-dependent kinase inhibitor 1A (CDKN1A)/p21 was upregulated. In addition, an increased abundance of microtubule-associated proteins (MAPs) such as MAP1B and MAP4 and of microtubule regulators was observed (Supplementary Fig. 6b).

One pathway through which the expression of cell cycle regulators is controlled is the DNA damage response. Indeed, GO (0.5–1 mM) triggered this pathway as shown by phosphorylation of the checkpoint kinase 1 (CHK1), an increased percentage of cells with γH2AX foci (Fig. 5d, e) and an upregulation of the p53/

p21 pathway, in line with the proteomics data (Fig. 5f and Supplementary Fig. 6c). The expression of p16, another inhibitor of cyclin-dependent kinases, was also enhanced (Fig. 5f and Supplementary Fig. 6c). After treatment with 1 mM GO, a small proportion of cells (~12%) acquired a senescent phenotype, as shown by positive staining for senescence-associated beta-galactosidase (SA-β-Gal) (Supplementary Fig. 6d). This may be related to persistent dicarbonyl and oxidative stress[36] since GO led to increased mitochondrial and cytosolic formation of reactive oxygen species (ROS) (Fig. 5g).

**CML modification of tubulin affects its acetylation and polymerization**. To understand how glycation of proteins may affect cell cycle progression, we focused on tubulins, which exhibited CML modification of different lysines upon treatment of cells with GO and in mouse organs (Fig. 6a and Supplementary Fig. 7a). CML-modified tubulin showed more intensively stained filaments when compared to tubulin in control cells (Fig. 6b) and an increased resistance against the depolymerizing agent nocodazole (Fig. 6c). To understand whether CML modification of tubulin affected its polymerization behavior, we exposed purified tubulin to GO. This triggered CML modification at sites that overlap with the ones identified in GO-treated cells (Fig. 6d and Supplementary Fig. 7b) and led to a significantly lower polymerization in vitro (Fig. 6e). These data suggest changes of microtubule dynamics upon GO treatment, which may contribute to the observed reduction of cell proliferation by impeding mitosis. To test this, we treated cells, which were synchronized in early S phase, with GO and monitored time-dependent expression of the mitotic marker p-H3 (S10) in immunoblotting experiments. In control cells, p-H3 (S10) signals were detected between 16 h and 18 h after releasing the cell cycle block, while GO-treated cells showed reduced intensity of p-H3 (S10) signals indicating a lower number of mitotic cells, which was independent from inhibition of G1 and G1/S transition (Fig. 6f).

CML modification of tubulin may alter microtubule dynamics directly or indirectly by interfering with other lysine modifications. Since tubulin is known to be regulated by acetylation of lysines[37] (Fig. 6a), we hypothesized that this may be influenced by CML modification. We were able to demonstrate opposite changes in CML modification and acetylation at lysine 58 (K58) of the tubulin β−4B chain in GO-treated cells. While GO induced CML modification of K58 at a stoichiometry of ~0.009% (Fig. 6g), K58 acetylation was concomitantly reduced (Fig. 6h). These data suggest that CML modification of tubulin is likely to have an impact on acetylation and may thereby affect its interactions with microtubule-binding proteins and regulatory

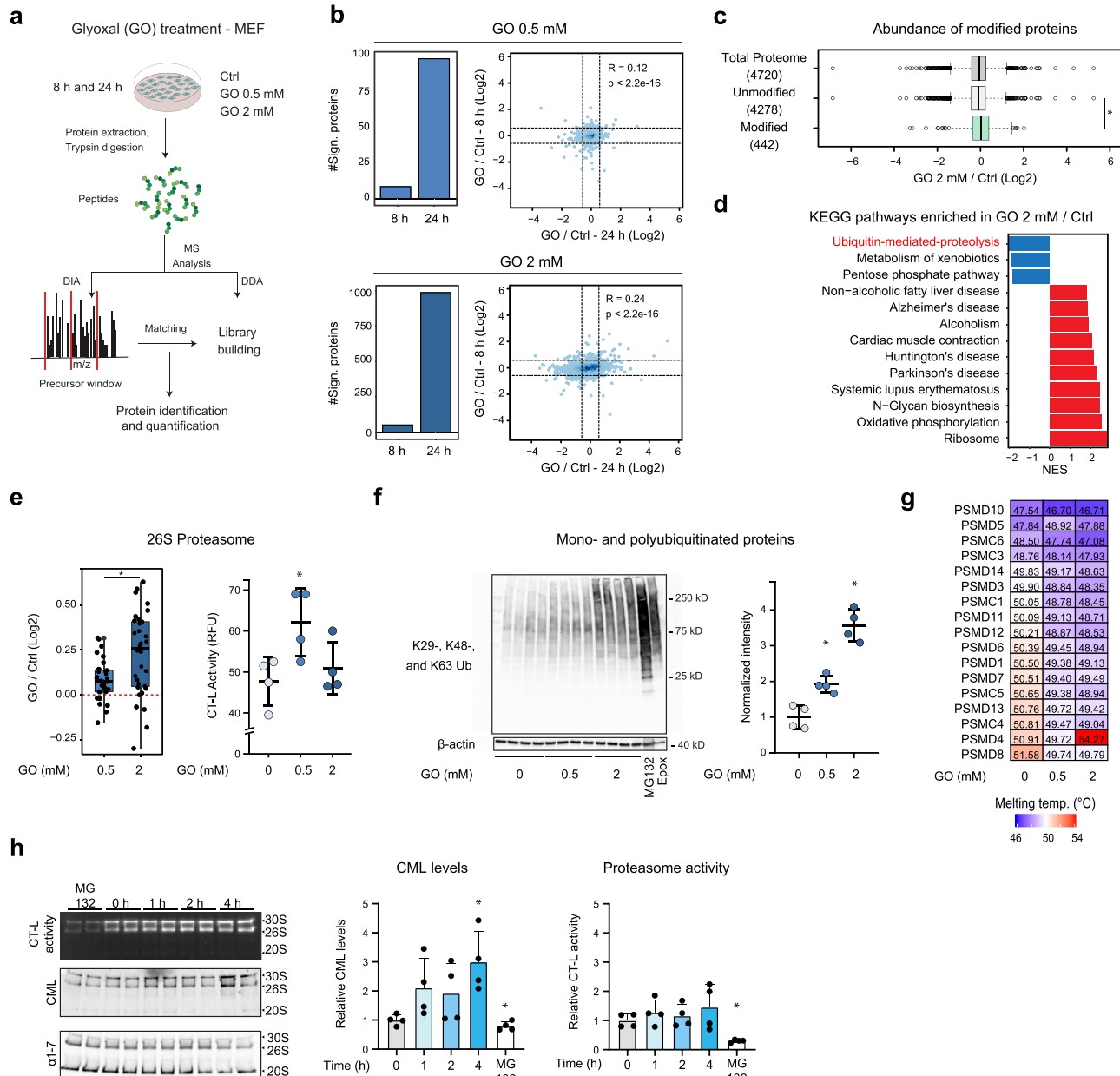

**Fig. 3 Alteration of proteostasis induced by glyoxal in MEF. a** Workflow for the analysis of proteome changes induced in MEF by GO. **b** Left, number of significant proteins (absolute log2 fold change>0.58 and $q < 0.05$) after 8 and 24 h GO treatment. Right, scatterplots comparing the log2 fold changes induced by GO at 8 or 24 h. Dashed lines represent the log2 fold change cut-offs used (±0.58). $n = 5$ (8 h), $n = 4$ (24 h). Pearson's product-moment correlation, two-sided. **c** Distribution of protein fold changes for 2 mM GO (24 h) vs. control (Ctrl). Average values of $n = 4$ were compared, * $p < 0.001$, Wilcoxon rank sum test with continuity correction, two-sided. **d** Gene Set Enrichment Analysis (GSEA) for KEGG pathways based on protein fold changes induced by 2 mM GO (24 h). Normalized enrichment score (NES) indicates pathways enriched among proteins that increase (red) or decrease (blue) upon GO treatment (FDR < 0.05). All quantified proteins were ranked according to their log2 fold change and used as input for GSEA. **e** Left, boxplot of fold changes in expression of members of the 26S proteasome after GO treatment (24 h). * $p < 0.01$, Wilcoxon Rank Sum test with continuity correction, two-sided. Right, proteasome chymotrypsin-like (CT-L) activity in MEF treated with GO (24 h). $n = 4$, mean + SD, * $p < 0.05$, one-way ANOVA using Geisser-Greenhouse correction. **f** Immunoblot for mono- and polyubiquitinated proteins from MEF treated with GO (24 h). Positive control: proteasome inhibitors MG132 (20 μM, 6 h) or epoxomicin (Epox, 10 nM, 96 h). $n = 4$, mean ± SD, * $p < 0.05$, one-way ANOVA using Geisser-Greenhouse correction. **g** Heatmap representing the melting point temperature of members of 26S proteasome using thermal proteome profiling. **h** Left, proteasome activity assay using native gel electrophoresis (top) after incubation of purified proteasome with GO (1 mM, 0, 1, 2, and 4 h, 37 °C). Immunoblot for CML (middle) and proteasome subunits α1-7 (bottom). Middle, quantification of CML-modification. Right, proteasome chymotrypsin-like activity (CT-L) normalized to α1-7 abundance. Positive control: purified proteasome incubated with MG132 (100 μM, 4 h, 37 °C). $n = 4$, mean + SD, * $p < 0.05$, one-way ANOVA. Source data are provided as a Source Data file. Specific $p$ values are listed in Supplementary Data 6. Related to Supplementary Fig. 3, 4 and Supplementary Data 3.

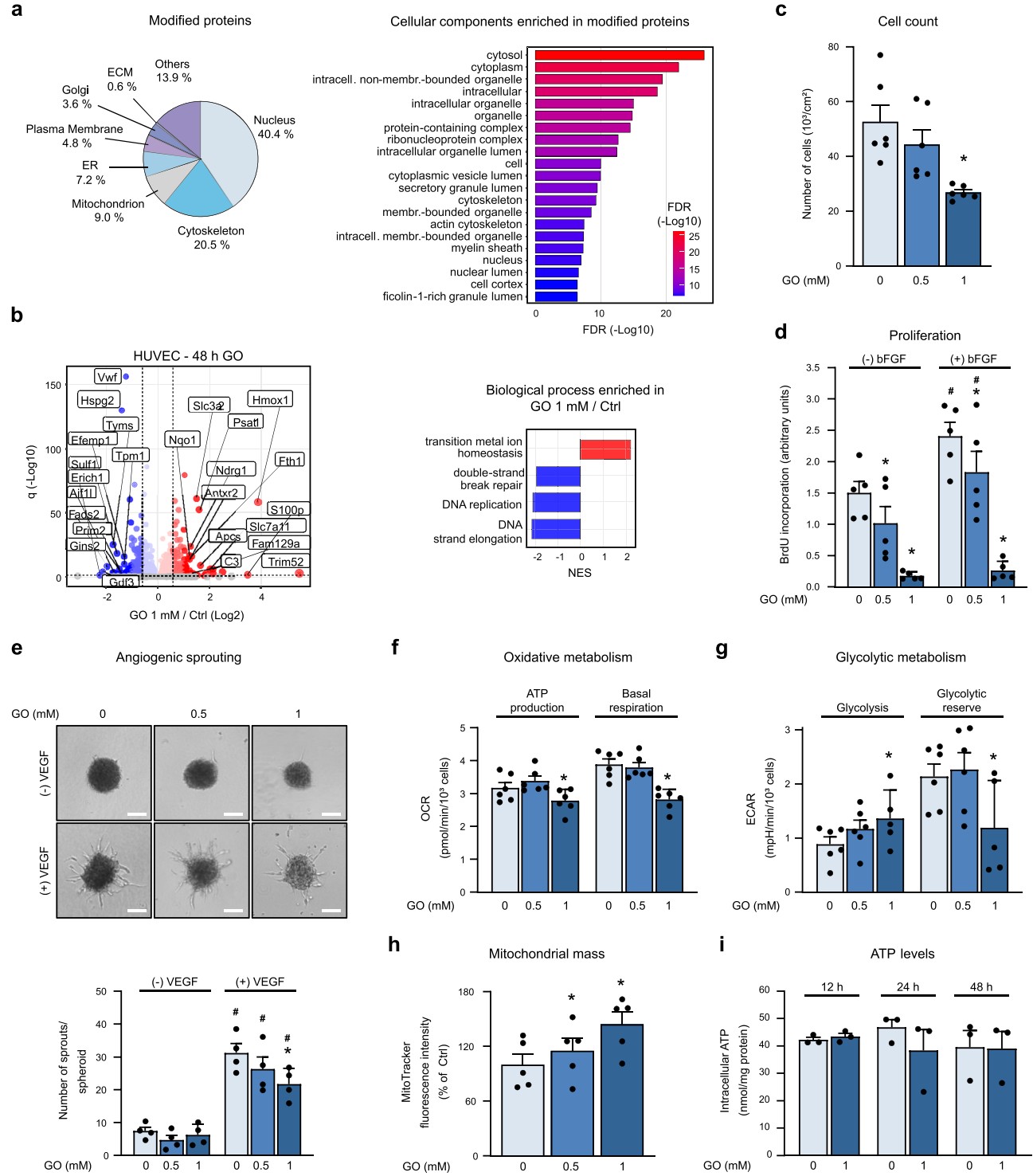

enzymes. Of note, GO treatment of HUVEC also led to reduced tubulin acetylation at lysine 40 (K40) (Supplementary Fig. 7c, d), which is known to protect microtubules from mechanical damage and to facilitate self-repair[38,39]. However, we were not able to detect CML modification of tubulin K40 for technical reasons, i.e., the peptide containing K40 was too long to be detected by our LC-MS workflow.

**Long-term treatment with lower doses of glyoxal inhibits endothelial proliferation.** To test whether inhibition of endothelial proliferation induced by short-term incubation with 0.5 or

1 mM GO may occur under pathophysiological relevant conditions, we treated cells with lower concentrations of GO (1–100 μM) for a longer period (14 days). We observed an inhibition of cell proliferation, which was significant after 14 days for all doses, and was characterized by a concentration-dependent increase in doubling time and a decrease in proliferation rate (Fig. 7a). Mass spectrometry analysis of long-term treated cells revealed an overall reduced proteome response compared to cells exposed to higher dose of GO for short time (Supplementary Fig. 8a) and almost no influence on the abundance of proteins involved in cell cycle regulation (Supplementary Fig. 8b). However, treatment of endothelial cells with 100 μM GO for 14 days

**Fig. 4 Glyoxal impairs the proliferation of HUVEC. a** Left, CM-modified proteins annotated to different cellular compartments according to Gene Ontology annotation after GO treatment (1 mM, 48 h). ECM: extracellular matrix; ER: endoplasmic reticulum. Right, Gene Ontology cellular component terms enriched in CM-modified proteins. False Discovery Rate (FDR) < 0.05. **b** Left, Volcano plot depicting proteins that significantly increase (red) or decrease (blue) abundance upon GO treatment (1 mM, 48 h) or remain unchanged (gray). Horizontal dashed line indicates a significance cut-off of q < 0.05 and vertical dashed lines an absolute fold change (log2) > 0.58. $n = 4$. Right, Gene set enrichment analysis for Gene Ontology biological process terms based on protein fold changes. Terms enriched among increased (red) or decreased (blue) proteins are shown. FDR < 0.05; NES: normalized enrichment score. c-i. HUVEC were treated with GO for 48 h. **c** Cell numbers were determined. $n = 6$. **d** Cells were stimulated with basic fibroblast growth factor (bFGF) (50 ng/ml, 24 h) and BrdU incorporation was measured. $n = 5$. **e** Spheroids were generated, embedded, and stimulated with vascular endothelial growth factor (VEGF) (50 ng/ml, 24 h). Representative pictures and sprout numbers per spheroid are shown. Scale bar=100 μm. $n = 4$. **f**, **g** Oxygen consumption rate (OCR) (**f**) and extracellular acidification rates (ECAR) (**g**) were measured via Seahorse technology and mitochondrial and glycolytic parameters calculated. $n = 6$. **h** Cells were stained with MitoTracker and analyzed by flow cytometry. $n = 5$. **i** Intracellular ATP levels were measured in cell extracts. $n = 3$. **c–i** Data are represented as mean + SEM. Statistical significance was analyzed using one-way or two-way repeated measurement ANOVA corrected via Holm–Šidák method. * $p < 0.05$ vs. control, # $p < 0.05$ vs. respective non-bFGF (**d**) or non-VEGF-treated sample (**e**). Source data are provided as a Source Data file. Specific $p$ values are listed in Supplementary Data 6. Related to Supplementary Fig. 5 and Supplementary Data 4.

was sufficient to induce cell cycle perturbations such as increased percentage of cells in G2 and S phases. This suggests a prolonged G2 phase, which may indicate a developing cell cycle arrest similar to, but less pronounced than the G2 arrest induced by 1 mM GO for 24 h (Fig. 7b) as well as a prolonged S phase, which is different from the G0/G1 arrest seen after 1 mM GO. We were not able to detect markers of DNA damage such as phosphorylation of CHK1 or increase of CDKN1A/p21 and only observed a mild increase of mitochondrial or cytosolic production of ROS (Fig. 7c–e). Of note, long-term treatment with low doses of GO increased the staining intensity of microtubules (Fig. 7f), indicating an alteration of tubulin function similar to what we observed after short-term incubation with higher doses and in vitro (Fig. 6b, e). Using PRM assays on CMLpepIP samples, we were able to detect CML modification at K58 after long-treatment of cells with 100 μM GO, even though to a lower extent than after short treatment with 1 mM GO (Fig. 7g, h). The acetylation of K40 was not altered by prolonged exposure to low doses of GO (Supplementary Fig. 8c). Together, these data indicate that glycation of tubulin may occur also under pathophysiological conditions triggering tubulin dysfunction and inhibition of cell proliferation.

## Discussion

The impairment of protein homeostasis is a hallmark of aging and age-associated diseases[40–42]. While age-related changes in protein abundance and turnover have been widely described, less is known about the extent and functional impact of PTM. Here, we focused on CML modification of proteins, a non-enzymatic PTM belonging to the class of AGEs[8]. We developed an antibody-based method for the enrichment of CML-modified peptides coupled to mass spectrometry for identification and quantification. The mapping of CML sites together with studying the total proteome in cultured cells and aging mouse organs allowed us to link carboxymethylation of proteins to specific cellular functions and to better understand how these processes may contribute to aging and age-associated metabolic diseases. We found that carboxymethylation occurs more often in high abundant proteins that display slower turnover. This corresponds to previous studies showing that global AGEs accumulate particularly in tissues characterized by slow protein turnover, such as crystallin lens, cartilage and skin[29–31]. Interestingly though, we detected carboxymethylation of proteins that spanned four orders of magnitude of protein abundance and, conversely, we did not identify modification for all the most abundant proteins. In addition to CML, we also detected carboxymethylation of protein N-termini, especially in mouse organs. We speculate that the higher number of CM-Nterm sites identified in organs might derive from the slower turnover of proteins in vivo as compared to cultured cells. This observation is in line with the results suggesting that slow

protein turnover is associated with an increased likelihood of carboxymethylation. Keeping in mind potential biases due to the limited sensitivity of our approach and specific protein sequences that are not amenable to enzymatic digestion, our data suggest the existence of a subset of proteins that are more prone to undergo carboxymethylation than others.

We extended our analysis to primary organs collected from young and old mice and showed that our enrichment strategy can identify hundreds of physiologically occurring CM sites, some of these previously reported in an independent study in the mouse heart[25]. The modified proteins were enriched for mitochondrial proteins, histones, cytoskeletal proteins and enzymes involved in detoxification, which might, at least in part, derive from sub-cellular differences in local pH or concentration of metabolites that promote AGE formation[43]. We additionally developed targeted proteomics assays that enable the quantification of a subset of CML sites in tissues without prior peptide enrichment. These analyses revealed that the level of CML modification for some sites, i.e., histone H4-K92 (corresponding to K91 in the mature protein), increased with aging in different organs, while others remained constant. Thus, preferential modification of certain proteins occurs, as previously suggested[44], illustrating the need for targeted investigation of AGE modification sites. Interestingly, increased CML modification in old mice was paralleled by downregulation of the glyoxalase network, which may, at least in part, explain the observed increase of CML-modified proteins in old mice. A similar age-associated decline of glyoxalases in rodents and humans was reported earlier[45,46].

Our data reveal a substantial overlap between the identified CML sites and other PTM, particularly acetylation and ubiquitination, suggesting that CML modification may interfere with biological processes mediated by these PTM. For instance, global glycation of histone H3 has been shown to compete with acetylation and methylation thereby disrupting chromatin architecture[15,16]. The age-dependent increase in CML modification of histone H4-K92 as observed in our study may have similar consequences by impeding K92 acetylation and glutarylation, which are involved in the regulation of chromatin structure and dynamics in response to DNA damage[47,48]. It remains to be investigated whether similar age-related changes of carboxymethylation can also occur on histone tails, which were not covered in our analysis due to experimental setup based on tryptic protein digestion.

To investigate other potential mechanisms by which AGEs might affect cellular phenotypes, we focused on perturbation of proteostasis. We show that low doses of GO increased the activity of the proteasome, while high doses did not alter proteasomal activity and led to an accumulation of ubiquitinated proteins. These data are in agreement with earlier reports about the effects of methylglyoxal on the UPS[49,50]. In line with this, low doses of

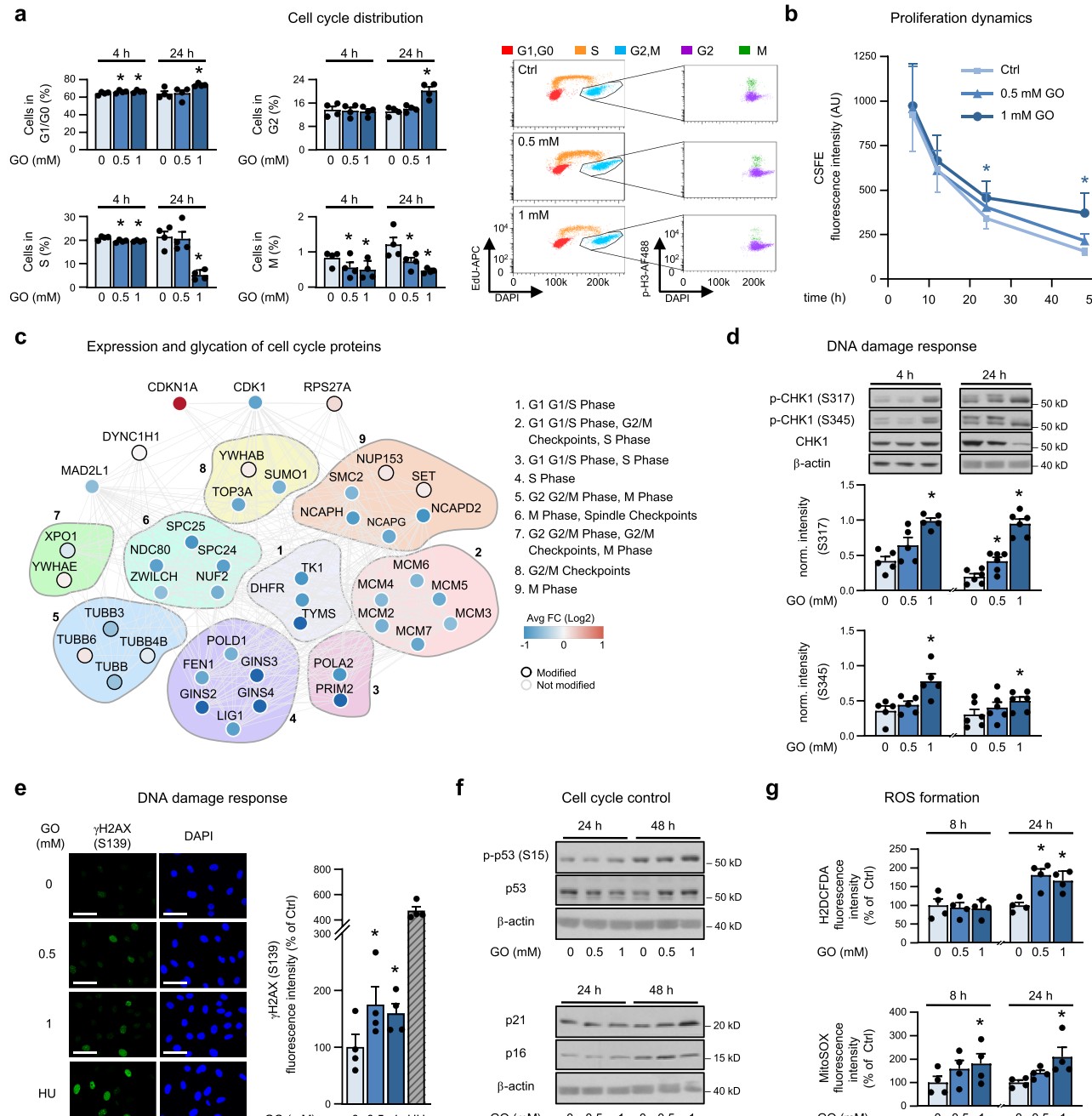

**Fig. 5 Perturbation of the cell cycle induced by glyoxal. a** HUVEC were treated with GO as indicated and subjected to cell cycle analysis applying a triple staining method (5-ethynyl-2'-deoxyuridine (EdU), p-H3 (S10), 4',6-diamidino-2-phenylindole (DAPI)). Ratio of cells in the respective cell cycle phases (left) and dot plots representative for the 24 h-values (right) are shown. $n = 4$. **b** HUVEC were stained with carboxyfluorescein succinimidyl ester (CFSE) and treated with GO. CFSE staining was quantified by flow cytometry. $n = 4$. **c** Cell cycle proteins were chosen according to Reactome annotation divided by their specific role in each phase. Only proteins with significantly altered expression (absolute fold change > 0.58 and $q < 0.05$) or modified by CML are shown. Each dot represents the protein fold change and the black edge signifies proven CML modification. $n = 4$. **d** GO-treated HUVEC were lysed and subjected to immunoblot analysis. $n = 5$ (4 h), $n = 6$ (24 h). **e** Following GO treatment of HUVEC (8 h), immunofluorescence staining of γH2AX (S139) and DAPI was performed. Hydroxyurea (HU, 2 mM, 1 h) served as positive control. Representative pictures and quantification of γH2AX (S139)-positive nuclei are shown. Ctrl: control. Scale bar = 50 μm. $n = 4$. **f** HUVEC were treated with GO, lysed and subjected to immunoblot analysis. Densitometric quantification is shown in Supplementary Fig. 6c. $n = 6$. **g** HUVEC were treated with GO and intracellular (dichlorodihydrofluorescein diacetate (H2DCFDA)) and mitochondrial ROS (MitoSOX) were measured by flow cytometry. $n = 4$. a, b, d, e, g. Bar graphs and the curve diagram show mean + SEM. Statistical significance was analyzed using one-way or two-way repeated measurement ANOVA corrected via Holm–Šidák method. For panel e the positive control (HU) was not included into ANOVA. * $p < 0.05$ vs. respective control. Source data are provided as a Source Data file. Specific $p$ values are listed in Supplementary Data 6. Related to Supplementary Fig. 6 and Supplementary Data 5.

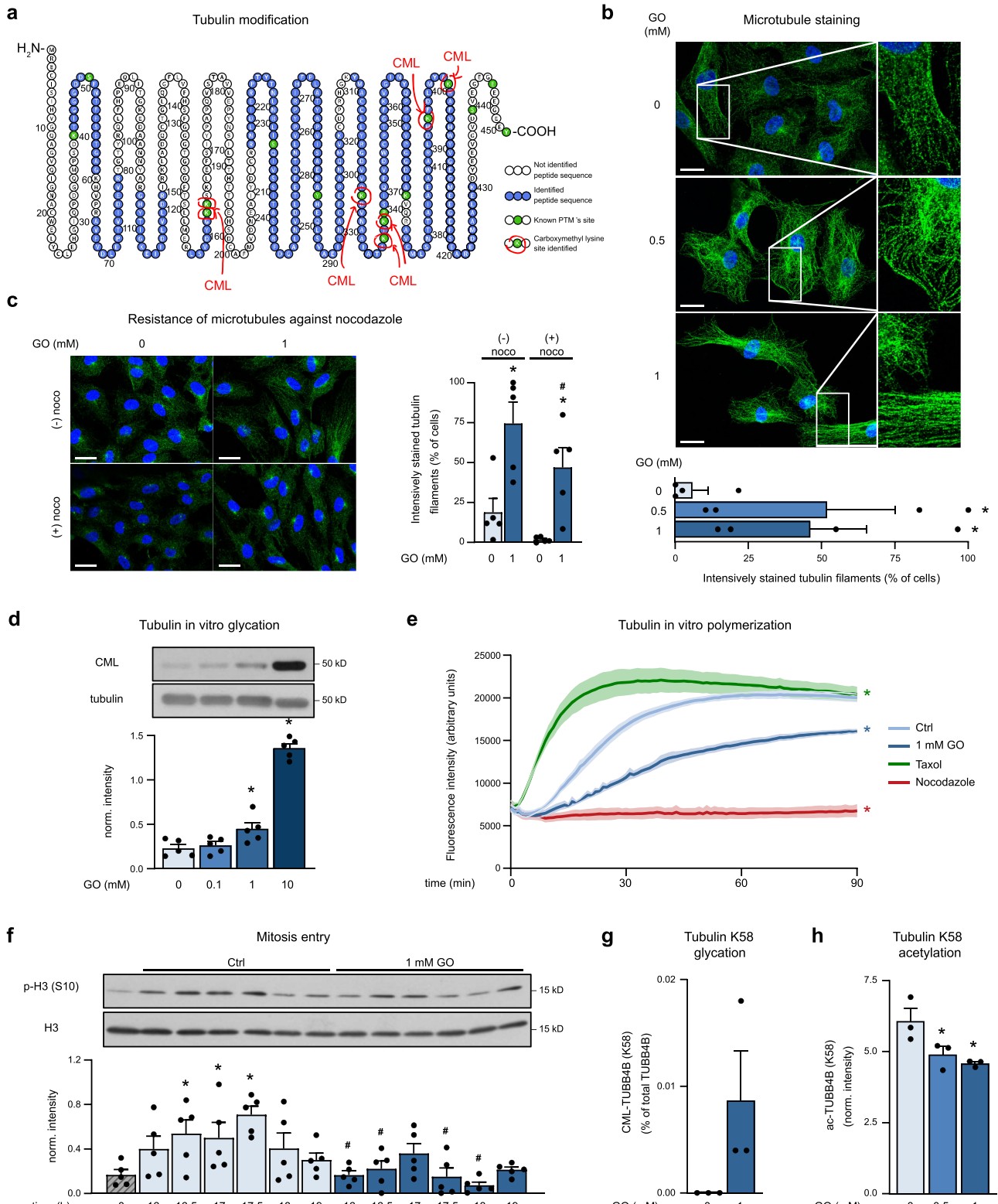

methylglyoxal led to an increased lifespan of *C. elegans*, while high doses (>1 mM) decreased lifespan[28]. Here, we provide evidence for an interference of GO with the thermal stability of the 19S proteasome. However, with the current data, we cannot establish whether the change of thermal stability of the proteasome is due to CM modification or other mechanisms mediated by GO. For example, a similar phenomenon was recently described in response to ATP deprivation in cells[51]. In contrast with a previous report[19], GO was not sufficient to directly inhibit the chymotrypsin-like activity of the 26S proteasome in vitro in our study suggesting indirect mechanisms, e.g., modification of regulators of proteasome activity or excessive accumulation of glycated substrates. Indeed, we identified several enzymes involved in the ubiquitin cycle, including ubiquitin itself, to be direct targets of carboxymethylation both in cells and tissues. Given that proteasome activity is known to decrease during

**Fig. 6 Glyoxal affects posttranslational modification and dynamics of tubulin. a** Scheme of detected CML modification sites for tubulin α-1B chain. **b** HUVEC were treated with GO (48 h) and α/β-tubulin and nuclei (DAPI) were stained. Representative immunofluorescence pictures (upper panel) and the percentage of cells with intensively stained tubulin filaments per high-power field (HPF) are shown (lower panel). Scale bar = 20 μm. $n = 4$. **c** After treatment of HUVEC with GO (1 mM, 48 h), nocodazole (noco) was added (10 μM, 2 min) and α/β-tubulin and nuclei were stained. Representative pictures and the percentage of cells with intensively stained tubulin filaments per HPF are shown. Scale bar = 20 μm. $n = 5$. **d** Purified porcine tubulin incubated with GO for 30 min on ice was analyzed in immunoblots. $n = 5$. **e** Purified porcine tubulin was pre-incubated with GO (1 mM, 30 min, on ice), or vehicle and polymerization induced by addition of cofactors at 37 °C was monitored in a fluorescence-based assay. Controls: taxol (3 μM) or nocodazole (20 μM) added to vehicle-treated tubulin at the start of polymerization. $n = 3$ (taxol, nocodazole), $n = 6$ (control (Ctrl), GO) mean ± SEM. Statistical significance was analyzed using one-way repeated measurement ANOVA corrected via Holm–Šidák method for areas under the curves. * $p < 0.05$ vs. control. **f** HUVEC were synchronized via a double thymidine block and treated with GO subsequent to releasing the block. Cells were analyzed in immunoblots $n = 5$. **g** HUVEC were treated with GO (48 h). The absolute level of CML modification at K58 on tubulin beta-4B chain (TUBB4B) was determined by PRM using the synthetic reference peptide _INVYYNEATGGK[CML]YVPR. The bar plot shows the fraction of CML-modified tubulin at K58 relative to total TUBB4B. Total TUBB4B levels were determined using the synthetic peptide _INVYYNEATGGK. $n = 3$. **h** HUVEC were treated with GO (24 h) and relative K58 acetylation on TUBB4B was quantified by label-free mass spectrometry using the endogenous peptide _INVYYNEATGGK[Ac]YVPR. $n = 3$. **b–d**, **f–h**: Bar graphs show mean + SEM. Statistical significance was analyzed using one-way or two-way repeated measurement ANOVA corrected via Holm–Šidák method. * $p < 0.05$ vs. control, # $p < 0.05$ vs. respective non-nocodazole-treated cells (**c**) or vs. respective non-GO-treated cells (**f**). Source data are provided as a Source Data file. Specific $p$ values are listed in Supplementary Data 6. Related to Supplementary Fig. 7.

aging[40,52–54], the elevated levels of AGEs in old tissues might contribute to proteostasis impairment via alteration of the UPS.

Finally, we investigated the impact of GO on the functional phenotype of endothelial cells, which play an important role in the pathogenesis of age-associated metabolic diseases[55,56]. Previous studies have shown that dicarbonyls induce endothelial dysfunction[57–59] and that AGEs interfere via RAGE-dependent signaling processes in endothelial cells[60,61]. Here, we describe that GO triggered a proliferation inhibition phenotype, which was not related to energy depletion or cytotoxic effects but linked to differentially expressed and/or CML-modified proteins related to cell cycle control. We found that short-term treatment of endothelial cells with 0.5–1 mM GO induced downregulation of various cell cycle regulators, particularly proteins involved in DNA replication, likely because CDKN1A/p21 and p16 were upregulated and restricted S phase entry[62,63]. Possible underlying mechanisms revealed to be an increased mitochondrial and cytosolic ROS formation and the induction of a DNA damage response[64] leading to an upregulation of the p53/p21 pathway[62,64]. In addition, GO may have caused damage by glycation of DNA leading to so-called nucleotide AGEs as recently described[65].

To understand how carboxymethylation of specific proteins may add to the GO-induced inhibition of proliferation in endothelial cells, we focused on tubulin, whose α and β chains exhibited several CML modifications. Tubulin chains assemble to protofilaments, which associate to microtubules and form the mitotic spindle[66]. A major characteristic of tubulin polymers is their dynamic instability, i.e. the rapid transition between growth and shrinkage[67], which is known to be regulated by various PTM such as detyrosination, polyglutamination or acetylation[68]. Our data reveal CML modification as a previously unknown tubulin modification, which is likely to have an impact on microtubule dynamics. In cells, GO treatment triggered the formation of intensively stained, nocodazole-resistant tubulin filaments, while carboxymethylation of purified tubulin decreased the polymerization rate in vitro. Of note, CML sites of purified tubulin overlapped with the sites identified in cells and mouse organs. Although it remains to be clarified whether microtubule dynamics is also affected in vivo, and whether and how this is related to the increased staining intensity of tubulin filaments, our data suggest that CML modification of tubulin may be functionally relevant in cells and involved in the antiproliferative effects of GO. When cells were synchronized in early S phase and the cell cycle block was released, a lower expression of the mitotic marker p-H3 (S10) was observed in GO-treated cells indicating an inhibition of mitosis.

CML modification in cultured cells and mouse organs often occurred on residues that are known to be modified by other PTM. In endothelial cells, for instance, we found seven CML-modified sites on α-tubulin (K163, K164, K326, K336, K338, K394, K401) and three CML sites on β-tubulin (K58, K324, K379), which overlap with previously described acetylation sites[37]. Moreover, we were able to demonstrate opposite changes of CML modification and acetylation for K58 of β-tubulin. This indicates that exposure of endothelial cells to GO may alter the tubulin code[69,70], which is known to program microtubules for specific functions[70]. In line with this, we also found a decrease in K40 acetylation in cells treated with GO (1 mM, 24 h), although CML modification at this site was not detectable for technical reasons. Since acetylation of K40 has been correlated with increased tubulin stability[38,39], its reduction may contribute to disturbed tubulin functionality. In general, changes in the tubulin code may alter interactions with tubulin-binding proteins, an effect, which may additionally be influenced by the increased abundance of several MAPs as shown in our proteomics analysis.

The described proliferation inhibition phenotype of endothelial cells induced by short-term exposure to 0.5–1 mM GO is likely to be of pathophysiological relevance, since we have also observed inhibition of proliferation when endothelial cells were treated with lower concentrations for longer time (1–100 μM, 14 days). Under these conditions, the reduced proliferation was not related to lower expression of cell cycle regulators, upregulation of CDKN1A/p21 or increased ROS formation but rather to alterations of tubulin as shown by an increased staining intensity of tubulin filaments and, for 100 μM GO, CML modifications on K58 of the tubulin β-4B chain. We speculate that upon exposure to pathophysiological GO levels over a long time, as it is the case in elderly people or diabetic patients, carboxymethylation of tubulin and other proteins may be an initial event, later followed by oxidative stress, stress responses and changes of the proteome. In line with this, it is well known that CML accumulates in tissues in patients with diabetes and/or cardiovascular diseases as well as in aging[8]. The formation of GO and its effect on endothelial cell proliferation may be one of the factors linking metabolic alterations to disturbed angiogenesis in diabetes[71] and aging[72] and may promote pathogenetic processes via this mechanism.

Together, this study uses an enrichment strategy coupled to mass spectrometry to unravel subsets of proteins prone to CML modification, and describes the functional consequences of this modification for proteostasis. By characterizing a proliferation inhibition phenotype in GO-treated endothelial cells, it expands our understanding of vascular dysfunction in aging and age-related metabolic disease and paves the way for further studies to

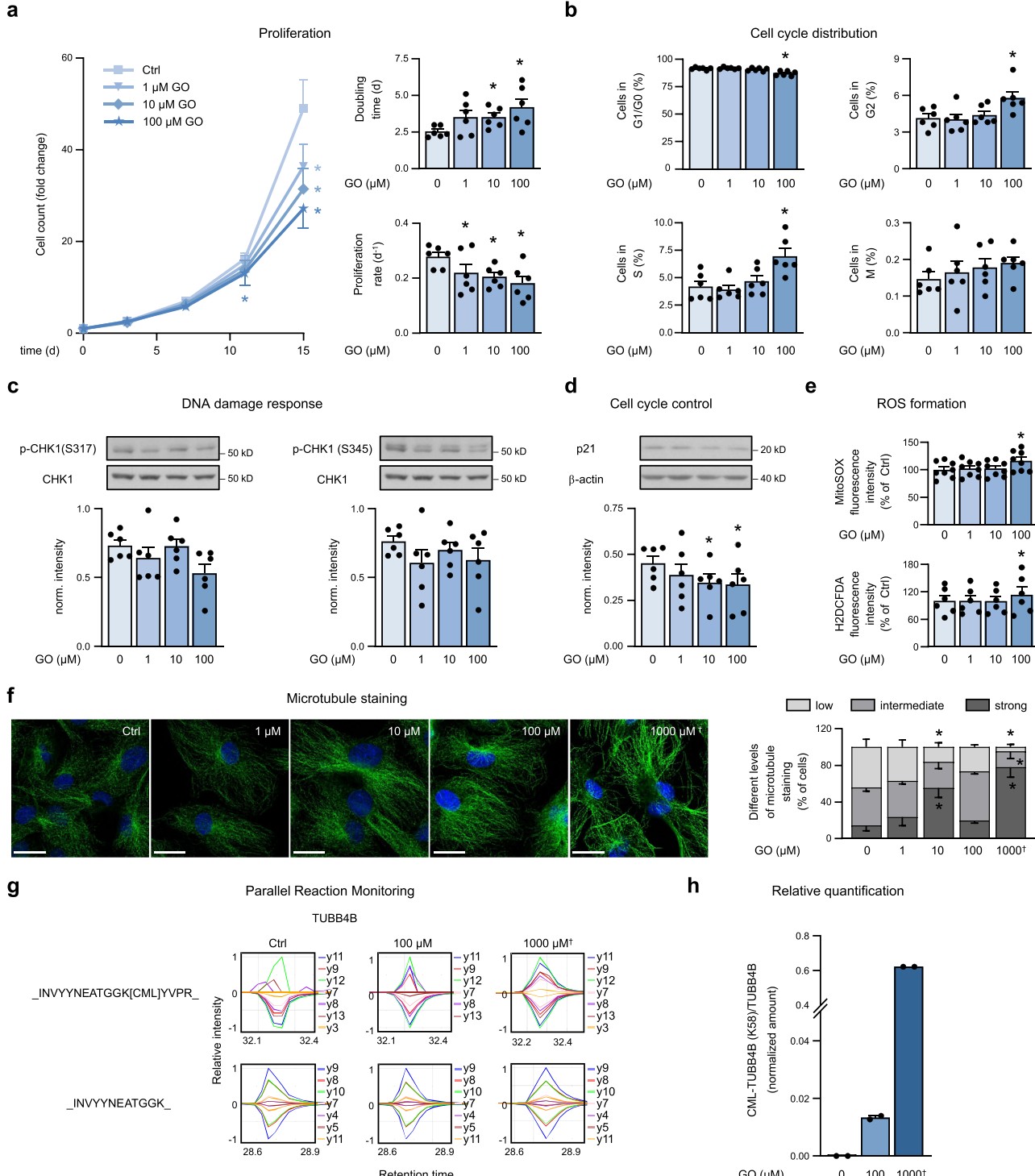

**Fig. 7 Long-term treatment with low doses of glyoxal impairs proliferation and affects tubulin. a–h** HUVEC were treated with 1–100 µM GO for 14 d starting one day after seeding (day 1–15). **a** Left, cell numbers were counted over time. Right, doubling time and proliferation rates were calculated based on cell numbers of days 11 and 15. $n = 8$. **b** HUVEC were subjected to cell cycle analysis by flow cytometry applying a triple staining method (EdU, p-H3 (S10), DAPI). The proportion of cells in the respective cell cycle phases is shown. $n = 6$. **c, d** Cells were subjected to immunoblot analysis. $n = 6$. **e** Intracellular ROS (dichlorodihydrofluorescein diacetate (H2DCFDA)) and mitochondrial ROS (MitoSOX) were measured by flow cytometry. $n = 6$ (H2DCFDA), $n = 8$ (MitoSOX). **f** Cells were stained using antibodies against α/β-tubulin and DAPI. Representative pictures (left) and the percentage of cells showing low, intermediate, or strong staining of microtubules per high-power field (right) are shown. Ctrl: control. Scale bar = 20 µm. $n = 5$. **g** Validation of CML-modified peptides by parallel reaction monitoring (PRM) using heavy spike-in peptides. **h** Relative quantification of _INVYYNEATGGK[CML]YVPR_ (from TUBB4B) in GO-treated HUVEC. Quantification was performed on elution samples after CMLpepIP. $n = 2$. **f–h** A treatment with 1000 µM GO for 48 h was used as the positive control ($\dagger$). **a–f, h**: Bar graphs show mean + SEM. Statistical significance was analyzed using one-way repeated measurement ANOVA corrected via Holm–Šidák method. * $p < 0.05$ vs. control. Source data are provided as a Source Data file. Specific p values are listed in Supplementary Data 6. Related to Supplementary Fig. 8 and Supplementary Data 4.

link CML modifications to functional phenotypes. The presented strategy has limitations of sensitivity as indicated by the low percentage (1–4%) of PSM deriving from CML-modified peptides even in samples from GO-treated cells. It can be envisaged, however, that future development of antibodies with higher specificity for CML-modified peptides will enable the identification of carboxymethylation sites that occur at lower abundance.

## Methods

**Chemicals.** M199 was purchased from Lonza (Verviers, Belgium). Fetal calf serum (FCS), human serum, endothelial growth supplement (ECGS), GO (HUVEC studies), hydroxyurea, nocodazole, antimycin A, thymidine, 2-deoxy-D-glucose, trypsin inhibitor, thrombin, aprotinin, 5-bromo-4-chloro-3-indolyl β-D-galacto-pyranoside (X-Gal), 4′,6-diamidino-2-phenylindole (DAPI), octyl β-D-glucopyr-anoside (IAA), aqueous $NH_3$, ATP, cOmplete™, EDTA-free protease inhibitor cocktail, HEPES, MOPS, NP40, MG132 and Ponceau S were purchased from Sigma (Taufkirchen, Germany). Protease inhibitor mixture complete, EDTA-free was obtained from Roche Diagnostics (Mannheim, Germany), fibrinogen from Merck/Millipore (Darmstadt, Germany) and Fluoromount-G® from Southern Biotech (Birmingham, AL, US), respectively. Bovine serum albumin-C (BSA-C) was from Aurion (Wageningen, The Netherlands) and goat serum from Cell Signaling Technology (Frankfurt, Germany). D-glucose, L-gluta-mine, sodium pyruvate, 5-ethynyl-2′-deoxyuridine, trypsin/EDTA and anhydrous dimethylsulfoxide (DMSO) derived from Thermo Fisher Scientific (Waltham, MA, USA). Oligomycin and carbonyl cyanide-4-(trifluoromethoxy)phenylhydrazone (FCCP) came from Abcam (Cambridge, UK). Carboxyfluorescein succinimidyl ester (CFSE) and 5-(and-6)-chloromethyl-2′,7′-dichlorodihydrofluorescein diace-tate (CM-H2DCFDA) were purchased from Invitrogen (Carlsbad, CA, US). o-Phenylenediamine (OPD) of the highest grade available was obtained from TCI (Tokyo, Japan). Isotopically labeled 2 Hydroxy-3 methyl-2,3 $^{13}C_2$-quinoxaline (pyruvic acid $^{13}C_2$ quinoxaline) was synthesized according to the method of Arun et al.[73], slightly modified by Henning et al.[74]. The identity of the target compound was verified by nuclear magnetic resonance experiments and accurate mass determination. GO (MEF studies), β-mercaptoethanol, glucose, dithiothreitol (DTT), EDTA, formic acid, ammonium bicarbonate, Tris, bovine serum albumin, formic acid, glycerol, potassium chloride, $MgCl_2$, Tween-20 and Triton X-100 were obtained from Carl Roth GmbH (Karlsruhe, Germany). Trifluoroacetic acid, acetonitrile and 2-propanol were from Biosolve BV (Valkenswaard, The Nether-lands) and glycine was from VWR International (Radnor, PA, USA). The pro-teasome substrate Suc-LLVY-AMC was from Ubiquitin-Proteasome Biotechnologies LLC, (Dallas, US).

**Antibodies.** If not otherwise stated, antibodies were used at a dilution of 1:1000. Antibodies raised against β-actin (HUVEC studies, 1:5000), vinculin, cleaved caspase 3 (D175), caspase 3, cleaved PARP (D214), PARP, histone 3 (1:10000), p-CHK1 (S317), p-CHK1 (S345), CHK1, γH2AX (S139), p-p53 (S15), p53, p21, α/β-tubulin (1:2000) and ac-tubulin (K40) were obtained from Cell Signaling Technology (Frankfurt, Germany). The antibody against p16 (1:500) was from BD Pharmingen (Heidelberg, Germany). Antibodies against CML (immunoblot, 1:10000), p-H3 (S10) (immunoblot) and proteasomal subunits 20 S alpha 1-7 were from Abcam (Cambridge, UK). The antibody against p-H3 (S10) (1:200) for flow cytometry was obtained from Merck/Millipore (Darmstadt, Germany) and cyclin A (1:500) antibody was from Santa Cruz Biotechnology (Dallas, TX, US). Antibodies against mono- and K48-, K63-linked mono- and polyubiquitinylated proteins (1:2000) were from Enzo Life Sciences (Farmingdale, NY, USA) and the antibody against β-actin (MEF studies, 1:2000) was from Sigma (Taufkirchen, Germany). Peroxidase-labeled anti-mouse and anti-rabbit IgG (1:5000) were from Kirkegaard and Perry Laboratories, Inc. (Gaithersburg, MD, USA) and anti-mouse IgG (1:1500) and anti-rabbit IgG (1:2000) from Dako GmbH (Hamburg, Ger-many). AlexaFluor®647-conjugated azide (AF647 azide, 0.9 μg/ml) and secondary AlexaFluor®488-conjugated goat anti-rabbit IgG (1:500) were from Thermo Sci-entific (Waltham, MA, USA). The CML antibody (CN1040) used for enrichment was developed using carboxymethylated KLH as immunogen and affinity-purified with CML-agarose. This antibody and the acetyl lysine antibody (agarose con-jugate) were purchased from ImmuneChem Pharmaceuticals Inc. (Burnaby, Canada).

**Mice.** All wild-type mice were C57BL/6J obtained from Janvier Labs (Le Genest-Saint-Isle, France) or from internal breeding at FLI. All animals were kept in a specific pathogen-free animal facility with a 12 h light/dark cycle, at a temperature of 20 °C ± 2 and humidity of 55% ± 15. Young mice were aged 3–4 months, old mice were aged 26–33 months. Mice had unlimited access to food (ssniff, Soest, Germany) during the experiment. For the analysis of total proteome, CMLpepIP and glyoxalase activity measurements only male mice were used. Mice were euthanized with $CO_2$ and organs were isolated, washed in PBS, weighted and immediately snap-frozen in liquid nitrogen before storage in −80 °C. All experi-ments were carried out according to the guidelines from Directive 2010/63/EU of the European Parliament on the protection of animals used for scientific purposes.

The protocols of animal maintenance and euthanasia were approved by the local authorities for animal welfare in the State of Thuringia, Germany.

**MEF cell culture and treatments.** SV40-immortalized mouse embryonic fibro-blasts (MEF, a kind gift of K. L. Rudolph, Leibniz Institute on Aging—Fritz Lip-mann Institute) were grown in high glucose DMEM (Sigma, D6429) until they reached 70% confluency. For GO treatment, DMEM was removed, cells were washed with PBS and live cell imaging medium (140 mM NaCl, 2.5 mM KCl, 1.8 mM CaCl₂, 1.0 mM MgCl₂, 20 mM HEPES, 4500 mg/l glucose, pH 7.4) was added. Subsequently, GO diluted in the same medium was added at final con-centrations of 0.5 or 2 mM for 8–24 h, as notified in figures. For all treatments, cells were snap-frozen in liquid nitrogen and stored at −80 °C until further processing.

**HUVEC cell culture and treatment.** HUVEC were isolated from anonymously acquired human umbilical cords according to the "Ethical principles for Medical Research Involving Human Subjects" (Declaration of Helsinki 1964) as previously described[75]. The protocol was approved by the Jena University Hospital Ethics Committee. The donors were informed and gave written consent. Briefly, after rinsing the cord veins with 0.9% NaCl, endothelial cells were detached with col-lagenase (0.01%, 3 min at 37 °C), suspended in M199/10% FCS, washed once (500 × g, 6 min) and seeded on a cell culture flask coated with 0.2% gelatin. Full growth medium (M199, 17.5% FCS, 2.5% human serum, 7.5 μg/ml ECGS, 7.5 U/ml heparin, 680 μM glutamine, 100 μM vitamin C, 100 U/ml penicillin, 100 μg/ml streptomycin) was added 24 h later. Cells were cultured until confluence and experiments usually carried out with HUVEC from the second passage. Detachment of cells was achieved with trypsin/EDTA after two washes in PBS and stopped with M199/10% FCS. Cells were counted using a Neubauer chamber. Seeding was performed at cell densities between 23,000/cm² and 27,500/cm² dependent on whether experiments were performed 48 or 72 h after seeding, respectively. For senescence and growth experiments, the seeding density was lower (8,000/cm² (senescence-associated β-galactosidase (SA-β-Gal) staining), 10,000/cm² (BrdU incorporation), 15,000/cm² (cell cycle), 20,500/cm² (thymidine block)). If not otherwise indicated, 30 mm-dishes were used for experiments. For mass spectrometry-based proteomics, detachment of HUVEC from 90 mm-dishes was performed with trypsin/EDTA and the reaction was stopped by an equal amount of trypsin inhibitor (1 mg/ml in PBS). Then, cells were washed twice in M199 (500 × g, 3 min) and snap-frozen in liquid nitrogen.

Treatment of HUVEC with GO was performed in experimental medium (i.e., full growth medium without vitamin C) and started 24 h (long-term incubations) or 48–72 h (short-term incubations) after seeding. Short-term incubations were performed with GO concentrations of 0.5 or 1 mM for 4–48 h, as notified in figures. For long-term incubations, 1, 10 or 100 μM GO was applied for 14 days. Within this period, GO was re-added in fresh medium every second day. When cells were passaged (first after two days and then every fourth day), GO was added 4 h after seeding. After GO treatment, cells were stimulated with growth factors if indicated and processed as described below.

**Sample preparation for glyoxal measurements in HUVEC.** HUVEC from 3 × 90 mm-dishes (approximately 7 × 10⁶ cells) were used per condition. Cells were washed twice with PBS (4 ml/dish), detached with trypsin/EDTA (1 ml/dish), added to HEPES buffer (10 mM HEPES (pH 7.4), 145 mM NaCl, 5 mM KCl, 1 mM MgSO₄, 1.5 mM CaCl₂, 10 mM glucose) containing 10% FCS (HEPES/FCS, 5 ml/dish) and pooled with HEPES/FCS rinsing solution (4 ml/dish). After centrifuga-tion of cells (500 × g, 2 min, room temperature), pellets were washed twice with PBS and snap-frozen in liquid nitrogen. For lysis, cells were resuspended in 200 μl ultrapure water, subjected to 3 freezing/thawing cycles and sonicated. Upon cen-trifugation (700 × g, 6 min, 4 °C), aliquots of the supernatants were taken for protein determination according to Lowry. 200 μl of each sample containing 1700 μg of protein were then combined with 200 μl of the reaction mixture (40% sodium formate buffer (2 M sodium formate in 20 mM EDTA, pH 4.0), 20% standard (1.25 μM pyruvic acid-¹³C₂-quinoxaline) and 40% OPD (2.75 mM o-phenylenediamine in 0.1% HCl). After an incubation on a rotator (24 h, 20 rpm, room temperature, in the dark), 100 μl of 2 N trifluoroacetic acid was added for protein precipitation. Following another incubation on a rotator (1 h, 20 rpm, room temperature, in the dark), 166 μl of 4 N ammonium hydroxide and 34 μl ultrapure water were added and samples were centrifuged (16,000 × g, 10 min, 4 °C). Supernatants were frozen until GO analysis was performed.

**Glyoxal determination by HPLC-MS/MS.** The HPLC apparatus (Jasco, Groß-Umstadt, Germany) consisted of a pump (PU-2080 Plus) with degasser (LG-2080-02) and quaternary gradient mixer (LG-2080-04), a column oven (Jasco Jetstream II) and an autosampler (AS-2057 Plus). Mass spectrometric detection was con-ducted on an API 4000 QTrap LC-MS/MS system (Sciex, Concord, ON, Canada) equipped with a turbo ion spray source using electrospray ionization (ESI) in positive mode. The chromatographic method and the source parameters as well as the quinoxaline-specific orifice and collision energies for mass spectrometric detection were published elsewhere[74]. Quantitation was performed using the standard addition method. More precisely, increasing concentrations of authentic reference compounds at factors of 0.5, 1, 2 and 3 times the concentration of the

analyte in the sample were added to separate aliquots of the sample after workup procedure. The aliquots were analyzed and a regression of response *versus* concentration was used to determine the concentration of the analyte in the sample. Calibration with this method resolves potential matrix interferences. All HUVEC samples from different experiments were analyzed in a single batch to exclude inter-assay variations. Potential losses during workup procedure and intra-batch changes of instrument sensitivity were corrected with pyruvic acid $^{13}C_2$-quinoxaline as internal standard.

**Sample preparation for quantitation of total CML levels.** MEF and mouse organs were lysed and proteins acetone precipitated as described in *Sample preparation for mass spectrometry-based proteomics*. HUVEC were detached by trypsin/EDTA, transferred to HEPES/FCS, centrifuged ($500 \times g$, 6 min), washed twice in 2 ml PBS and subjected to one freezing/thawing cycle in liquid nitrogen. Pellets were resuspended in 200 µl ice-cold Tris buffer (50 mM Tris (pH 7.4), 2 mM EDTA, 1 mM EGTA, 50 mM NaF, 150 mM NaCl, 10 mM $Na_4P_2O_7$, 1 mM $Na_3VO_4$, 1% Triton X-100, 0.1% SDS, 0.5% sodium deoxycholate, 1 mM PMSF, 10 µl/ml protease inhibitor cocktail) and incubated on ice for 30 min. After homogenization using a tissue homogenizer, proteins were precipitated by adding TCA to a final concentration of 10%. Samples were centrifuged ($4000 \times g$, 5 min, 4 °C), pellets were washed twice in ice-cold 80% acetone and stored at −80 °C until further processing.

**Quantitation of total CML levels by HPLC-MS/MS.** Protein pellets were reconstituted in PBS and 250 µl aliquots of protein extracts (1 mg/ml) were reduced by addition of 100 µl $NaBH_4$ solution (15 mg/ml in 0.01 M NaOH) and were shaken for 1 h at room temperature. Samples were dried in a vacuum concentrator (Savant-Speed-Vac Plus SC 110A combined with a Vapor Trap RVT 400, Thermo Fisher Scientific, Bremen, Germany). 800 µl of 6 M HCl was added and the solution was heated 20 h at 110 °C under an argon atmosphere. Volatiles were removed in a vacuum concentrator and the residue was dissolved in 300 µl of ultrapure water. Samples were filtered through 0.45 µm cellulose acetate Costar SpinX filters (Corning Inc., Corning, USA). After complete hydrolysis, the amount of amino acids in hydrolysates was determined by ninhydrin assay and referenced to a calibration of L-leucine concentrated between 5 and 100 µM[30]. The absorbance was determined at 546 nm with an Infinite M200 microplate reader (Tecan, Männedorf, Switzerland) using 96-well plates.

Chromatographic separations were performed on a stainless-steel column (XSelect HSS T3, 250×3.0 mm, RP18, 5 µm, Waters, Milford, USA) using a flow rate of 0.7 ml/min and a column temperature of 25 °C. Eluents were ultrapure water (A) and a mixture of methanol (Biosolve, 0013684102BS) and ultrapure water (7:3, (v/v); B), both supplemented with 1.2 ml/l heptafluorobutyric acid. Samples were injected (10 µl) at 2% B and run isocratic for 2 min, gradient was changed to 14% B within 10 min (held for 0 min), 87% B within 22 min (held for 0 min), 100% B within 0.5 min (held for 7 min) and 2% B within 2.5 min (held 8 min).

A PU-2080 Plus quaternary gradient pump with degasser and an AS-2057 Plus autosampler (Jasco, Gross-Umstadt, Germany) were used. The mass analyses were performed using an API 4000 quadrupole instrument (Applied Biosystems, Foster City, USA) equipped with an API source using electrospray ionization. The HPLC system was connected directly to the probe of the mass spectrometer. Nitrogen was used as sheath and auxiliary gas. To measure CML, the scheduled multiple-reaction monitoring (sMRM) mode of HPLC-MS/MS was used. Quantitation was based on the standard addition method using known amounts of pure CML standard to compensate for matrix effects. The authentic reference compound was added at 0.5, 1, 2, and 4 times the concentration of the analyte in the sample and correlation coefficients were 0.9 or higher.

**Thermal proteome profiling (TPP).** MEF were grown in 150 mm dishes until they reached 70–80% confluence. They were treated at different concentration of GO (0, 0.5 mM and 2 mM) for 8 h in imaging medium (as described in "*MEF cell culture and treatments*"). Cells were harvested using trypsin/EDTA phenol red (0.05%) and counted in order to have $10^6$ cells per condition in PBS. Thereupon, cell suspensions were split into 10 individual 0.2 ml PCR tubes in equal volumes (100 µl each tube) and quickly span down to reach a final volume of 20 µl of cell suspension in each tube. Tubes were heated at different temperatures for 3 min using a 96 well dry bath ThermoQ (Hangzhou Bioer Technology, Hangzhou, China). Temperatures were: 37, 41, 44, 47, 50, 53, 56, 60, 63, and 67 °C. Cells were then incubated at 25 °C for 3 min, before adding 80 µl of 0.625% NP40 and being snap-frozen in liquid nitrogen. Cells were lysed by a thaw-freeze-thaw cycle consisting of an incubation step for 5 min at 25 °C, followed by snap freezing and an additional incubation at 25 °C for 5 min. Lysates were transferred to 7 × 20 mm polycarbonate thick-wall tubes (Beckman Coulter, Krefeld, Germany) for ultra-centrifugation at 100,000 × g for 20 min at 4 °C (centrifuge Optima TLX with rotor TLA 100, Beckman Coulter). Identical volumes of the resulting supernatants (32 µl) estimated to correspond to approximately 30 µg of protein extract were then taken for the 37 °C sample. Samples were reduced with 10 mM DTT for 15 min at 45 °C and cysteine alkylated with freshly prepared 15 mM IAA for 30 min at 25 °C, in the dark. Following reduction and alkylation, proteins were precipitated with ice-cold

acetone and digested into peptides, as described in "Sample preparation for mass spectrometry-based proteomics".

**Sample preparation for mass spectrometry-based proteomics.** HUVEC were treated for 14 days (1–100 µM) or 48 h (1 mM) with GO. HUVEC and MEF cell pellets ($3 \times 10^6$ and $1 \times 10^6$, respectively) were thawed, reconstituted in 150 and 50 µl of ice-cold PBS and lysed by addition of 150 and 50 µl of 2x lysis buffer (100 mM HEPES pH 8.0, 20 mM DTT, 2% SDS), respectively. Mouse organs were thawed and transferred into Precellys® lysing kit tubes (Keramik-kit 1.4/2.8 mm, 2 ml (CKM)) containing 1 ml of PBS supplemented with 1 tab of cOmplete™, Mini, EDTA-free Protease Inhibitor per 50 ml. For homogenization, tissues were shaken twice at 6000 rpm for 30 s using Precellys® 24 Dual (Bertin Instruments, Montigny-le-Bretonneux, France) and the homogenate was transferred to new 2 ml Eppendorf tubes. Based on estimated protein content (5% of fresh tissue weight for liver and 8% for heart and kidney), 100 µg of protein was processed for further analyses. Volumes were adjusted using PBS and one volume equivalent of 2× lysis buffer was added.

Samples were sonicated in a Bioruptor Plus (Diagenode, Seraing, Belgium) for 10 cycles with 1 min ON and 30 s OFF with high intensity at 20 °C. Samples were quickly centrifuged and a second sonication cycle was performed as described above. The lysates were centrifuged at $18,407 \times g$ for 1 min and transferred to new 1.5 ml Eppendorf tubes. Subsequently, samples were reduced using 10 mM DTT for 30 min at room temperature and alkylated using freshly made 15 mM IAA for 30 min at room temperature in the dark. Subsequently, proteins were acetone precipitated and digested using LysC (Wako sequencing grade) and trypsin (Promega sequencing grade), as described by Buczak et al.[76] The digested proteins were then acidified with 10% (v/v) trifluoroacetic acid and desalted using Waters Oasis® HLB µElution Plate 30 µm following manufacturer instructions. The eluates were dried down using a vacuum concentrator and reconstituted in 5% (v/v) acetonitrile, 0.1% (v/v) formic acid. For Data-Independent Acquisition (DIA) based analysis, samples were transferred to an MS vial, diluted to a concentration of 1 µg/µl, and spiked with iRT kit peptides (Biognosys, Zurich, Switzerland) prior to analysis by LC-MS/MS. For Tandem Mass Tags (TMT) based analysis, samples were further processed for TMT labeling as described below.

**TMT labeling and high pH peptide fractionation for organ aging proteome and TPP.** Reconstituted peptides (at 1 µg/µl) were buffered using 100 mM HEPES buffer pH 8.5 (1:1 ratio) for labeling. 10–20 µg peptides were taken for each labeling reaction. TMT-10plex reagents (Thermo Scientific, Waltham, MA, USA) were reconstituted in 41 µl 100% anhydrous DMSO. TMT labeling was performed by addition of 1.5 µl of the TMT reagent. After 30 min of incubation at room temperature with shaking at 600 rpm in a thermomixer (Eppendorf, Hamburg, Germany), a second portion of TMT reagent (1.5 µl) was added and incubated for another 30 min. After checking labeling efficiency, samples were pooled (45–50 µg total), desalted with Oasis® HLB µElution Plate and subjected to high pH fractionation prior to MS analysis.

Offline high pH reverse phase fractionation was performed using a Waters XBridge C18 column (3.5 µm, 100 × 1.0 mm, Waters) with a Gemini C18, 4 × 2.0 mm SecurityGuard (Phenomenex) cartridge as a guard column on an Agilent 1260 Infinity HPLC, as described in[76]. Forty-eight fractions were collected along with the LC separation, which were subsequently pooled into 16 fractions (for liver and kidney) and 24 fractions (heart). Pooled fractions were dried in a vacuum concentrator and then stored at −80 °C until LC-MS/MS analysis.

**LC-MS/MS based on data-independent acquisition (DIA) for MEF and HUVEC.** Peptides (approximately 1 µg) were separated using a nanoAcquity UPLC M-Class system (Waters Milford, USA) with a trapping (nanoAcquity Symmetry C18, 5 µm, 180 µm × 20 mm) and an analytical column (nanoAcquity BEH C18, 1.7 µm, 75 µm × 250 mm). The outlet of the analytical column was coupled directly to a Q-exactive HF-X or Q-exactive HF (Thermo Fisher, Waltham, MA, USA) using the Proxeon nanospray source. Solvent A was water, 0.1% formic acid and solvent B was acetonitrile, 0.1% formic acid. The samples (approx. 1 µg) were loaded onto the trapping column with a constant flow of solvent A at 5 µl/min. Trapping time was 6 min. Peptides were eluted via the analytical column with a constant flow of 0.3 µl/min. During the elution step, the percentage of solvent B increased in a nonlinear fashion from 0% to 40% in 90 min. Total runtime was 115 min, including clean-up and column re-equilibration. The peptides were introduced into the MS via a Pico-Tip Emitter 360 µm OD × 20 µm ID; 10 µm tip and a spray voltage of 2.2 kV was applied. The capillary temperature was set at 300 °C. The radio frequency (RF) ion funnel was set to 40%.

Data from a subset of conditions were first acquired in data-dependent acquisition (DDA) mode to contribute to a sample specific spectral library. The conditions were as follows: Full scan MS spectra with mass range 350–1650 $m/z$ were acquired in profile mode in the Orbitrap with resolution of 60,000 FWHM. The filling time was set at maximum of 20 ms with limitation of $3 \times 10^6$ ions. The "Top N" method was employed to take the 15 most intense precursor ions (with an intensity threshold of $4 \times 10^4$) from the full scan MS for fragmentation (using HCD normalized collision energy, 27%) and quadrupole isolation (1.6 Da window) and measurement in the Orbitrap (resolution 15,000 FWHM, fixed first mass 120 m/z).

The peptide match 'preferred' option was selected and the fragmentation was performed after accumulation of $2 \times 10^5$ ions or after filling time of 25 ms for each precursor ion (whichever occurred first). MS/MS data were acquired in profile mode. Only multiply charged (2+–5+) precursor ions were selected for MS/MS. Dynamic exclusion was employed with maximum retention period of 20 s and relative mass window of 10 ppm. Isotopes were excluded. In order to improve the mass accuracy, internal lock mass correction using a background ion (m/z 445.12003) was applied.

For DIA, the same gradient conditions were applied to the LC as for the DDA and the MS conditions were varied as described: Full scan MS spectra with mass range 350–1650 m/z were acquired in profile mode in the Orbitrap with resolution of 120,000 FWHM. The default charge state was set to 3+. The filling time was set at maximum of 60 ms with limitation of $3 \times 10^6$ ions. DIA scans were acquired with 34 mass window segments of differing widths across the MS1 mass range. HCD fragmentation (stepped normalized collision energy; 25.5, 27, 30%) was applied and MS/MS spectra were acquired with a resolution of 30,000 FWHM with a fixed first mass of 200 m/z after accumulation of $3 \times 10^6$ ions or after filling time of 40 ms (whichever occurred first). Data were acquired in profile mode. For data acquisition and processing of the raw data Xcalibur 4.0 and Tune version 2.9 (both Thermo Fisher) were employed.

**Data processing for DIA**. DpD (DDA plus DIA) libraries were then created by searching both the DDA runs and the DIA runs using Spectronaut Pulsar (v10/11, Biognosys, Zurich, Switzerland). The data were searched against species-specific protein databases (Uniprot *Mus musculus* or *Homo sapiens*, reviewed entry only, release 2016_01) with a list of common contaminants appended. The data were searched with the following modifications: carbamidomethyl (C) as fixed modification, and oxidation (M), acetyl (protein N-term) and carboxymethyllysine (CML) as variable modifications. A maximum of 3 missed cleavages was allowed. The library search was set to 1% false discovery rate (FDR) at both protein and peptide levels. This library contained 92,917 precursors, corresponding to 5,393 protein groups for HUVEC, and 83,207 precursors, corresponding to 4,725 protein groups for MEF using Spectronaut protein inference. DIA data were then uploaded and searched against this spectral library using Spectronaut Professional (v.11 or v.15) and default settings. Relative quantification was performed in Spectronaut for each pairwise comparison using a two-sided t-test performed at the precursor level followed by multiple testing correction and default settings, except: Major Group Quantity = Sum peptide quantity; Major Group Top N = OFF; Minor Group Quantity = Sum precursor quantity; Minor Group Top N = OFF; Data Filtering = Q value sparse; Normalization Strategy = Local normalization; Row Selection = Q value complete. The data (candidate tables) and protein quantity data reports were then exported and further data analyses and visualization were performed with R (v.3.6.3) and R studio server (v. 1.2.5042) using in-house pipelines and scripts.

**LC-MS/MS based on tandem mass tags (TMT) for organ aging proteome and TPP**. For TMT experiments, fractions were resuspended in 10 µl reconstitution buffer (5% (v/v) acetonitrile, 0.1% (v/v) trifluoroacetic acid in water) and 3 µl were injected. Peptides were analyzed as described in[76] using a nanoAcquity UPLC system (Waters) fitted with a trapping (nanoAcquity Symmetry C18, 5 µm, 180 µm × 20 mm) and an analytical column (nanoAcquity BEH C18, 2.5 µm, 75 µm × 250 mm), and coupled to an Orbitrap Fusion Lumos (Thermo Fisher Scientific, Waltham, MA, USA). Briefly, peptides were separated using a 130 min nonlinear gradient. Full scan MS spectra with mass range 375–1500 m/z were acquired in profile mode in the Orbitrap with resolution of 60,000 FWHM using the quad isolation. The RF on the ion funnel was set to 40%. The filling time was set at maximum of 100 ms with an AGC target of $4 \times 10^5$ ions and 1 microscan. The peptide monoisotopic precursor selection was enabled along with relaxed restrictions if too few precursors were found. The most intense ions (instrument operated for a 3 s cycle time) from the full scan MS were selected for MS2, using quadrupole isolation and a window of 1 Da. HCD was performed with collision energy of 35%. A maximum fill time of 50 ms for each precursor ion was set. MS2 data were acquired with fixed first mass of 120 m/z. The dynamic exclusion list was with a maximum retention period of 60 s and relative mass window of 10 ppm. The instrument was not set to inject ions for all available parallelizable time. For the MS3, the precursor selection window was set to the range 400–2000 m/z, with an exclude width of 18 m/z (high) and 5 m/z (low). The most intense fragments from the MS2 experiment were co-isolated (using Synchronus Precursor Selection=8) and fragmented using HCD (65%). MS3 spectra were acquired in the Orbitrap over the mass range 100–1000 m/z and resolution set to 30000. The maximum injection time was set to 105 ms and the instrument was set not to inject ions for all available parallelizable time.

**Data processing for TMT**. TMT-10plex data from aging mouse organs were processed using Proteome Discoverer v2.0 (Thermo Fisher Scientific, Waltham, MA, USA). Data were searched against the fasta database (Uniprot *Mus musculus* database, reviewed entry only, release 2016_01) using Mascot v2.5.1 (Matrix Science) with the following settings: Enzyme was set to trypsin, with up to 1 missed cleavage. MS1 mass tolerance was set to 10 ppm and MS2 to 0.5 Da. Carbamidomethyl cysteine was set as a fixed modification and oxidation of methionine as

variable. Other modifications included the TMT-10plex modification from the quantification method used. The quantification method was set for reporter ions quantification with HCD and MS3 (mass tolerance, 20 ppm). The false discovery rate for PSMs was set to 0.01 using Percolator[77].

Reporter ion intensity values for the PSMs were exported and processed with procedures written in R (v.3.6.3) and R studio server (v. 1.2.5042), as described in[78]. Briefly, PSMs mapping to reverse or contaminant hits, or having a Mascot score below 15, or having reporter ion intensities below $1 \times 10^3$ in all the relevant TMT channels were discarded. TMT channels intensities from the retained PSMs were then $\log_2$ transformed, normalized and summarized into protein group quantities by taking the median value using MSnbase[79]. At least two unique peptides per protein were required for the identification and only those peptides with no missing values across all 10 channels were considered for quantification. Protein differential expression was evaluated using the limma package[80]. Differences in protein abundances were statistically determined using an unpaired, two-sided Student's t test moderated by the empirical Bayes method. P values were adjusted for multiple testing using the Benjamini–Hochberg method (FDR, denoted as "adj. p")[81].

**Data processing for TPP**. For the analysis of the TPP sample data (TMT10plex, high pH fractionated, MS3 data acquisition), raw data files were first processed through the preMascot process of the isobarquant package[34]. After the .mgf files had been generated, these were processed via Mascot Daemon (Matrix Science) using Mascot version 2.5.1. Firstly, species-specific database including decoy was created by concatenating the forward entries (Uniprot *Mus musculus* database, reviewed entry only, release 2016_01) to the reversed sequences of the database. Data were then searched against the relevant database with the following settings: Enzyme = trypsin, 3 missed cleavages allowed. Modifications: carbamidomethyl (C) and TMT10plex (K) as fixed modifications; oxidation (M), carboxymethyllysine (K) and TMT10plex (N-term) as variable. MS1 tolerance was 10 ppm, and 0.5 Da for MS2. When the .dat files for each fraction had been generated, these were subjected to the postMascot process of isobarquant, combining all the outputs into a single, merged, TMT10plex quantified protein output for further processing of melting point curves using the TPP package in R[82].

**Enrichment of CML-modified peptides (CMLpepIP)**. Lysates containing approximately 1 mg protein for HUVEC and MEF and 5 mg for tissues were digested as described in "Sample preparation for mass spectrometry-based proteomics" with minor modifications: the digestion buffer used was 3 M Urea, 100 mM HEPES, 5% (v/v) acetonitrile and the ratio of LysC and trypsin used was 1:150 enzyme to protein. The digests were acidified with 10% (v/v) trifluoroacetic acid and then desalted with Waters Oasis® HLB 96-well Plate 30 µm (Waters Corp., Milford, MA, USA). In this process, the columns were conditioned with $2 \times 1000$ µl solvent B (80% (v/v) acetonitrile; 0.05% (v/v) formic acid) and equilibrated with $2 \times 1000$ µl solvent A (0.05% (v/v) formic acid in milliQ water). The samples were loaded, washed 2 times with 1000 µl solvent A, and then eluted with 500 µl solvent B. The eluates were dried down and dissolved in 200 µl of IP buffer (50 mM MOPS, pH 7.3, 10 mM KPO₄ pH 7.5, 50 mM HEPES, 2.5 mM octyl β-D-glucopyranoside) at concentration of 5 µg/µl followed by sonication in a Bioruptor Plus (5 cycles with 1 min ON and 30 s OFF with high intensity at 20 °C). Ten percent of the sample was kept and analyzed separately as input control. A pre-cleaning step was applied by incubating each sample with 10 µl of Protein A magnetic beads (New England Biolabs GmbH, Frankfurt, Germany) for 1 h at 4 °C in tube roller (15 rpm) (STARLAB Tube roller Mixer RM Multi-1). Samples were transferred to a magnetic rack (DynaMag™-2, Invitrogen). The supernatant was transferred into a new 1.5 ml Eppendorf tube and incubated overnight with 30 µg of pan (ε-N) CML antibody (250 µg/ml) at 4 °C on tube roller (15 rpm) (STARLAB Tube roller Mixer RM Multi-1). Subsequently, 150 µl of Protein A magnetic beads were added to the samples and incubated for 1 h at 4 °C on tube roller as previously. Samples were then transferred to a magnetic rack and the flow through was collected in a fresh tube, and beads were washed 3 times in 300 µl IP buffer. The enriched peptides were then eluted 3 times in 54 µl of 0.1 M glycine pH 2.6. The fraction of elution, flow through and the input were desalted using Macro Spin Column C18 columns (Harvard Apparatus, Cambridge, MA, USA) following manufacturer instructions. The eluates were then dried down in a vacuum concentrator, dissolved in 5% (v/v) acetonitrile, 0.1% (v/v) formic acid and directly analyzed by LC-MS/MS for the MEF and HUVEC. An additional step of high pH peptide fractionation was performed for eluates from mouse organs, as described in "TMT labeling and high pH peptide fractionation for organ aging proteome and TPP". Eight pooled fractions from high pH fractionation were measured by LC-MS/MS for each sample.

**LC-MS/MS based on data-dependent acquisition (DDA) for CMLpepIP**. Peptides were separated using the nanoAcquity UPLC system (Waters) fitted with a trapping (nanoAcquity Symmetry C18, 5 µm, 180 µm × 20 mm) and an analytical column (nanoAcquity BEH C18, 1.7 µm, 75 µm × 250 mm). The outlet of the analytical column was coupled directly to an Orbitrap Fusion Lumos (Thermo Fisher Scientific, Waltham, MA, USA) using the Proxeon nanospray source. Solvent A was water, 0.1% (v/v) formic acid and solvent B was acetonitrile, 0.1% (v/v) formic acid.

The samples (500 ng) were loaded with a constant flow of solvent A at 5 µl/min onto the trapping column. Trapping time was 6 min. Peptides were eluted via the analytical column with a constant flow of 0.3 µl/min. During the elution step, the percentage of solvent B increased in a linear fashion from 3% to 25% in 30 min, then increased to 32% in 5 more minutes and finally to 50% in a further 0.1 min. Total runtime was 60 min. The peptides were introduced into the mass spectrometer via a Pico-Tip Emitter 360 µm OD × 20 µm ID; 10 µm tip (New Objective) and a spray voltage of 2.2 kV was applied. The capillary temperature was set at 300 °C. The RF lens was set to 30%. Full scan MS spectra with mass range 375-1500 m/z were acquired in profile mode in the Orbitrap with resolution of 120,000 FWHM. The filling time was set at maximum of 50 ms with limitation of $2 \times 10^5$ ions. The "Top Speed" method was employed to take the maximum number of precursor ions (with an intensity threshold of $5 \times 10^3$) from the full scan MS for fragmentation (using HCD collision energy, 30%) and quadrupole isolation (1.4 Da window) and measurement in the ion trap, with a cycle time of 3 s. The MIPS (monoisotopic precursor selection) peptide algorithm was employed but with relaxed restrictions when too few precursors meeting the criteria were found. The fragmentation was performed after accumulation of $2 \times 10^3$ ions or after filling time of 300 ms for each precursor ion (whichever occurred first). MS/MS data were acquired in centroid mode, with the rapid scan rate and a fixed first mass of 120 m/z. Only multiply charged (2+–7+) precursor ions were selected for MS/MS. Dynamic exclusion was employed with maximum retention period of 60 s and relative mass window of 10 ppm. Isotopes were excluded. Additionally, only 1 data-dependent scan was performed per precursor (only the most intense charge state selected). Ions were injected for all available parallelizable time. In order to improve the mass accuracy, a lock mass correction using a background ion (m/z 445.12003) was applied. For data acquisition and processing of the raw data, Xcalibur 4.0 (Thermo Scientific, Waltham, MS, USA) and Tune version 2.1 were employed.

### Data processing for DDA for CMLpepIP.

Spectromine v.2 (Biognosys AG, Schlieren, Switzerland) was used to search raw data against species-specific databases (Uniprot *Mus musculus* or *Homo sapiens*, reviewed entry only, release 2016_01) with a list of common contaminants appended. The data were searched with the following modifications: carbamidomethyl (C) as fixed modification, and oxidation (M), acetyl (protein N-term), carboxymethyllysine (CML) and carboxymethyl (protein N-term) as variable modifications (Supplementary Data 1). The carboxymethyl modification mass shift used was 58.0054793084 Da ($C_2H_2O_2$) on not C-terminal lysines (CML) or at protein N-termini on any residue.

The mass error tolerance was set as dynamic, described as follow: SpectroMine performs two calibration searches: based on the first-pass calibration (rough calibration), the ideal tolerance for the second-pass calibration is defined; based on the second-pass calibration (finer calibration), the ideal tolerance for the main search is defined. A maximum of three missed cleavages were allowed. Peptide and protein level 1% FDR were applied using a target-decoy strategy. A PTM localization filter (Min. localization threshold = 0.75) was activated for variable modifications (PTM localization algorithm based on Sharma et al.[83]). Spectromine outputs were then used to perform downstream analyses.

### Enrichment of acetylated peptides.

HUVEC pellets (3 biological replicates per condition/time point) corresponding to approximately 500 µg of proteins were used. Cell pellets were thawed, lysed and digested as described in the paragraph "Sample preparation for mass spectrometry-based proteomics". The digested peptides were acidified with 10% (v/v) trifluoroacetic acid and then desalted with Waters Oasis® HLB 96-well Plate 30 µm following manufacturer instructions. The eluates were dried down and dissolved in 500 µl of IP buffer (50 mM MOPS pH 7.3, 10 mM $KPO_4$ pH 7.5, 50 mM NaCl, 2.5 mM Octyl β-D-glucopyranoside) to reach a peptide concentration of 1 µg/µl, followed by sonication in a Bioruptor (5 cycles with 1 min ON and 30 s OFF with high intensity at 20 °C). 10% of the sample was kept to use it as input. Agarose beads coupled to antibody against acetyl lysine were washed three times with washing buffer (20 mM MOPS pH 7.4, 10 mM $KPO_4$ pH 7.5, 50 mM NaCl) before incubation with each sample-peptide for 1.5 h on a rotating well at 750 rpm (STARLAB Tube roller Mixer RM Multi-1). Samples were transferred into Clearspin filter microtubes (0.22 µm) (Dominique Dutscher SAS, Brumath, France) and centrifuged at 4 °C for 1 min at 2000 × g. Beads were washed first with IP buffer (three times), then with washing buffer (three times) and finally with 5 mM ammonium bicarbonate (three times). Thereupon, the enriched peptides were eluted first in basic condition using 50 mM aqueous $NH_3$, then using 0.1% (v/v) trifluoroacetic acid in 10% (v/v) 2-propanol and finally with 0.1% (v/v) trifluoroacetic acid. Elutions were dried down and reconstituted in MS buffer A (5% (v/v) acetonitrile, 0.1% (v/v) formic acid), acidified with 10% (v/v) trifluoroacetic acid and then desalted with *Waters Oasis® HLB µElution Plate 30 µm*. Desalted peptides were finally dissolved in MS buffer A and analyzed by LC-MS/MS.

### LC-MS/MS based on data-independent acquisition (DIA) for acetylated peptides.

Peptides were separated using the nanoAcquity UPLC system (Waters) fitted with a trapping (nanoAcquity Symmetry C18, 5 µm, 180 µm × 20 mm) and an analytical column (nanoAcquity BEH C18, 1.7 µm, 75 µm × 250 mm). The outlet of the analytical column was coupled directly to an Orbitrap Fusion Lumos (Thermo Fisher Scientific, Waltham, MA. USA) using the Proxeon nanospray source. Solvent A was water, 0.1% (v/v) formic acid and solvent B was acetonitrile, 0.1% (v/v) formic acid. The samples were loaded with a constant flow of solvent A at 5 µl/min onto the trapping column. Trapping time was 6 min. Peptides were eluted via the analytical column with a constant flow of 0.3 µl/min. During the elution step, the percentage of solvent B increased in a nonlinear fashion from 0% to 40% in 40 min. Total runtime was 60 min, including clean-up and column re-equilibration. The peptides were introduced into the mass spectrometer via a Pico-Tip Emitter 360 µm OD × 20 µm ID; 10 µm tip (New Objective) and a spray voltage of 2.2 kV was applied. The capillary temperature was set at 300 °C. The RF lens was set to 30%. Data from each sample were first acquired in DDA mode. The conditions were as follows: Full scan MS spectra with mass range 350-1650 m/z were acquired in profile mode in the Orbitrap with resolution of 60,000 FWHM. The filling time was set at maximum of 50 ms with limitation of $2 \times 10^5$ ions. The "Top Speed" method was employed to take the maximum number of precursor ions (with an intensity threshold of $5 \times 10^4$) from the full scan MS for fragmentation (using HCD collision energy, 30%) and quadrupole isolation (1.4 Da window) and measurement in the Orbitrap (resolution 15,000 FWHM, fixed first mass 120 m/z), with a cycle time of 3 s. The MIPS (monoisotopic precursor selection) peptide algorithm was employed but with relaxed restrictions when too few precursors meeting the criteria were found. The fragmentation was performed after accumulation of $2 \times 10^5$ ions or after filling time of 22 ms for each precursor ion (whichever occurred first). MS/MS data were acquired in centroid mode. Only multiply charged (2+–7+) precursor ions were selected for MS/MS. Dynamic exclusion was employed with maximum retention period of 15 s and relative mass window of 10 ppm. Isotopes were excluded. In order to improve the mass accuracy, internal lock mass correction using a background ion (m/z 445.12003) was applied. For data acquisition and processing of the raw data Xcalibur 4.0 (Thermo Scientific, Waltham, MA, USA)) and Tune version 2.1 were employed.

For the DIA data acquisition, the same gradient conditions were applied to the LC as for the DDA and the MS conditions were varied as described: Full scan MS spectra with mass range 350–1650 m/z were acquired in profile mode in the Orbitrap with resolution of 120,000 FWHM. The filling time was set at maximum of 20 ms with limitation of $5 \times 10^5$ ions. DIA scans were acquired with 30 mass window segments of differing widths across the MS1 mass range with a cycle time of 3 s. HCD fragmentation (30% collision energy) was applied and MS/MS spectra were acquired in the Orbitrap with a resolution of 30,000 FWHM over the mass range 200–2000 m/z after accumulation of $2 \times 10^5$ ions or after filling time of 70 ms (whichever occurred first). Ions were injected for all available parallelizable time. Data were acquired in profile mode.

### Data processing for DIA for acetylated peptides.

DpD (DDA plus DIA) libraries were created by searching both DDA and DIA runs using Spectronaut Pulsar (v.13), as described in "Data processing for DIA" with the following modification: acetyl (K) was included as variable modification. The library contained 17,051 precursors, corresponding to 3,227 protein groups using Spectronaut protein inference. DIA data were then uploaded and searched against this spectral library using Spectronaut Professional (v.13) and default settings. Intensities of precursors deriving from acetylated peptide were obtained from the peptide report table and filtered for a localization score >= 0.75 and further processed using in house written scripts in R (v.3.6.3) and R studio server (v. 1.2.5042). Intensities were summarized at the level of acetylation site by summing the intensities of all the precursors containing a given acetylation sites. Acetylation site intensities were normalized across runs by $\log_2$ transformation and median centering.

### Automated high pH peptide fractionation.

Peptides from the protein digestion of HUVEC treated with GO as described in the paragraph "Sample preparation for mass spectrometry-based proteomics" were reconstituted (at 0.5 µg/µl) using 5% (v/v) acetonitrile, 0.1% (v/v) trifluoroacetic acid in water. The fractionation was performed using Agilent Bravo Automated Liquid Handling Platform (Agilent Technologies, Santa Clara, USA) equipped with pipetting head 96 AM Assay Map (Agilent Technologies, Santa Clara, USA). All the buffers and the samples were transferred before starting the protocol into individual U-bottom 96 well plates (Greiner Bio-One International GmbH, Frickenhausen, Germany). The cartridge used for the protocol were RP-S cartridge (Agilent Technologies, Santa Clara, USA). The fractionation was performed using the Fractionation v1.0 protocol suggested by the manufacturer and described in Kuras et al.[84] with the following changes. Only about 10 µg of peptides per sample were used for the fractionation. The eluates were dried down using a vacuum concentrator, and reconstituted samples in 5% (v/v) acetonitrile, 0.1% (v/v) formic acid and analyzed by PRM as described in the paragraph "Parallel Reaction Monitoring (PRM) for CML-modified peptides".

### Parallel reaction monitoring (PRM) for CML-modified peptides.

Twenty-two peptides containing CML-modification were selected among the most confident and consistently identified peptides from CMLpepIP and their isotopically labeled version (heavy arginine (U-$^{13}C_6$;U-$^{15}N_4$) or lysine (U-$^{13}C_6$; U-$^{15}N_2$) at the C-term was added) synthesized by JPT Peptide Technologies GmbH (Berlin, Germany). Lyophilized peptides were delivered and reconstituted in 20% (v/v) acetonitrile,

0.1% (v/v) formic acid and further pooled together in a ratio 1:1. An aliquot of the pooled peptides, corresponding to approximately 150 fmol per peptide, was analyzed by both DDA and DIA LC-MS/MS and used for assay generation using Spectrodive v.9-10 (Biognosys AG, Schlieren, Switzerland).

Peptides were separated using a nanoAcquity UPLC M-Class system (Waters, Milfors, MA, USA) with a trapping (nanoAcquity Symmetry C18, 5 μm, 180 μm × 20 mm) and an analytical column (nanoAcquity BEH C18, 1.7 μm, 75 μm × 250 mm). The outlet of the analytical column was coupled directly to a Q-exactive HF (Thermo Fisher, Waltham, MA, Germany) or an Orbitrap Fusion Lumos (Thermo Fisher Scientific, Waltham, MA, USA) using the Proxeon nanospray source. Solvent A was water, 0.1% (v/v) formic acid and solvent B was acetonitrile, 0.1% (v/v) formic acid. Peptides were eluted via the analytical column with a constant flow of 0.3 μl/min. During the elution step, the percentage of solvent B increased in a nonlinear fashion from 0% to 40% in 40 min. Total runtime was 60 min, including clean-up and column re-equilibration. PRM acquisition was performed in a scheduled fashion for the duration of the entire gradient (after instrument calibration in an unscheduled mode) using the "DIA" mode with the following settings: resolution 120,000 FWHM, AGC target $3 \times 10^6$, maximum injection time (IT) 250 ms, isolation window 0.4 m/z. For each cycle, a "full MS" scan was acquired with the following settings: resolution 120,000 FWHM, AGC target $3 \times 10^6$, maximum injection time (IT) 10 ms, scan range 350 to 1650 $m/z$. Peak group identification was performed using SpectroDive and manually reviewed. Quantification was performed using spike-in approach.

For CML sites quantification in mouse organs, not enriched digested peptides were spiked with synthetic heavy peptides at a concentration of 6 fmol/μl and analyzed by scheduled PRM as described above, using the peptide sequences listed in Supplementary Data 2 (sheet 7_PRM list).

For absolute quantification of tubulin CML modification in HUVEC, digested peptides were spiked with synthetic heavy peptides at a concentration of 3 fmol/μl, fractionated (as described in "Automated high pH peptide fractionation"), and the obtained fractions analyzed by scheduled PRM as described above using the peptide sequences listed in Supplementary Data 2 (sheet 8_PRM list_HUVEC).

For all relative quantifications, quantities of endogenous peptides (light) exported from SpectroDive were normalized across samples by dividing for the integrated intensity of the Base Peak Chromatogram extracted for each sample from "full MS" scans using Xcalibur v4.1. For absolute quantification, the ratio between endogenous (light) and reference (heavy) peptides were used.

**Immunoblot for mono- and polyubiquitinated proteins of MEF treated with glyoxal.** The same cell lysates used for proteasome activity assay were used. Lysates were thawed and centrifuged at 20817 × g, for 15 min at 4 °C to remove debris and the supernatant was transferred to a new tube. Based on EZQ assay performed on it, 20 μg of proteins were used. Samples were sonicated for 10 cycles (1 min ON and 30 s OFF) using a Bioruptor Plus with 4× loading buffer (1.5 M Tris pH 6.8, 20% (w/v) SDS, 85% (v/v) glycerin, 5% (v/v) β-mercaptoethanol). Proteins were separated on 4–20% Mini-Protean® TGX™ Gels (BioRad, Neuberg, Germany) by sodium dodecyl sulfate polyacrylamide gel electrophoresis (SDS-PAGE) using a Mini-Protean® Tetra Cell system (BioRad, Neuberg, Germany). Proteins were transferred to a nitrocellulose membrane (Millipore) using a Trans-Blot® Turbo™ Transfer Starter System. Membranes were stained with Ponceau S for 5 min on a shaker (Heidolph Duomax 1030), washed with milliQ water, imaged on a Molecular Imager ChemiDoc™ XRS + Imaging system (BioRad, Neuberg, Germany) and destained by 2 washes with PBS and 2 washes in TBST for 5 min (Tris-buffered saline (TBS, 25 mM Tris, 75 mM NaCl), with 0.5% (v/v) Tween-20). After incubation for 1 h in blocking buffer (3% bovine serum albumin (w/v) in TBST), membranes were stained overnight with primary antibodies against mono- and K29-, K48-, and K63-linked mono- and polyubiquitinylated proteins or β-actin diluted in blocking buffer (1:5000) at 4 °C on a tube roller (BioCote® Stuart® SRT6). Membranes were washed 3 times with TBST for 10 min at room temperature (RT) and incubated with horseradish peroxidase coupled secondary antibodies at room temperature for 1 h (1:2000 in 0.3% (w/v) BSA in TBST). After 3 more washes for 10 min in TBST, chemiluminescent signals were detected using an ECL (enhanced chemiluminescence) detection kit (Thermo Fisher Scientific, Waltham, MA, USA). Signals were acquired on the Molecular Imager ChemiDoc™ XRS + Imaging system (BioRad, Neuberg, Germany) and analyzed using the Fiji application[85]. Membranes were stripped using stripping buffer (1% (w/v) SDS, 0.2 M glycine, pH 2.5), washed 3 times with TBST, blocked and incubated with the second primary antibody, if necessary.

**Native gel electrophoresis and in-gel proteasome assay.** Purified 26S proteasome (Enzo Life Sciences, NY, USA) was thawed on ice and immediately used for the analysis. 2 μg of proteasome per replicate were diluted in buffer (10 mM TRIS, containing 25 mM potassium chloride, 1.1 mM MgCl₂, 0.1 mM EDTA, 1 mM DTT, 1 mM sodium azide, 2 mM ATP, pH 7.0, and 35% glycerol) to reach the concentration of 0.2 μg/ μl, and incubated for 0, 1, 2, and 4 h with 1 mM GO at 37 °C. As negative control the same amount of proteasome was treated with 100 μM MG132 for 4 h at 37 °C. Seventy percent of the sample was subjected to native gel electrophoresis to reveal the various proteasome complexes (30S, 26S, 20S) using native PAGE 3 to 12% Bis-Tris gel (Invitrogen, Carlsbad, US). The gel was running for 2 h at 4 °C with constant 150 V. The gel was then incubated in 50 mM Tris, pH 7.4, 5 mM MgCl₂, 1 mM ATP, and 100 μM proteasome substrate Suc-LLVY-AMC for 15 min at 37 °C to assay chymotrypsin-like (CT-L) activity. Proteasome bands were visualized under UV. Following the CT-L activity assay, proteins in native gels were wet transferred for 2 h, at 4 °C to nitrocellulose membranes. Blots were then blocked in 3% BSA, 0.5% TBST and immunoblotted with monoclonal antibodies against CML and against the alpha 1, 2, 3, 5, 6, and 7 subunits of the 20S proteasome (1:1,000 dilution in 0.2% BSA, 0.1% Tween 20 in PBS) overnight at 4 °C. Membranes were further incubated with horseradish peroxidase-conjugated secondary antibodies for 1 h. Proteins were detected using the Pierce™ ECL Western Blotting Substrate (Thermo Scientific, Waltham, USA) and the intensity of the band were analyzed by Image Lab TM (v6.0.1, Bio-Rad Laboratories, Inc., Hercules, USA).

**Proteasome activity assay on MEF treated with glyoxal.** Proteasome activity assay was performed using the 20S proteasome activity assay kit (Millipore, Billerica, MA, USA) following the manufacturer's instructions. Briefly, cell pellets were thawed and ice-cold lysis buffer (50 mM HEPES pH 7.5, 5 mM EDTA, 150 mM NaCl, 1% (v/v) Triton X-100, 2 mM ATP) was added. Samples were left on ice for 30 min with quick vortex steps any 10 min and then centrifuged at 20817 × g for 15 min at 4 °C to remove any debris. Protein estimation was performed in a small aliquot using the EZQ® Protein Quantitation Kit (Thermo Scientific, Waltham, MA, USA). 50 μg of protein extract were incubated with fluorophore-linked peptide substrate (LLVY-7-amino-4-methylcoumarin [AMC], from the same kit) for 60 min at 37 °C. Proteasome activity was measured by quantification of fluorescent units derived from cleaved AMC at 380/460 nm using a microplate reader m1000 (Tecan). Samples were measured together with positive control supplied with the kit.

**Glyoxalase 1 activity assay.** Glyoxalase 1 activity was measured using the ab241019 Glyoxalase I Assay Kit (Colorimetric, Abcam, Cambridge, UK) following manufacturer's instructions. Briefly, equal volumes of tissue homogenates (obtained as described above) from heart, liver and kidney, corresponding to an estimated protein amount of 50 μg were analyzed. The substrate mix was prepared and left for 10 min at room temperature in the darkness to allow the formation of hemimercaptal. Afterwards, samples were transferred to a 96 well-plate and the substrate mix was added. The formation of S-D-lactoylglutathione (SLG) was monitored for 30 min in a kinetic mode by using a microplate reader m1000 (Tecan) at OD 240 nm. The activity was then calculated using the formula suggested by the vendors and normalized for equal loading using immunoblot against β-actin (as described in "Immunoblot for mono and polyubiquitinated proteins of MEF treated with glyoxal") performed on the same homogenates.

**Other data analyses.** Gene Ontology enrichment based on ClueGO v2.5.6[86] was performed using the protein IDs reported in the Supplementary Data 2 as input. The following parameters were applied for all mouse organs: Kappa score threshold 0.4; minimum percentage =5 and number of genes =8; statistical test used = enrichment/depletion, two-side hypergeometric test. For the overlap between HUVEC and MEF, the same parameters were applied, except: minimum percentage =4 and number of gene =2. KEGG pathway and Gene Ontology enrichments for MEF and HUVEC total proteome were performed by Gene Set Enrichment Analysis (GSEA) with WebGestalt[87] using the entire lists of quantified proteins ranked on the basis of the measured log2 fold changes.

**HUVEC cell counting.** Cells were washed with PBS, detached with 250 μl trypsin/ EDTA and carefully mixed with 800 μl HEPES/FCS. An aliquot was counted using a Neubauer counting chamber.

**HUVEC lysis and immunoblot.** HUVEC were lysed in ice-cold Tris buffer (50 mM Tris (pH 7.4), 2 mM EDTA, 1 mM EGTA, 50 mM NaF, 10 mM Na₄P₂O₇, 1 mM Na₃VO₄, 1 mM DTT, 1% Triton X-100, 0.1% SDS, 1 mM PMSF, 10 μl/ml protease inhibitor cocktail) for 15 min on ice, scraped and centrifuged (700 × g, 6 min). Aliquots of supernatants were used for protein determination according to Lowry. Lysates were supplemented with Laemmli buffer, subjected to SDS-PAGE (25–50 μg lysate protein/lane) and blotted onto polyvinylidene difluoride (PVDF) membranes. The membranes were blocked for 1 h in TBST (20 mM Tris (pH 7.6), 137 mM NaCl, 0.1% (w/v) Tween® 20) containing 5% non-fat dried skimmed milk. Thereafter, blots were incubated with primary antibodies (diluted in TBST containing 5% BSA) overnight at 4 °C followed by incubation with horseradish peroxidase-conjugated secondary antibodies for 1 h. Proteins were detected using the enhanced chemiluminescence (ECL) reagent (GE Healthcare, Chicago, IL, USA) or Western Lightning Plus-ECL reagents (Perkin Elmer, Waltham, MA, USA). The intensity of bands was quantified by densitometry using the ImageJ 1.52p software (NIH, Bethesda, MD, US). Phospho- and acetylation-specific signals were normalized to the signal of the respective total proteins. For evaluation of protein expression, the signals were normalized to β-actin.

**5-Bromo-2′-deoxyuridine (BrdU) proliferation assay.** Glyoxal treatment of HUVEC was performed for 24 h in cell culture dishes and, after reseeding, for

another 24 h in 24-well plates. Thereafter, cells were starved in starvation medium (M199, 2% FCS, 7.5 U/ml heparin, 680 μM glutamine, 100 U/ml penicillin, 100 μg/ml streptomycin) for 4 h and stimulated with bFGF for 24 h. DNA synthesis was analyzed using the Cell Proliferation ELISA, BrdU (colorimetric) kit from Roche Diagnostics (Mannheim, Germany) according to the manufacturer's protocol. In brief, during the last 5 h of experimental incubation 10 μM BrdU were added. Then, cells were fixed for 30 min in Fix/Denat solution. 200 μl peroxidase-conjugated anti-BrdU-antibody (1:100) were added per well and incubated for 75 min protected from light. Cells were washed three times with PBS before adding 200 μl substrate solution. Reaction was stopped after 5–20 min depending on the color development by adding 50 μl of 1 M $H_2SO_4$ for 1 min. The solution was transferred into a 96-well plate and absorption was measured at 450 nm.

**CFSE proliferation assay.** HUVEC were washed twice with warm (37 °C) HEPES buffer containing 0.25% human serum albumin (HEPES/HSA) and incubated with 5 μM CFSE for 15 min at 37 °C. After two washing steps with warm HEPES/FCS, cells were incubated with GO in 1 ml experimental medium for the indicated times. Thereafter, cells were washed twice with PBS, detached with 300 μl trypsin/EDTA and transferred to 700 μl of HEPES/FCS. The cell suspension was pooled with 1 ml HEPES/FCS obtained after rinsing the dish. 10 ml of HEPES buffer were added and samples centrifuged (500 × g, 1 min). Pellets were resuspended in 300 μl PBS and subjected to flow cytometric analysis. The gating strategy is described in Supplementary Fig. 9a. Median values were evaluated using the FlowJo™ v7.6.5 software (Becton, Dickinson and Company, Ashland, OR, US).

**Angiogenesis assay.** Glyoxal treatment was performed for 24 h in 90 mm-cell culture dishes and, during generation of spheroids, for another 24 h in 96-well plates. Spheroids were prepared according to a published protocol[88]. Cells suspended in growth medium were mixed at a 4:1 ratio with methylcellulose (12 mg/ml) and 3000 cells/well were cultured in 96-well round-bottom plates for 24 h. The formed spheroids were collected, centrifuged (200 × g, 4 min) and washed with HEPES buffer. Then, spheroids were transferred to a fibrinogen solution (1.8 mg/ml in HEPES buffer) containing 20 U/ml aprotinin to obtain a suspension with approximately 100 spheroids per ml. 300 μl of this suspension together with 0.2 U thrombin were added per well of a 24-well plate. The plate was incubated for 20 min at 37 °C to allow the formation of a fibrin gel. To equilibrate the gel with medium, M199 containing 2% FCS, 680 μM glutamine, 100 U/ml penicillin and 100 μg/ml streptomycin was added twice for 15 min. Thereafter, spheroids were cultured in the same medium and stimulated with 50 ng/ml VEGF for 24 h. Finally, spheroids were fixed on ice by adding 1 ml 4% paraformaldehyde per well for 10 min. After two washing steps with PBS, spheroid sprouting was viewed by light microscopy and pictures were taken (Axio-Vert 200, Carl Zeiss, Oberkochen, Germany). The number of sprouts was analyzed using cellSens image analysis software (Olympus, Tokyo, Japan).

**Cell synchronization with double thymidine block.** Cells were synchronized at G1/S boundary by using double thymidine block[89]. One day after seeding, 2 mM thymidine was added and incubated with cells for 18 h. After washing (twice with PBS, once with M199), cells were cultured in full growth medium for 9 h and subsequently incubated with 2 mM thymidine for another 18 h. Cells were washed as described and GO was added in experimental medium for the indicated times.

**Cell cycle analysis.** During the last 30 min of GO treatment, 10 μM 5-ethynyl-2′-deoxyuridine (EdU) were added to HUVEC cultured on 90 mm-dishes. Cells were washed twice with PBS, detached with 1 ml trypsin/EDTA and transferred to 4 ml of HEPES/FCS. The cell suspension was pooled with 4 ml HEPES/FCS obtained after rinsing the dish and samples were centrifuged. All centrifugations were carried out at 500 × g for 3 min at room temperature if not otherwise stated. Cell pellets were washed once in 1 ml BSA buffer (1% BSA in PBS), resuspended in 100 μl of the same buffer, fixed with 100 μl of 4% paraformaldehyde for 15 min, centrifuged and resuspended in 300 μl PBS. 700 μl of 100% ethanol were added dropwise under constant gentle shaking and samples were frozen overnight. The next day, cells were centrifuged (700 × g, 3 min, room temperature), washed in 1 ml BSA buffer and permeabilized in 100 μl Triton-based BSA buffer (TBB, 0.2% Triton X-100 in BSA buffer) for 30 min. Then, cells were centrifuged, resuspended in 500 μl BSA buffer, incubated for 1 h, centrifuged again and incubated in 150 μl of primary antibody solution (1:200 p-H3 (S10) antibody in TBB) for 2 h. This was followed by another addition of 500 μl TBB, centrifugation and incubation of cells in 150 μl of secondary antibody solution (1:500 AF488 goat anti-rabbit antibody in TBB) for 1 h. Next, after adding 500 μl TBB, centrifugation and an additional washing with 1 ml BSA buffer, cells were incubated in 100 μl of Click-iT reaction cocktail (2.5 mM CuSO4, 1:200 AF647 azide (stock solution 180 μg/ml), 50 mM sodium ascorbate in PBS) for 30 min. Subsequently, another washing with 1 ml BSA buffer was performed and cells were incubated in 300 μl BSA buffer containing 1 μg DAPI for 30 min. Samples were subjected to flow cytometric analysis with triple detection of AF488, AF647 and DAPI. Data were acquired using a BD FACSCanto II or BD LSRFortessa equipped with a violet (405 nm; filter set for the DAPI channel—450/50BP), blue (488 nm; filter set for the FITC/AF488 channel—530/30BP; 502LP) and red (633 nm; filter set for the APC/AF647 channel—660/20BP) using BD FACSDIVA software v 7.0 (Becton, Dickinson and Company).

The gating strategy is described in Supplementary Fig. 9b. Percentages of cells in the respective cell cycle phases were evaluated using the FlowJo™ v7.6.5 software.

**Seahorse analysis of cells.** HUVEC were seeded into Seahorse XF96 Cell Culture Microplates (3,000 cells/well; Agilent Technologies, Santa Clara, CA, USA), incubated for 24 h and then treated with GO for 48 h. Mito stress test: Medium was replaced with Seahorse XF Base Medium (103334-100, Agilent Technologies, pH adjusted to 7.4), supplemented with 10 mM D-glucose, 2 mM L-glutamine and 1 mM sodium pyruvate. Cells were then cultured for another hour in a CO2-free incubator at 37 °C. Oxygen consumption rates (OCR) and extracellular acidification rates (ECAR) were monitored at basal conditions and after sequential injections of 2 μM oligomycin to block the mitochondrial ATP synthase, 2 μM carbonyl cyanide-4-(trifluoromethoxy)phenylhydrazone (FCCP) to uncouple oxidative phosphorylation and 2 μM antimycin A to fully inhibit mitochondrial respiration. Glycolysis stress test: Medium was replaced with Seahorse XF Base Medium (pH adjusted to 7.4) supplemented with 2 mM L-glutamine, and cells were cultured for another hour in a CO2-free incubator at 37 °C. OCR and ECAR were monitored at basal conditions and after sequential injections of 10 mM D-glucose, 2 μM oligomycin and 50 mM 2-deoxy-D-glucose, an inhibitor of glycolysis.

Measurements in both settings were performed in 3 min mix and 3 min measure cycles at 37 °C in six replicates per condition on a Seahorse XFe96 Analyzer (Agilent Technologies). OCR and ECAR were depicted as pmol/min and mpH/min, respectively, and normalized to the exact cell number of each well measured by high-content microscopy. Wave software (Agilent Technologies) was used to analyze the datasets.

**High-content microscopy.** Cell supernatants were removed from the Seahorse XF96 microplate and cells were fixed for 10 min with 100% methanol at room temperature. Cells were then washed once with PBS and incubated for 10 min with 1 μg/ml DAPI at room temperature. After two more washing steps with PBS, cell nuclei were counted on an ImageXpress Micro confocal high-content imaging system (Molecular Devices, San Jose, CA, USA).

**ATP measurements.** The intracellular ATP content was determined using the ATP Kit SL from Biotherma (Handen, Sweden) according to the manufacturer's protocol. At the end of experimental incubations, cell proteins were denatured by adding 500 μl ethanol per dish. After evaporation of the ethanol, 250 μl Tris buffer of the assay kit was added and one freezing/thawing cycle in liquid nitrogen was performed. Cells were then scraped off, centrifuged (700 × g, 5 min) and supernatants subjected to ATP measurements. For normalization, cells in identically treated dishes were lysed with solubilization buffer (100 mM NaOH, 1.9 M Na2CO3, 1% SDS) and the protein content was determined according to Lowry.

**Intracellular and mitochondrial ROS measurement.** 30 min before the end of the indicated GO treatment, HUVEC were washed with PBS, 600 μl of the respective staining solution were added, GO was re-added and incubation was completed. The staining solutions contained 5 μM CM-H2DCFDA or 3 μM MitoSOX™ (both Thermo Scientific, MA, Waltham, USA) diluted in HEPES/HSA for the detection of intracellular or mitochondrial ROS, respectively. Cells were washed with PBS, detached with 300 μl trypsin/EDTA and transferred to 700 μl HEPES/FCS. The cell suspension was pooled with 1 ml HEPES/FCS obtained from rinsing the dish, 10 ml HEPES buffer were added and cells centrifuged (500 × g, 1 min). Cell pellets were resuspended in 300 μl PBS and subjected to flow cytometry analysis. The gating strategy is described in Supplementary Fig. 9a. Median values were evaluated using the FlowJo™ v7.6.5 software.

**Detection of mitochondrial mass.** Thirty minutes before the end of the indicated GO treatment HUVEC were washed with PBS and 600 μl of 100 nM MitoTracker™ (Thermo Scientific, Waltham, MA, USA) diluted in HEPES/HSA were added. GO was re-added and incubation was completed. Cells were washed with PBS, detached with 300 μl trypsin/EDTA and transferred to 700 μl HEPES/FCS. The cell suspension was pooled with 1 ml HEPES/FCS obtained from rinsing the dish, 10 ml HEPES/HSA were added and cells centrifuged (500 × g, 1 min). Pellets were resuspended in 300 μl PBS and subjected to flow cytometry analysis. The gating strategy is described in Supplementary Fig. 9a. FlowJo™ v7.6.5 software was used to evaluate the median values of MitoTracker™-positive cells

**SA-β-Gal staining.** After GO incubation of second-passage HUVEC for 48 h, cells were washed twice with 1 ml cold PBS and fixed with 1 ml of a 2% formaldehyde/0.2% glutaraldehyde solution for 3 min. SA-β-Gal staining was performed according to a published protocol[90,91]. After two washing steps with PBS, 1.5 ml staining solution (40 mM citric acid/Na phosphate buffer, 5 mM K4[Fe(CN)6] •3H2O, 5 mM K3[Fe(CN)6], 150 mM NaCl, 2 mM MgCl2, 1:20 X-Gal (20 mg/ml in DMF) in water, pH 6.0) was incubated with cells at 37 °C overnight. Pictures were taken (EVOS™ FL Auto, Thermo Scientific, Waltham, MA, USA) and the number of SA-β-Gal-positive cells was counted using ImageJ 1.52p (NIH, Bethesda, MD, US) and normalized to the total cell number.

**γH2A.X (S139) immunofluorescence staining**. HUVEC were washed with warm (37 °C) HEPES buffer, fixed in ice-cold 4% paraformaldehyde for 15 min and washed twice with PBS. Cells were permeabilized in PBS containing 0.3% Triton X-100 for 5 min. Blocking solution (1% BSA-C, 5% goat serum in PBS) was added for 1 h. After two washing steps with PBS, samples were incubated in γH2A.X (S139) antibody diluted in blocking solution (1:400) in a humidified chamber overnight at 4 °C. Cells were washed twice with PBS and incubated with AF488-labeled goat anti-rabbit secondary antibody (1:500 in blocking solution) for 2 h. After two washing steps with PBS, DAPI (1 µg/ml in PBS) was added for 10 min and cells were washed three times with PBS before mounting the coverslips on microscopic slides using Fluoromount-G. Intensities of nuclear γH2A.X (S139) staining were measured using ImageJ 1.52p (NIH, Bethesda, MD, US).

**α/β-Tubulin and ac-Tubulin (K40) immunofluorescence staining**. Washing, fixation and permeabilization of HUVEC were performed as described for γH2A.X (S139) staining. Thereafter, cells were incubated for 2 h with primary antibodies against α/β-tubulin or ac-tubulin (K40) diluted in blocking solution (1:100). After two washing steps with PBS, an AF488-labeled goat anti-rabbit secondary antibody (1:500 in blocking solution) was added for 1 h. Cells were washed three times with PBS. Then, DAPI (1 µg/ml in PBS) was added for 10 min and cells were washed three times with PBS before mounting the coverslips on microscopic slides using Fluoromount-G. For immunofluorescence, a LEICA DMi8 TCS SP8 inverted laser scanning microscope was employed (Leica Biosystems, Wetzlar, Germany) and pictures were taken. Evaluation was performed via a blinded approach, in which cells with intensively stained tubulin filaments per high-power field were counted and normalized to the number of DAPI-stained nuclei. Alternatively, in experiments with long-term incubations of cells with 1–100 µM GO, the proportion of cells with low, intermediate and intensive tubulin staining normalized to DAPI-stained nuclei was determined.

**Tubulin in vitro modification**. For in vitro modification 2 µg/µl porcine tubulin dissolved in general tubulin buffer (Cytoskeleton Inc., Denver, CO, US) were incubated with 0.1, 1 or 10 mM GO in the presence of 1 mM GTP for 30 min on ice. After incubation, an aliquot of the sample was supplemented with Laemmli buffer, boiled for 5 min and subjected to CML immunoblot. Another aliquot was analyzed by mass spectrometry as described in the paragraph "LC-MS/MS based on Data-Dependent Acquisition (DDA) for CMLpepIP" to detect site-specific CML modification. The data were processed as described in the paragraph "Data processing for DDA for CMLpepIP" and searched against the Uniprot Sus Scrofa database (Proteome ID: UP000008227, reviewed and not entries combined, release date 2012).

**Tubulin polymerization assay**. Tubulin polymerization was performed using the fluorescent tubulin polymerization assay kit (Cytoskeleton Inc., Denver, CO, US). First, tubulin glycation was induced by adding 1 mM GO to 2 µg/µl tubulin and incubating the sample in a total volume of 40 µl and the presence of 25% glycerol for 30 min on ice. Subsequently, 10 µl of a 5 mM GTP solution were added and the samples were transferred to a prewarmed 96-well plate, whose wells contained 3 µM taxol, 20 µM nocodazole or vehicle (5 µl each). Microtubule formation was followed by measuring the time-dependent increase of the fluorescent signal (excitation: 340 nm, emission: 410 nm) at 37 °C.

**Statistics**. MEF experiments were performed in independent replicates using cells from different passages with n values as indicated. HUVEC experiments were performed in independent replicates using cells from different donors with n values as specified in figure legends. In all experiments, *n*-values represent biological replicates. Data are expressed as mean ± SEM, unless otherwise stated. Statistical significance was analyzed using one-way or two-way repeated measurement ANOVA with Holm–Šidák correction for multiple comparisons, unless otherwise stated. In boxplots throughout the manuscript, the horizontal line represents the median, the bottom and top of the box the 25th and 75th percentile, respectively, and the whiskers extend 1.5 folds the interquartile range. Tests were performed using SigmaPlot 14.0 (Systat Software GmbH, Erkrath, Germany), GraphPad Prism (GraphPad Software, LLC, v8.3.0), R (v.3.6.3) and R studio server (v. 1.2.5042). All specific p values are listed in Supplementary Data 6.

**Reporting summary**. Further information on research design is available in the Nature Research Reporting Summary linked to this article.

## Data availability

The source data of all analyses are supplied as Supplementary Data 1–5 and in the associated source data file. The mass spectrometry data generated in this study have been deposited to the ProteomeXchange Consortium via the PRIDE[91] partner repository under accession code PXD027526 (DDA data for CMLpepIP); PXD021883 (TMT-10plex data for organ aging proteome); PXD021985 (DIA for MEF and HUVEC cells); PXD021891 (DIA for acetylated peptides); PXD027468 (DIA for long-term treatment with low doses of GO). The Uniprot protein databases used for proteomic analysis are accessible at: https://ftp.uniprot.org/pub/databases/uniprot/previous_releases/release-

2016_01/. In addition, the proteomics data presented in this manuscript are available via a R shiny webserver (https://genome.leibniz-fli.de/shiny/orilab/CMLsites/). Source data are provided with this paper.

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

## Acknowledgements

The authors gratefully acknowledge support from the FLI Core Facilities Proteomics, FACS, functional genomics and the Mouse Facility. The authors acknowledge Konrad Böhm, Helmut Pospiech and Johannes Jungwirth (FLI, Jena) for providing advice and tools for cell cycle analysis, Elke Teuscher (Institute of Molecular Cell Biology, Jena) for her excellent technical assistance and the isolation and culture of HUVEC, Claudia Ender and Amod Godbole (Institute of Molecular Cell Biology, Jena) for taking immuno-fluorescent pictures, Titus Lohfink (Institute of Chemistry-Food Chemistry, Halle) for CML analysis in HUVEC, Max Tiessen and Domenico Di Fraia (FLI Jena) for implementing the R shiny webserver, and Julia Heiby and Ellen Späth (FLI Jena) for proof-reading the manuscript. R.H. receives funds from the Deutsche Forschungsgemeinschaft (DFG, RTG1715 and RTG2155). A.O. acknowledges funding from the DFG (RTG2155), the Else Kröner Fresenius Stiftung (award number: 2019_A79), the Deutsches Zentrum für Herz-Kreislaufforschung (award number: 81×2800193) and the Fritz-Thyssen foundation (award number: 10.20.1.022MN). This research was also supported by the European Regional Development Fund (Grant ID: EFRE HSB 2018 0019) and the federal state of Thuringia providing technical equipment. The FLI is a member of the Leibniz Association and is financially supported by the Federal Government of Germany and the State of Thuringia.

## Author contributions

Conceptualization: S.D.S., A.O., R.H. Investigation: S.D.S., K.S., J.M.K., A.L., T.L.R., T.B., T.D., C.H., C.M. Methodology: S.D.S., K.S., J.M.K. Data analysis: S.D.S., K.S., J.M.K., L.P., T.B, A.O., A.L., T.L.R., R.H. Resources: A.O., R.H., M.A.G. Supervision: Z.Q.W., M.A.G., A.O., R.H. Visualization: S.D.S., K.S., A.O. Writing—original draft: S.D.S., K.S., A.O., R.H. Writing—review & editing: T.B., L.P., and Z.Q.W.

## Funding

## Competing interests

The authors declare no competing interests.
