## [Peer Review File · Nature Communications]

REVIEWER COMMENTS

Reviewer #1 (Remarks to the Author):

Carboxymethyllysine (CML) is one of the protein modifications derived from advanced glycation end products (AGE). Despite that CML is known to accumulate during aging and in diseases including diabetes, a systematic mapping of CML sites across the human proteome is lacking. By immunoenriching CML peptides, this manuscript identified over a thousand CML sites in aged mouse organs and primary human cells, many of which have not been reported. The authors showed a decrease in GLO1 network protein expression level and GLO1 activity in aged mouse tissues, which is in accordance with the well-recognized upregulation of AGE level during aging, yet they found that not all CML sites increased in aged tissue. Among the CML substrates, there are tubulins and proteins involved in the ubiquitin-proteasome system or cell cycle regulation. The authors showed that at low [μ M] level, glyoxal upregulates 26S proteasome and leads to accumulation of ubiquitinated proteins, inhibits cell proliferation by arresting G1-S and G2-M transition, and reduces tubulin filament dynamics, which may affect mitosis. Possibility due to technical challenges, the authors did not establish whether these effects of glyoxal are mediated by protein CML, neither did they quantify the changes in CML level of proteins involved in the ubiquitin-proteasome system or cell cycle regulation in response to glyoxal treatment. The authors were able to detect an increase in tubulin K58 CML level with 1mM glyoxal treatment, however, how the CML modification contributes to tubulin filament stability is still unclear.

This manuscript is a valuable resource for the community in that it provides the first proteome-wide characterization for protein substrates and modification sites of CML, an important modification implicated in many pathological conditions. While the mechanisms remain incomplete, the finding that glyoxal induces cell cycle arrest and dysregulated protein degradation and that related proteins can be CML modified raises the possibility that, *in vitro* and *in vivo*, glyoxal's effects may be in part mediated by directly modifying the proteins involved in the biological processes. In general, adequate number of replicates are performed to ensure the validity of the results, and figures are clearly annotated. Therefore, this reviewer recommends that this manuscript be accepted after addressing the following concerns.

- The title of this manuscript, "Mapping sites of carboxymethyllysine modification on proteins reveals its consequences for proteostasis and cell proliferation", is not accurate, as the authors did not demonstrate that carboxymethyllysine is involved in proteostasis and cell proliferation.
- The glyoxal concentration used in this manuscript is \sim 1000 times higher than physiological level[1], which brings the question of whether the effects of glyoxal on cell proliferation, ubiquitin-proteasome system and tubulin dynamics still hold true *in vivo*. Are there any pathological conditions where glyoxal levels accumulate to [μ M] range? To demonstrate the *in vivo* relevance of these findings, the authors are suggested to validate their results *in vivo* or treat cells with physiological

level of glyoxal. Alternatively, the authors could measure glyoxal concentration in cells of their interest

- Authors claimed that carboxymethyllysine (CML) is one of the protein modifications derived from advanced glycation end product (AGE). How abundant is CML? What is the justification. If there is no solid evidence, this statement should be removed.
- Chemical reaction and regulation of CML is not familiar for many readers. This reviewer would suggest to include a figure to show this, either in main figure or in the sup info.
- They are suggested to measure intracellular Glyoxal levels in aged vs young mice tissue.
- In mouse tissue, did you identify other histone sites using IP-MS/MS? No other sites on histone H4 except for H4K92 is identified. Are there CML sites on N-terminal tails of histones?
- Figure 1G and 2B. How to explain the vast divergency of CML sites in different species and different organs? Are those organ-specific CML proteins also the most abundant proteins specifically in that organ? Because the modification is caused by chemical reaction, we assume that the modification is related with the protein abundance and glyoxal.
- Figure 2B and 2C. The author should specify in the figure legend whether the CML sites are mapped in old mouse tissue, or young mouse tissue, or combined.
- Figure 2D. Some PTM can be very dynamic and changes drastically from cell types to cell types. The statement that "CML modification compete with other PTMs" is overstated if the so-called overlapping PTMs are not detected under the same condition as CML. Therefore, this figure is not very helpful.
- Figure 2E. This panel shows that old mice and young mice can be distinguished by global proteomics data of heart, liver and kidney. However, it is not very relevant to the scope of this manuscript since this manuscript focuses on the biological functions of CML. This reviewer suggests the authors to explore instead whether old and young mice can be differentiated by their CML profile.
- Figure 2H and 2I. Among the 18 CML modifications that can be detected by PRM, how many of them are upregulated, downregulated, or not changed in aged organs?
- Figure 2. The authors are suggested to validate age-related total CML increase by CML western blot of young and aged mice organs.
- Figure 3D and 3E. How would you explain the seemingly contradictory observation that while KEGG enrichment analysis showed a downregulation of ubiquitin-mediated proteolysis pathway (3D) after 2mM GO treatment, while most 26S proteasome subunits are in fact upregulated by 2mM GO (3E)?
- Figure 3E left and 3G. It appears that this figure is a result of one replicate. Otherwise, the author should specify the number of replicates performed.

- Figure 3G. 26S proteasome subunits that have reduced thermal stability after GO treatment are not CML substrates (Figure S3), indicating a possibility that the reduced thermal stability of 26S proteasome may be regulated by CML-independent mechanisms.
- Figure 6B and 6E. GO induces tubulin filament stabilization, yet reduces tubulin filament stabilizing modification K40 acetylation. Could you please elaborate the possible mechanisms?
- Figure 3,4,5,6. Where applicable, validate the change of CML level of relevant proteins by PRM in different cell stages, during angiogenic sprouting, with or without glyoxal treatment. It is possible CML level changes for some but not all proteins during these processes. Then the site-specific quantification will provide a useful correlation information and suggest which proteins and which CML sites might be most relevant in these conditions.

Reference:

[1] Rabbani, N., Xue, M. & Thornalley, P.J. Dicarbonyls and glyoxalase in disease mechanisms and clinical therapeutics. *Glycoconj J* 33, 513–525 (2016). <https://doi.org/10.1007/s10719-016-9705-z>

Reviewer #2 (Remarks to the Author):

In the current manuscript by Sanzo et al, the authors developed specific antibody-based IP enrichment approach and used mass spectrometry to map the post translational modification of lysine residues called (ϵ -N) Carboxymethyl Lysine in proteome. CML modification is considered as one among major AGEs with relevance to metabolic disease and ageing associated disorders. Using isolated proteins from fibroblast and endothelial cells treated with glyoxal in vitro and from organs of young and aged mice authors identified several CML modification sites. Observed results suggest that glyoxal treatment at higher concentration promote CML modification and increased endogenous CML modification was detected in aged mice. The abundance of CML in older mice could be possibly due to impaired or less detoxification AGEs system including Glo1.

Further, the authors provide evidence to show that CML modification targets the Ubiquitin proteasome system and induce proteasome machinery. CML modification is also found to inhibit cell proliferation by regulating the expression of key cell cycle regulators. Finally, authors showed that glyoxal alters the key structural tubulin with several CML modification. Key finding is that CML modification competes with or reduce other PTM of proteins that are necessary for normal cell growth and function.

Overall the study is well designed, and the experiments were performed in required biological replicates with appropriate controls. The methods are detailed, and all the necessary data related to this manuscript was provided and raw data were submitted in public repository. The observed results support the claimed conclusions. Although the current approach did not cover the entire sites of CML modification due to acceptable technical difficulties the current data opens several avenues to understand the functional role of CML modification in aging and associated diseases.

The following are minor corrections and suggestions

- According to Fig 2F, there is no significant difference in Glo1 or Glo1 network proteins in heart muscle compared to kidney and liver. This suggest normal functioning of Glyoxalase system in heart muscle and chances for less CML modification in heart compared to liver and kidney. However, Fig 2H result suggest a significant CML modification in heart and kidney compared to liver, which seems contradicting. Do authors have any explanation for this?
- In Fig 2I, did authors analyzed CML modification of K263 and K154 in liver and kidney? What is CML modification status.
- There should be a space between SI unit and the value. E.g. 2 mM and not 2mM. Check all the figures and apply this throughout the manuscript.
- Page 15 line 15 – Kindly note that “strongly structured tubulin” seems an inappropriate term. I highly recommend removing it, until evidence is provided by biophysical experiments demonstrating increased strength. Also, I can see the staining intensity of tubulin increased, which is not sufficient to conclude the strength of tubulin filaments (MT). From the images Fig. 6B, Does the space between the microtubule are reduced leading to aggregation? This leads to next question, Does the expression of microtubule-associated proteins (MAPs), e.g. tau, changes or CML alters interaction between MAP and MT changes? If possible, using the existing the data set, identify the changes in levels of MAPs.
- Also, Page 20 (discussion about MT stability) – A) line 15 – The diameter of MT is 25 nm. Based on immunostaining, it is not possible to identify thinner MT filaments in control and thicker filaments in CML modified. May be MT filaments are more accessible for immunostaining after CML modification resulting in intense staining of MT filaments. Thus, without electron microscopy MT thickness cannot

be concluded. B) Line 16 & 17 – “Accordingly, our data reveal an increased stability of CML-modified microtubule”, without examination of in vivo MT dynamics, I will suggest removing this conclusion.

- Page 15, line 16, 17 – Nocodazole treatment cannot determine the status of microtubule dynamics. The currently data only suggest that MT are resistant to nocodazole, which cannot be interpreted as impaired MT dynamics. In vivo visualization of rate of MT growth and shrinkage upon treatment of GO (with tagged tubulin and EB1 protein) will be the conclusive experiment to demonstrate impaired MT dynamics. Modify the text and please only mention that MTs are resistant to nocodazole.

- What is the expression status of acetyl transferases proteins or deacetylases? Because, K40 which is not attached by GO as well as K58 which is attached by GO, both are reduced in acetylation. Please discuss this in the discussion section.

-

Minor issues:

- The possible reasons for several fold increase in CML levels at treatment of GO 1 mM Vs 0.5 mM should be discussed.

- What is the reason to use 2 mM of GO in Figure 3 and 1 mM of GO in Figure 4, as well as the treatment time (24 h Vs 48 h)? How 2 mM of GO affected the parameters that were measured in figure 4 (cell proliferation, mitochondrial mass, OCR, etc.)? Does 2 m GO highly toxic HUVEC cells but not to MEF?

- In figure 3D KEGG pathways analysis for enriched proteins in glyoxal treated MEF cells displayed upregulation of proteins involved in Oxidative phosphorylation, whereas glyoxal treatment reduced energy production and basal respiration vis oxidative metabolism in HUVEC cells treated with glyoxal figure 4F. Did this suggest cell specific effect of glyoxal on mitochondrial bioenergetics?

- Fig S4 delete legend corresponding to γ H2AX (S139) immunofluorescence.

- How long the cells were treated with glyoxal for senescence assays? after how many days of culturing the senescence assays (immunoblot analysis and SA-beta-gal assay) was performed? Include this detail in materials and methods.

- Page 20, line 32 and 33; Does glyoxal treatment hamper the function of global PTM machinery to favor glycation? It seems that although K40 is not glycated its acetylation is reduced upon glyoxal treatment. Did authors find other sites with reduced PTM without CML modification?

Page 6 line 17, 18 authors claim that CML targets protein in organ specific manner.

Reviewer #3 (Remarks to the Author):

Sanzo et al. generated CML modification maps using different cells and mouse tissues. CML is a widespread PTM, but its functions remain largely unknown. The result presented by this paper is probably the first large-scale dataset for CML, hence is a valuable resource for relevant studies. The experiments were well designed and performed, and the manuscript was written logically and clearly. I would recommend to accept this paper for publication after the authors address the following concerns.

Major

1. The greatest drawback of this study is that the enrichment efficiency of the commercial CML antibody is bad. In many cases, only less than 1% of all captured peptides were CML modified. The low specificity strongly hampers the impact of the result. The immunogen is very critical in developing such antibodies. If the immunogen is a single CML modified peptide or protein, this antibody may not be suitable for enrichment purpose in large-scale studies. The authors should find out the immunogen information and state it in the revised manuscript. More, if the authors continue to be interested in this topic, I strongly recommend the authors to develop their own antibody for enrichment in the future studies.
2. The biological consequence reported by this paper was largely associated with glyoxal treatment. However, whether this is directly caused by the CML modification was unknown. Besides induction of CML, many other known and unknown effects exist for glyoxal. I would suggest the authors to elucidate at least one CML-directed mechanism. For example, the authors can incubate purified proteasome with glyoxal in vitro, and then examine the CML sites on all proteasome subunits, and corresponding proteasome activity. This simple system will allow to exclude any other indirect effects caused by glyoxal. A similar in vitro experiment can also be considered for tubulin.
3. Is there any sequence motif featured by CML modification? The authors claimed that CML competes with other lysine modifications. I'm curious if the CML and Kac peptides share a common sequence motif. Moreover, were there any higher structural features observed on these CML localizations. As the CML is a non-enzymatic modification, I would think lysine residues at protein surface may be prone to be modified.

4. Figure 2FG. GLO1 had a very mild decrease at expression level in aged mouse hearts, but had an unproportioned decrease in its activity. How should we understand the relationship between the protein abundance and activity? Similarly, Kidney had a greater decrease in abundance, but less effect on activity.
5. Figure 3DE. Proteasome is the major component of UPS. Why did fig 3D show “ubiquitin-mediated proteolysis” was downregulated, but proteasome was increased in 3E upon glyoxal treatment?
6. The melting temperatures of proteasome subunits actually only changed a little with glyoxal treatment. Whether this could lead to an impairment on proteasome activity is uncertain. Do all of them (fig 3G) contain a CML modification? The melting curves of these proteins should be added in the supplementary files. More, the TPP results for non-proteasome proteins with CML modification should be added. Do they also have a decreased melting temperature with glyoxal? Figure S3E is inappropriate just because only a small subgroup of the total proteome could be modified by CML.
7. Can the authors estimate modification degree (stoichiometry) of CML? Taken K40 on tubulin as an example, how much was the acetylation and CML? This is again relevant to the competition mechanisms between CML and other lysine modification. Lysine acetylation frequently occurs at a very low level (<1%) on non-histone proteins, how CML competes with Kac is an unanswered question. Why doesn't CML just choose to modify the remaining free lysine? As far as this reviewer knows, competition is not a dominant mechanism in regulating different lysine modifications.
8. Page 4, line 5. Were there any criteria in choosing peptides for PRM analysis? I wonder besides the two successful examples shown in figure 1E and F, how about other 14 validated peptides? Could the authors include those data in the supplementary files?
9. The authors claimed that CML is likely to modify high abundant proteins, which might be an artifact. The antibody used in this study has low specificity and sensitivity, thus it just couldn't enrich those CML modified peptides from low abundance proteins.
10. Page 6 Line 12. The observed lower CML level in brain is just against the result that CML tends to affect slow turnover proteins. Brain contains many more long-lived proteins than liver does. Could the authors provide any explanation?
11. It's very interesting to know CML tends to occurs on slow turnover proteins. However, as CML is a non-enzymatic modification, how was this preference achieved?
12. The authors should carefully check if iodoacetamide-induced carbamidomethylation on lysine, though it mostly happens on cysteine, interferes the CML identification. The mass gains of these two modifications are 57 and 58, respectively. A mis-picking of monoisotopic peak may lead to misidentifications of CML. More, the exact delta masses used for modifications in database search should be added in the method.
13. CML theoretically may not be easily cut by trypsin. The authors may want to check if all identified CML sites are non-C-terminal lysine. This will be a useful evidence to enhance the identification confidence.

14. The CML modifies different target proteins in different samples. For example, the protein quality control related proteins were not significantly CML modified in mouse organs like they were in the MEF cells. Is there any data showing how CML on proteostasis related proteins in mouse tissues and mouse aging?

Minor

1. Can glyoxal directly react with lysine and induce CML in vitro?
2. More information, such as catalog number and immunogen, about the antibody used for enriching CML-containing peptides should be provided.
3. Figure 4A. 38.6% of CML proteins are nucleus proteins (upper). Why isn't "nucleus" an enriched term in GO (lower)?
4. Page 3, Line 29. The authors stated the CMLpepIP was "reproducible". However, the data in Figure S1a and S1b were not about reproducibility. The data in figure S1c actually showed the CMLpepIP was not ideal in reproducibility, as only a very small fraction of sites could be detected in all three replicates. For example, the 3rd replicate of MEF control group only had 19 CML sites, whereas the 2nd replicate got 155 sites. I can imagine that the quantification will bear even larger variations between replicates. The authors should rephrase their statement and deliver the right message.
5. Supplementary tables should be organized and named better. There is no table number in the excel files.
6. Though the authors used antibody against CML, I'm curious if the authors got any carboxymethylation at other positions, such as N-terminus and cysteine.

Carboxymethylation of proteins – mapping sites of modification and consequences for proteostasis and cell proliferation

Point-to-point reply to the reviewer's comments

We would like to thank all the reviewers for their constructive criticisms that helped us to improve our manuscript. Below we address the reviewers' comments in a point-to-point reply. In addition, we have labeled new or altered text passages in the manuscript in red.

Reviewer's comments

Reviewer #1

Carboxymethyllysine (CML) is one of the protein modification derived from advanced glycation end product (AGE). Despite that CML is known to accumulate during aging and in diseases including diabetes, a systematic mapping of CML sites across the human proteome is lacking. By immuno-enriching CML peptides, this manuscript identified over a thousand CML sites in aged mouse organs and primary human cells, many of which has not been reported. The authors showed a decrease in GLO1 network protein expression level and GLO1 activity in aged mouse tissues, which is in accordance with the well-recognized upregulation of AGE level during aging, yet they found that not all CML sites increased in aged tissue. Among the CML substrates, there are tubulins and proteins involved in ubiquitin-proteasome system or cell cycle regulation. The authors showed that at low [mM] level, glyoxal upregulates 26S proteasome and leads to accumulation of ubiquitinated-proteins, inhibits cell proliferation by arresting G1-S and G2-M transition, and reduced tubulin filaments dynamic, which may affect mitosis. Possibility due to technical challenges, the authors did not establish whether these effects of glyoxal are mediated by protein CML, neither did they quantify the changes in CML level of proteins involved in ubiquitin-proteasome system or cell cycle regulation in response glyoxal treatment. The authors was able to detect an increase in tubulin K58 CML level with 1mM glyoxal treatment, however, how the CML modification contribute to tubulin filament stability is still unclear. This manuscript is a valuable resource for the community in that it provides the first proteome-wide characterization for protein substrates and modification sites of CML, an important modification implicated in many pathological conditions. While the mechanisms remains incomplete, the finding that glyoxal induces cell cycle arrest and dysregulated protein degradation and that related proteins can be CML modified raises the possibility that, *in vitro* and *in vivo*, glyoxal's effects may be in part mediated by directly modifying the proteins involved in the biological processes. In general, adequate number of replicates are performed to ensure the validity of the results, and figures are clearly annotated. Therefore, this reviewer recommend that this manuscript be accepted after addressing the following concerns.

We thank the reviewer for his positive overall judgment of our work.

Comment 1: The authors did not establish whether the effects of glyoxal are mediated by protein CML neither they did quantify the changes of CML levels of proteins involved in UPS or cell cycle regulation. How K58 CML modification contributes to tubulin stability is still unclear

We have now included experimental evidence that CML modification of tubulin *in vitro* is sufficient to alter its dynamics (see also reply to Reviewer 3, comment 2). We have also demonstrated that a prolonged (14 days) exposure to lower doses of glyoxal is sufficient to reduce growth and perturb cell cycle dynamics in HUVEC. Under these experimental conditions, we did not observe induction of DNA damage as seen for short-term incubations with 1 mM glyoxal, but we were still able to detect CML modification on K58 of tubulin (see also reply to Reviewer 1, comment 3). Both results point to the fact that CML-modification of tubulin is involved in glyoxal-mediated inhibition of cell proliferation, although the contribution of individual modified sites such as K58 remains to be determined.

Page 16, line 19: *To understand whether CML modification of tubulin affected its polymerization behavior, we exposed purified tubulin to glyoxal. This triggered CML modification at sites that overlap with the ones identified in glyoxal-treated cells (Fig. 6D and Supplementary Fig. 7B) and led to a significantly lower polymerization in vitro (Fig. 6E). These data suggest changes of microtubule dynamics upon glyoxal treatment, which may contribute to the observed reduction of cell proliferation by impeding mitosis.*

Figure 6. Glyoxal (GO) affects post-translational modification and dynamics of tubulin.

D. Purified porcine tubulin incubated with GO for 30 min on ice was analyzed in immunoblots. $n=5$. **E.** Purified porcine tubulin was pre-incubated with GO (1 mM, 30 min, on ice) and polymerization induced by addition of cofactors at 37 °C was monitored in a fluorescence-based assay. Controls: taxol (3 μ M) or nocodazole (20 μ M). $n=3-6$, mean \pm SEM. Statistical significance was analyzed using one-way repeated measurement ANOVA corrected via Holm-Šidák method for areas under the curves. * $p < 0.05$ vs. control.

Surprisingly, we have also shown that CML modification of the proteasome *in vitro* is not sufficient to alter its major proteolytic activity, suggesting that the inhibition of proteasome activity observed in cells might be mediated by indirect mechanisms, e.g., modification of regulators of proteasome activity or excessive accumulation of glycated substrates (see reply to Reviewer 3, comment 2). In summary, our new experiments provide two examples of both direct, CML-mediated as well as indirect effects of glyoxal on protein function.

Comment 2: The title of this manuscript, “Mapping sites of carboxymethyllysine modification on proteins reveals its consequences for proteostasis and cell proliferation”, is not accurate, as the authors did not demonstrate that carboxymethyllysine is involved in proteostasis and cell proliferation

We thank the reviewer for pointing this. Given the reviewer’s comment and the fact that we have now identified not only sites of CML but also carboxymethylation of protein N-termini, we have modified the title of our manuscript as follows:

Carboxymethylation of proteins – mapping sites of modification and consequences for proteostasis and cell proliferation

Comment 3: The glyoxal concentration used in this manuscript is ~1000 times higher than physiological level [1], which brings the question of whether the effects of glyoxal on cell proliferation, ubiquitin-proteasome system and tubulin dynamics still hold true *in vivo*. Are there any pathological conditions where glyoxal levels accumulate to [mM] range? To demonstrate the *in vivo* relevance of these findings, the authors are suggested to validate their results *in vivo* or treat cells with physiological level of glyoxal. Alternatively, the authors could measure glyoxal concentration in cells of their interest.

We thank the reviewer for raising this important point. Our rationale for choosing relatively high doses of glyoxal was driven by the necessity to emulate a process (carboxymethylation) in cell culture that in living organisms occurs on a time scale of months or years. However, we are aware that the concentrations of glyoxal that we supplemented to cell culture media are 2-3 orders of magnitude higher than what has been estimated in tissues. Therefore, to address this comment, we have carried out two complementary sets of experiments:

1. We used LC-MS to quantify the intracellular levels of glyoxal following treatment of HUVEC with exogenous glyoxal or vehicle control in growth medium containing 20 % serum. Our analysis demonstrates that treatment with 1 mM glyoxal for 24 h raises the intracellular levels of glyoxal approximately 12-fold of physiological levels. We have included this analysis in Fig. 1B of the revised manuscript and added the following text to the results paragraph of the revised manuscript.

Page 3, line 17: *The uptake of glyoxal in cells was validated in HUVEC, where intracellular levels increased by ~12 fold after treatment with 1 mM glyoxal for 24 h (Fig. 1B).*

Figure 1. Antibody-based enrichment of CML-modified peptides.

B. Absolute quantification of intracellular glyoxal (GO) levels in HUVEC after treatment with GO (1 mM, 24 h). n=5, mean + SD, ** p<0.01, paired t-test.

2. To mimic conditions of chronic exposure to glyoxal, we have repeated HUVEC cell culture experiments using lower concentrations of glyoxal (1 - 100 µM) applied for a longer period of time (14 days). We could show that these treatments also led to inhibition of cell proliferation, which was accompanied by an increased staining intensity of tubulin filaments, and, in response to 100 µM glyoxal, CML modification at lysine 58 (K58) of the tubulin β-4B chain. The abundance of proteins involved in cell cycle regulation was largely unaffected and only mild oxidative stress and moderate changes in cell cycle distribution occurred.

We have included this new data set in the manuscript (see result part, chapter *Long-term treatment with lower doses of glyoxal inhibits endothelial proliferation*, page 19, and Fig. 7 and Supplementary Fig. 8).

Together, these data indicate that carboxymethylation of tubulin may mediate tubulin dysfunction and inhibition of cell proliferation under pathophysiological relevant conditions. Under *in vivo* conditions, the exposure to pathophysiological doses of glyoxal is much longer. We speculate that the effects described here may then be induced over time. Carboxymethylation of proteins may be an initial event, later followed by oxidative stress, stress responses and changes of the proteome.

Figure 7. Long-term treatment with low doses of glyoxal (GO) impairs proliferation and affects tubulin.

A-H. HUVEC were treated with 1-100 μM GO for 14 d starting one day after seeding (day 1 - 15). **A. Left**, cell numbers were counted over time. **Right**, doubling time and proliferation rates were calculated based on cell numbers of days 11 and 15. $n=8$. **B.** HUVEC were subjected to cell cycle analysis by flow cytometry applying a triple staining method (EdU, p-H3 (S10), DAPI). The proportion of cells in the respective cell cycle phases is shown. $n=6$. **C,D.** Cells were subjected to immunoblot analysis. $n=6$. **E.** Intracellular ROS (dichlorodihydrofluorescein diacetate (H2DCFDA)) and mitochondrial ROS (MitoSOX) were measured by flow cytometry. $n=6-8$. **F.** Cells were stained using antibodies against $\alpha\beta$ -tubulin and DAPI. Representative pictures (left) and the percentage of cells showing low, intermediate, or strong staining of microtubules per high-power field (right) are shown. Scale bar=20 μm . $n=5$. **G.** Validation of CML-modified peptides by parallel reaction monitoring (PRM) using heavy spike-in peptides. **H.** Relative quantification of INVYYNEATGGK[CML]YVPR (from TUBB4B) in GO-treated HUVEC. Quantification was performed on elution samples after CMLpepIP. **F-H.** A treatment with 1000 μM GO for 48 h was used as the positive control (†). **A-F,H:** Bar graphs show mean + SEM.

Statistical significance was analyzed using one-way repeated measurement ANOVA corrected via Holm-Šidák method. * $p < 0.05$ vs. control. Related to Supplementary Fig. 8 and Supplementary Table 4.

Supplementary Figure 8.

A-C. One day after seeding, 1 - 100 μM glyoxal (GO) was added to HUVEC and treatment was continued for 14 d (day 1 - 15). A treatment with 1000 μM GO for 48 h was used as positive control (†) (A,B). **A.** Principal component analysis of proteome data from HUVEC treated with 1, 10, 100, 1000† μM GO and untreated control cells (Ctrl). The smaller dots represent individual samples and the larger dots the centroids of each treated/untreated-matched group. Ellipses represent 95 % confidence intervals. The percentage of variance explained by the first two principal components (PC) axes is reported in the axis titles. $n=4$. **B.** Heatmap representing protein abundance changes of proteins involved in cell cycle, used for the analysis in Fig. 5C. Cell cycle proteins were selected according to Reactome annotation and filtered as significantly altered expression (absolute log2 fold change > 0.58 and $q < 0.05$) in the condition 1000 μM†. Additionally, the protein expression of Hmx1 (Heme oxygenase 1) is shown as an example of a protein consistently affected by all GO treatments. **C.** Detection of tubulin K40 acetylation. HUVEC were treated with GO, lysed and subjected to immunoblot analysis. $n=6$. Related to Fig. 7 and Supplementary Table 4.

Comment 4: Authors claimed that carboxymethyllysine (CML) is one of the protein modifications derived from advanced glycation end product (AGE). How abundant is CML? What is the justification. IF there is no solid evidence, this statement should be removed.

We thank the reviewer for this comment. Indeed, previous studies that have quantified absolute levels of AGEs in plasma and tissues have shown that CML is one of the most abundant AGEs. In order to clarify this point, we have included citations of these two key publications in the introduction paragraph of the revised manuscript:

Page 2, line 12: *One of the most abundant AGEs in vivo is N(6)-carboxymethyllysine (CML)^{5,6},...*

⁵ Hohmann C, et al. Detection of Free Advanced Glycation End Products in Vivo during Hemodialysis. *J Agric Food Chem* **65**, 930-937 (2017).

⁶ Baldensperger T, Eggen M, Kappen J, Winterhalter PR, Pfirrmann T, Glomb MA. Comprehensive analysis of posttranslational protein modifications in aging of subcellular compartments. *Sci Rep* **10**, 7596 (2020).

Comment 5: Chemical reaction and regulation of CML is not familiar for many readers. This referee would suggest to include a figure to show this, either in main figure or in the sup info.

We thank the reviewer for this suggestion. We have included a schematic representation of the reactions leading to CML formation in Supplementary Fig. 1A of the revised manuscript.

Supplementary Figure 1.

A. Simplified scheme of reaction cascade for the formation of carboxymethyllysine starting from glucose (left to right) or starting from glyoxal (right to left).

Comment 6: They are suggested to measure intracellular Glyoxal levels in aged vs young mice tissue.

As indicated in the reply above, we were able to successfully measure intracellular glyoxal concentrations in HUVEC. However, we decided not to perform the measurements on mouse tissues because we had concerns regarding potential confounding factors that could alter the outcome of the analysis. These include the presence of blood-derived glyoxal in freshly isolated organs as well as the impact of organ storage. To perform this type of analysis, we would have had to isolate organs from a new set of young and aged animals to perform the extraction from fresh tissues, which was not possible during the time given for revision.

Comment 7: In mouse tissue, did you identify other histone sites using IP-MS/MS? No other sites on histone H4 except for H4K92 is identified. Are there CML sites on N-terminal tails of histones?

We thank the reviewer for this comment. As indicated in the reply to Reviewer 3, comment 21 below, we have revised our data analysis strategy to, e.g., include variable modification of protein N-termini by carboxymethylation. In the new analysis, we identified additional sites of CML on other histone proteins beyond H4K92. We have included the list of these sites on histone proteins in the revised manuscript (Supplementary Fig. 2C). None of these sites occur on histone N-terminal tails. However, we would like to point out that histone N-terminal tails are typically not covered by standard LC-MS analysis based on tryptic digestion, because of the high content of basic residues. Therefore, we cannot exclude on the basis of our data that additional CML sites might occur in histone tails. We have clarified this point in the revised version of the manuscript.

Page 6, line 27: *Carboxymethylation appears to target proteins largely in an organ-specific manner (Fig. 2B). However, we identified a subset of proteins modified in all the organs tested. These included mainly mitochondrial and nuclear proteins, such as histones (Supplementary Fig. 2C,D).*

Page 22, line 15: *It remains to be investigated whether similar age-related changes of carboxymethylation can also occur on histone tails, which were not covered in our analysis due to experimental setup based on tryptic protein digestion.*

C

ID	Gene	CM site
P62806	HIST1H4A	Heart, Kidney, Liver (K92)
P43277	HIST1H1D	Heart (K64)
P10853	HIST1H2BF, HIST1H2BM, HIST1H2BB, HIST1H2BH, HIST2H2BB, HIST1H2BC, HIST1H2BK, HIST1H2BP	Heart (K47; K35)
P22752	HIST1H2AB, HIST3H2A, HIST1H2AF, HIST1H2AH, HIST1H2AK, H2AFJ	Heart, Kidney, Liver (K96)
P02301	H3F3C, H3F3A	Liver (K123)
Q64523	HIST2H2AC, HIST2H2AA1	Liver (K96)

D

Supplementary Figure 2.

C. Carboxymethylation (CM) sites of histones identified by CMLpeIP in all analyzed mouse organs. D. Proteins identified as carboxymethylated in all analyzed mouse organs (50 proteins from the Venn diagram in Fig. 2B) divided by cell compartment as defined by Gene Ontology annotation.

Comment 8: Figure 1G and 2B. How to explain the vast divergency of CML sites in different species and different organs? Are those organ-specific CML proteins also the most abundant proteins specifically in that organ? Because the modification is caused by chemical reaction, we assume that the modification is related with the protein abundance and glyoxal.

We thank the reviewer for this comment. Indeed, we observed a high degree of divergency in CML sites identified in different cell types and organs. This appears to relate to both protein abundance as well as turnover both in cultured cells as well as heart (Supplementary Fig.1E, 2E). The bias towards more abundant proteins might derive from the limit of detection of our approach. We have now estimated at least for one site (K58 on tubulin) that CML modification occurs at ~0.009 % of the total tubulin β -4B chain (Fig. 6G). We mentioned this potential bias in our discussion.

Page 21, line 24: *Interestingly though, we detected carboxymethylation of proteins that spanned four orders of magnitude of protein abundance and, conversely, we did not identify modification for all the most abundant proteins. Keeping in mind potential biases due to the limited sensitivity of our approach and specific protein sequences that are not amenable to enzymatic digestion, our data suggest the existence of a subset of proteins that are more prone to undergo carboxymethylation than others.*

However, we also detected a subset of modified proteins that were identified across different cell types and organs. These include for instance histones and several mitochondrial proteins. We speculate that the latter could derive from sub-cellular differences in local pH or concentration of metabolites that promote AGEs formation⁴⁵. We now indicate this possibility in the discussion of the revised manuscript.

Page 6, line 27: *Carboxymethylation appears to target proteins largely in an organ-specific manner (Fig. 2B). However, we identified a subset of proteins modified in all the organs tested. These included mainly mitochondrial and nuclear proteins, such as histones (Supplementary Fig. 2C,D).*

Page 21, line 32: *The modified proteins were enriched for mitochondrial proteins, histones, cytoskeletal proteins and enzymes involved in detoxification, which might, at least in part, derive from sub-cellular differences in local pH or concentration of metabolites that promote AGE formation⁴⁵.*

⁴⁵ Figlia G, Willnow P, Teleman AA. Metabolites Regulate Cell Signaling and Growth via Covalent Modification of Proteins. *Dev Cell* 54, 156-170 (2020).

Supplementary Figure 2.

D. Proteins identified as coxymethylated in all analyzed mouse organs (50 proteins from the Venn diagram in Fig. 2B) divided by cell compartment as defined by Gene Ontology annotation.

Comment 9: Figure 2B and 2C. The author should specify in the figure legend whether the CML sites are mapped in old mouse tissue, or young mouse tissue, or combined.

We have specified in the text and in the legend of Fig. 2 that the CML sites mapped derive from both young and old tissues.

Page 6, line 24: *By combining data from young and old mice, we identified 198, 71 and 105 unique CML sites and 81, 116, and 88 CM-Nterm sites in heart, kidney, and liver, respectively (Supplementary Table 2).*

Figure 2. The CM-modified proteome in mouse organs.

A. Workflow for the analysis performed on mouse organs from young (Y) and old (O) mice. Tandem Mass Tag (TMT) was used for monitoring protein abundance, CMLpepIP for CM site identification and PRM for quantification. **B.** Venn diagram of CM-modified proteins (left) and sites (right) identified in heart, kidney and liver from young and old mice combined. For each tissue, only unique entries were considered for the overlap. P-value was calculated by Fisher's Exact Test. **C.** Enrichment of Gene Ontology biological processes among CM sites identified from heart, kidney and liver. Enrichment was performed using the Cytoscape App ClueGO.

Comment 10: Figure 2D. Some PTM can be very dynamic and changes drastically from cell types to cell types. The statement that “CML modification compete with other PTMs” is overstated if the so-called overlapping PTMs are not detected under the same condition as CML. Therefore, this figure is not very helpful.

We thank the reviewer for pointing this out. We have now removed such statement from the introduction and result paragraphs to avoid any over-statement.

Comment 11: Figure 2E. This panel shows that old mice and young mice can be distinguished by global proteomics data of heart, liver and kidney. However, it is not very relevant to the scope of this manuscript since this manuscript focuses on the biological functions of CML. This reviewer suggests the authors to explore instead whether old and young mice can be differentiated by their CML profile

We thank the reviewer for this comment. In our manuscript, we used the analysis of CML-enriched fractions exclusively for identification of CML sites. We decided to refrain from using quantitative information from these data because of the multiple sample preparation steps that could introduce biases in label-free analysis. Rather, we opted for using targeted approaches for a subset of abundant CML sites that could be detected by PRM in total organ lysates. Although more focused, we believe that this type of analysis guarantees a more accurate estimation of the level of CML in young and old organs. To avoid any confusion for the readers, we have now moved the PCA plots referring to total proteome to the supplementary material (Supplementary Fig. 2F), and clearly state that these are used exclusively to assess the quality of the proteome samples obtained from young and old mice.

Page 7, line 4: *We used principal component analysis to confirm distinct proteome signatures between young and old mice for all the three organs (Supplementary Fig. 2F).*

Comment 12: Figure 2H and 2I. Among the 18 CML modifications that can be detected by PRM, how many of them are upregulated, downregulated, or not changed in aged organs?

We detected 16 out of those 18 CML sites, for which we had successfully developed PRM assays. In detail, 13 were detected in heart and kidney and 4 in liver using the elutions from CMLpepIP. As mentioned above, we opted to perform PRM on total organ lysates for the comparison of CML-modified peptides in young and old mice. In total organ lysates, we could quantify 4 CML sites in heart (Hist1h4, Hba, Slc25a4, Atp5f1c), and 2 in kidney and liver (Hist1h4 and Hba) in sufficient number of replicates. Of these, only Hist1h4-K92 increased significantly in all organs during aging.

Comment 13: Figure 2. The authors are suggested to validate age-related total CML increase by CML western blot of young and aged mice organs.

We thank the reviewer for this suggestion. We have included validation of total CML increase in aged tissue by immunoblot in heart, liver and kidney (Fig. 2H), as well as by LC-MS for kidney and liver (Supplementary Fig. 2H).

Figure 2. The CM-modified proteome in mouse organs.

H. Immunoblot (left) and densitometry-based quantification (right) for CML-modified proteins in mouse organs from young and old mice. n=6-8, mean + SD, ** p<0.01, *** p<0.001, **** p<0.0001, unpaired t-test.

H

Supplementary Figure 2.

H. Quantification of total CML levels during aging in mouse kidney and liver (Y=young, O=old). n=8, line shows the median, ** p<0.01, *** p<0.001, unpaired multiple t-test corrected via Holm-Šidák method.

Comment 14: Figure 3D and 3E. How would you explain the seemingly contradictory observation that while KEGG enrichment analysis showed a downregulation of ubiquitin-mediated proteolysis pathway (3D) after 2mM GO treatment, while most 26S proteasome subunits are in fact upregulated by 2mM GO (3E)?

We thank the reviewer for this comment and apologize for the lack of clarity in the original manuscript. We observed that indeed certain components of the UPS (i.e., ubiquitin ligases) are decreased in abundance following exposure to high concentration of glyoxal, while components of the proteasome are increased, presumably as a compensatory mechanism (Fig. R1). The same trend can be observed in MEF as well as HUVEC, although to a lesser extent (Fig. 3E, Supplementary Fig. 3C,D). We have now clarified this apparent discrepancy in the revised manuscript:

Page 9, line 33: *At this time point, the abundance of CM-modified proteins was significantly increased (Fig. 3C), while KEGG pathways related to the UPS, mainly including ubiquitin-conjugating enzymes, were decreased (Fig. 3D and Supplementary Fig. 3C). Conversely, we found a significant increase of components of the 26S proteasome induced by glyoxal treatment in both MEF and HUVEC (Fig. 3E left and Supplementary Fig. 3D). This was accompanied by an increased proteasomal activity in MEF after 0.5 mM glyoxal but not after 2 mM (Fig. 3E right).*

Figure R1.*

Box plot representing the effect of glyoxal treatment on the expression of ubiquitin ligases and proteasome in MEF and HUVEC compared to untreated controls. Data were filtered for proteins being characterized as subunits of the proteasome (blue), and ubiquitin ligases E1-E2 (red) and E3 (green).

* Figure R1 is shown in this point-by-point response letter but not included in the revised manuscript.

Comment 15: Figure 6B and 6E. GO induces tubulin filament stabilization, yet reduces tubulin filament stabilizing modification K40 acetylation. Could you please elaborate the possible mechanisms?

We thank the reviewer for this comment. We have now performed experiments with purified tubulin revealing that CML modification of tubulin by glyoxal leads to decreased polymerization *in vitro* (Fig. 6D,E). Based on these results, we are now reluctant in discussing tubulin filament stabilization by glyoxal as previously suggested from the intense tubulin filament staining. We rather speculate that CML modification of tubulin interferes with dynamic instability of microtubules by slowing down polymerization and, may be, depolymerization as well. In this case, the decreased K40 acetylation, i.e. a loss of microtubule protection, could add to the direct effect of glyoxal on tubulin dynamics.

We have changed our discussion accordingly and deleted the statement of an *increased stability of CML-modified microtubules*.

Page 16, line 17: *CML-modified tubulin showed more intensively stained filaments when compared to tubulin in control cells (Fig. 6B) and an increased resistance against the depolymerizing agent nocodazole (Fig. 6C). To understand whether CML modification of tubulin affected its polymerization behavior, we exposed purified tubulin to glyoxal. This triggered CML modification at sites that overlap with the ones identified in glyoxal-treated cells (Fig. 6D and Supplementary Fig. 7B) and led to a significantly lower polymerization in vitro (Fig. 6E). These data suggest changes of microtubule dynamics upon glyoxal treatment, which may contribute to the observed reduction of cell proliferation by impeding mitosis.*

Figure 6. Glyoxal (GO) affects post-translational modification and dynamics of tubulin.

D. Purified porcine tubulin incubated with GO for 30 min on ice was analyzed in immunoblots. n=5. **E.** Purified porcine tubulin was pre-incubated with GO (1 mM, 30 min, on ice) and polymerization induced by addition of cofactors at 37 °C was monitored in a fluorescence-based assay. Controls: taxol (3 μM) or nocodazole (20 μM). n=3-6, mean ± SEM. Statistical significance was analyzed using one-way repeated measurement ANOVA corrected via Holm-Šidák method for areas under the curves. * p<0.05 vs. control.

B Matched sites of tubulin glycation

ID Porcine	Sequence	ID Human	Gene	Protein name	CM site Human
Q2XVP4	DVNAAIATIK(1)TK(1)R	P68363	TUBA1B	Tubulin alpha-1B chain	K336; K338
Q2XVP4	FDLMYAK(1)R	P68363	TUBA1B	Tubulin alpha-1B chain	K401
Q2XVP4	GDVVPK(1)DVNAAIATIK(1)TK	P68363	TUBA1B	Tubulin alpha-1B chain	K336; K326
Q2XVP4	LDHK(1)FDLMYAK	P68363	TUBA1B	Tubulin alpha-1B chain	K394
Q2XVP4	LDHK(1)FDLMYAK(1)R	P68363	TUBA1B	Tubulin alpha-1B chain	K401; K394
Q2XVP4	LSVDYGK(1)K(1)SK	P68363	TUBA1B	Tubulin alpha-1B chain	K163; K164
Q767L7	ISVYYNEATGGK(1)YVPR	P07437	TUBB	Tubulin beta chain	K58
Q767L7	MAVTFIGNSTAIQELFK(1)R	P07437	TUBB	Tubulin beta chain	K379
A0A5G2R693	ISVYYNEASSHK(1)YVPR	Q13509	TUBB3	Tubulin beta-3 chain	K58
A0A287A275	INVYYNEATGGK(1)YVPR	P68371	TUBB4B	Tubulin beta-4B chain	K58
P02554	MSMK(1)EVDEQMLNVQNK	P68371	TUBB4B	Tubulin beta-4B chain	K324

Supplementary Figure 7.

B. Table of identified CML sites matched between purified porcine tubulin and endogenous tubulin in HUVEC.

Comment 16: Figure 3,4,5,6. Where applicable, validate the change of CML level of relevant proteins by PRM in different cell stages, during angiogenic sprouting, with or without glyoxal treatment. It is possible CML level changes for some but not all proteins during these processes. Then the site-specific quantification will provide a useful correlation information and suggest which proteins and which CML sites might be most relevant in these conditions.

We have designed PRM assays for tubulin K58CML modification and used them (i) to estimate the absolute levels of modification (Fig. 6G), and (ii) to measure relative CML modification changes across different glyoxal treatments (short term, high concentration, as in the original manuscript, and long-term, low concentration, new data included in the revised manuscript (Fig. 7G, H)). Since the current study does not address the role of specific CML modification in angiogenesis but rather used sprouting assays as a read out of proliferation-related endothelial cell function, we decided not to repeat PRM assays under angiogenic sprouting conditions. The latter would require a separate evaluation of tip and stalk cells and thus the establishment of a large-scale sprouting assay, which is beyond the scope of this study.

Figure 6. Glyoxal (GO) affects post-translational modification and dynamics of tubulin.

G. HUVEC were treated with GO (48 h). The absolute level of CML modification at K58 on TUBB4B was determined by PRM using the synthetic reference peptide _INVYYNEATGGK[CML]YVPR_. The bar plot shows the fraction of CML-modified tubulin at K58 relative to total TUBB4B. Total TUBB4B levels were determined using the synthetic peptide _INVYYNEATGGK_. n=3.

Figure 7. Long-term treatment with low doses of glyoxal (GO) impairs proliferation and affects tubulin.

HUVEC were treated with 1-100 μM GO for 14 d starting one day after seeding (day 1 - 15). **G.** Validation of CML-modified peptides by parallel reaction monitoring (PRM) using heavy spike-in peptides. **H.** Relative quantification of _INVYYNEATGGK[CML]YVPR_ (from TUBB4B) in GO-treated HUVEC. Quantification was performed on elution samples after CMLpepIP.

Reviewer #2

In the current manuscript by Sanzo et al, the authors developed specific antibody-based IP enrichment approach and used mass spectrometry to map the posttranslational modification of lysine residues called (ϵ -N) Carboxymethyl Lysine in proteome. CML modification is considered as one among major AGEs with relevance to metabolic disease and ageing associated disorders. Using isolated proteins from fibroblast and endothelial cells treated with glyoxal in vitro and from organs of young and aged mice authors identified several CML modification sites. Observed results suggest that glyoxal treatment at higher concentration promote CML modification and increased endogenous CML modification was detected in aged mice. The abundance of CML in older mice could be possibly due to impaired or less detoxification AGEs system including Glo1.

Comment 1: According to Fig 2F, there is no significant difference in Glo1 or Glo1 network proteins in heart muscle compared to kidney and liver. This suggest normal functioning of Glyoxalase system in heart muscle and chances for less CML modification in heart compared to liver and kidney. However, Fig 2H result suggest a significant CML modification in heart and kidney compared to liver, which seems contradicting. Do authors have any explanation for this?

We thank the reviewer for this comment. Indeed, GLO1 activity does not depend only on the abundance of the enzyme, but also on the levels of glutathione (GSH). The activity of GLO1 is proportional to GSH levels and decreases if cellular cytosolic GSH is diminished, e.g., upon oxidative stress. GSH levels are known to be reduced in old tissues, including heart, liver and kidney³³. We therefore speculate that the observed reduced GLO1 activity in old tissues is due to a combination of reduced GLO1 and GSH levels. We have more clearly discussed this important point in the revised manuscript:

Page 7, line 9: In addition, we found a general trend for a reduced abundance of proteins involved in the GLO1 network in old mice in all organs tested (Fig. 2F). This was accompanied by a decrease of GLO1 activity in organ lysates from old mice (Fig. 2G). Notably, the decrease of enzyme activity was more pronounced than the reduction in protein levels, suggesting that additional mechanisms, e.g., limited availability of cofactors such as glutathione³³ might contribute to the observed decrease in GLO1 activity in old organs.

Comment 2: In Fig 2I, did authors analyzed CML modification of K263 and K154 in liver and kidney? What is CML modification status.

We have indeed analyzed the CML modification of K263 on ADP/ATP translocase type 1 and of K154 on ATP synthase subunit γ also for kidney and liver. However, the quantification of these modified sites in total organ lysates was not possible for liver and kidney, likely due to sensitivity limit of our approach. We show the quantification of CML modification on K263 and K154 in the heart as one example for modifications not altered with aging pointing to the fact that age-related changes may show some site specificity.

Comment 3: There should be a space between SI unit and the value. E.g. 2 mM and not 2mM. Check all the figures and apply this throughout the manuscript.

We apologize for the inconsistency that we have now fixed in the revised manuscript.

Comment 4: Page 15 line 15 – Kindly note that “strongly structured tubulin” seems an inappropriate term. I highly recommend removing it, until evidence is provided by biophysical experiments demonstrating increased strength. Also, I can see the staining intensity of tubulin increased, which is not sufficient to conclude the strength of tubulin filaments (MT). From the images Fig. 6B, Does the space between the microtubule are reduced leading to aggregation? This leads to next question, Does the expression of microtubule-associated proteins (MAPs), e.g. tau, changes or CML alters interaction between MAP and MT changes? If possible, using the existing the data set, identify the changes in levels of MAPs.

We thank the reviewer for this valuable comment. We agree that our approach does not allow to evaluate strength or structure of microtubules after glyoxal treatment. We have now deleted this term and instead used *intensively stained filaments* to describe the observed changes in immunofluorescence. We have no evidence for a reduction of space between tubulin filaments. We found, however, an increased abundance of microtubule-associated proteins (MAPs)

³³ Hazelton GA, Lang CA. Glutathione contents of tissues in the aging mouse. *Biochem J* **188**, 25-30 (1980).

such as MAP1B and MAP4 and of microtubule regulators. Both, increased expression of MAPs and CML modification of tubulin may affect the interaction of MAPs with tubulin and contribute to altered microtubule dynamics.

We have included these data in the text of the results part, in the discussion and in Supplementary Fig. 6B.

Page 14, line 27: *In addition, an increased abundance of microtubule-associated proteins (MAPs) such as MAP1B and MAP4 and of microtubule regulators was observed (Supplementary Fig. 6B).*

Page 24, line 7: *In general, changes in the tubulin code may alter interactions with tubulin-binding proteins, an effect, which may additionally be influenced by the increased abundance of several MAPs as shown in our proteomics analysis.*

Supplementary Figure 6.

B. HUVEC were treated with GO at the indicated concentrations for 48 h. Box plot (left) and heatmap (right) show the effect of GO treatment of HUVEC on proteins assigned as “microtubule associated” from Gene Ontology Cellular Compartment annotation. n=4, * q<0.05.

Comment 5: Also, Page 20 (discussion about MT stability) – A) line 15 – The diameter of MT is 25 nm. Based on immunostaining, it is not possible to identify thinner MT filaments in control and thicker filaments in CML modified. May be MT filaments are more accessible for immunostaining after CML modification resulting in intense staining of MT filaments. Thus, without electron microscopy MT thickness cannot be concluded. B) Line 16 & 17 – “Accordingly, our data reveal an increased stability of CML-modified microtubule”, without examination of in vivo MT dynamics, I will suggest removing this conclusion.

We thank the reviewer for this important comment. We have removed the statements about microtubule thickness and stability. To understand whether CML modification may affect microtubule dynamics, we performed experiments with purified tubulin, which was exposed to glyoxal and thereafter subjected to a polymerization assay *in vitro*. We observed decreased polymerization of CML-modified tubulin and concluded that glyoxal may alter microtubule dynamics. The new data are included in the result paragraph describing the data on tubulin modification and function and in the discussion.

Page 16, line 17: *CML-modified tubulin showed more intensively stained filaments when compared to tubulin in control cells (Fig. 6B) and an increased resistance against the depolymerizing agent nocodazole (Fig. 6C). To understand whether CML modification of tubulin affected its polymerization behavior, we exposed purified tubulin to glyoxal. This triggered CML modification at sites that overlap with the ones identified in glyoxal-treated cells (Fig. 6D and Supplementary Fig. 7B) and led to a significantly lower polymerization in vitro (Fig. 6E). These data suggest changes of microtubule dynamics upon glyoxal treatment, which may contribute to the observed reduction of cell proliferation by impeding mitosis.*

Page 23, line 19: *Our data reveal CML modification as a previously unknown tubulin modification, which has an impact on microtubule dynamics. In cells, glyoxal treatment triggered the formation of intensively stained, nocodazole-resistant tubulin filaments, while carboxymethylation of purified tubulin decreased the polymerization rate in vitro. Of note, CML sites of purified tubulin overlapped with the sites identified in cells and mouse organs. Although it remains to be clarified whether microtubule dynamics is also affected in vivo, and whether and how this is related to the increased staining intensity of tubulin filaments, our data suggest that CML modification of tubulin may be functionally relevant in cells and involved in the antiproliferative effects of glyoxal.*

Figure 6. Glyoxal (GO) affects post-translational modification and dynamics of tubulin.

D. Purified porcine tubulin incubated with GO for 30 min on ice was analyzed in immunoblots. $n=5$. **E.** Purified porcine tubulin was pre-incubated with GO (1 mM, 30 min, on ice) and polymerization induced by addition of cofactors at 37 °C was monitored in a fluorescence-based assay. Controls: taxol (3 μ M) or nocodazole (20 μ M). $n=3-6$, mean \pm SEM. Statistical significance was analyzed using one-way repeated measurement ANOVA corrected via Holm-Šidák method for areas under the curves. * $p<0.05$ vs. control.

B Matched sites of tubulin glycation

ID Porcine	Sequence	ID Human	Gene	Protein name	CM site Human
Q2XVP4	DVNAAIATIK(1)TK(1)R	P68363	TUBA1B	Tubulin alpha-1B chain	K336; K338
Q2XVP4	FDLMYAK(1)R	P68363	TUBA1B	Tubulin alpha-1B chain	K401
Q2XVP4	GDVVPK(1)DVNAAIATIK(1)TK	P68363	TUBA1B	Tubulin alpha-1B chain	K336; K326
Q2XVP4	LDHK(1)FDLMYAK	P68363	TUBA1B	Tubulin alpha-1B chain	K394
Q2XVP4	LDHK(1)FDLMYAK(1)R	P68363	TUBA1B	Tubulin alpha-1B chain	K401; K394
Q2XVP4	LSVDYGK(1)K(1)SK	P68363	TUBA1B	Tubulin alpha-1B chain	K163; K164
Q767L7	ISVYYNEATGGK(1)YVPR	P07437	TUBB	Tubulin beta chain	K58
Q767L7	MAVTFIGNSTAIQELFK(1)R	P07437	TUBB	Tubulin beta chain	K379
A0A5G2R693	ISVYYNEASSHK(1)YVPR	Q13509	TUBB3	Tubulin beta-3 chain	K58
A0A287A275	INVYYNEATGGK(1)YVPR	P68371	TUBB4B	Tubulin beta-4B chain	K58
P02554	MSMK(1)EVDEQMLNVQNK	P68371	TUBB4B	Tubulin beta-4B chain	K324

Supplementary Figure 7.

B. Table of identified CML sites matched between purified porcine tubulin and endogenous tubulin in HUVEC.

Comment 6: Page 15, line 16, 17 – Nocodazole treatment cannot determine the status of microtubule dynamics. The currently data only suggest that MT are resistant to nocodazole, which cannot be interpreted as impaired MT dynamics. In vivo visualization of rate of MT growth and shrinkage upon treatment of GO (with tagged tubulin and EB1 protein) will be the conclusive experiment to demonstrate impaired MT dynamics. Modify the text and please only mention that MTs are resistant to nocodazole.

We thank the reviewer for pointing this out. We have now described that glyoxal-treated cells show nocodazole-resistant microtubules as suggested. To get hints for functional consequences of tubulin carboxymethylation, we performed experiments with purified tubulin and observed CML modification (Fig. 6D and Supplementary Fig. 7B) as well as decreased polymerization upon glyoxal treatment (Fig. 6E). From these experiments, we speculated that glyoxal may also alter microtubule dynamics *in vivo* although we have not performed experiments *in vivo* (see responses to comment 5 of this reviewer).

Page 16, line 17: CML-modified tubulin showed more intensively stained filaments when compared to tubulin in control cells (Fig. 6B) and an increased resistance against the depolymerizing agent nocodazole (Fig. 6C).

Page 23, line 19: Our data reveal CML modification as a previously unknown tubulin modification, which has an impact on microtubule dynamics. In cells, glyoxal treatment triggered the formation of intensively stained, nocodazole-resistant tubulin filaments, while carboxymethylation of purified tubulin decreased the polymerization rate *in vitro*.

Comment 7: The possible reasons for several fold increase in CML levels at treatment of GO 1 mM Vs 0.5 mM should be discussed.

We speculate that this might be due to exhaustion of the detoxification systems. Indeed, we observed decreased protein levels of detoxifying enzymes following treatment with the highest concentration of glyoxal, especially in MEF (Fig. R2). We have discussed this possibility in the revised manuscript.

Figure R2.*

Heatmap representing the abundance of protein part of the glyoxalase 1 (GLO1) network in MEF and HUVEC treated with glyoxal for 24 and 48 h respectively. Only proteins quantified in all conditions are displayed. The network was extracted from <https://string-db.org> using input GLO1. n=4.

Page 3, line 22: *We noted a sharp increase in CML levels at the respective highest concentrations of glyoxal, likely due to a saturation of cell detoxifying systems.*

Comment 8: What is the reason to use 2 mM of GO in Figure 3 and 1 mM of GO in Figure 4, as well as the treatment time (24 h Vs 48 h)? How 2 mM of GO affected the parameters that were measured in figure 4 (cell proliferation, mitochondrial mass, OCR, etc.)? Does 2 m GO highly toxic HUVEC cells but not to MEF?

We thank the reviewer for this comment. Indeed, MEF and HUVEC show a different sensitivity towards glyoxal as shown below in a cell survival assay performed using different concentrations of glyoxal (Fig. R3). In HUVEC, we started our glyoxal experiments with a treatment time of 48 h, but later, to better understand the mechanisms of glyoxal effects, we included lower incubations times (4 h, 24 h) as well.

Figure R3.*

Effect of glyoxal (GO) on cell numbers quantified by nuclear staining (Hoechst 33342). MEF or HUVEC were seeded into a 96-well plate with a density of 10,000 cells/well. After 24 h, cells were stimulated with glyoxal in the range of 0 - 8 mM for 24 h. After fixation with 4 % paraformaldehyde, cells were stained and counted on an automated ImageXpress Micro Confocal microscope (Molecular Devices, LLC). The quantification was performed on the acquired images using the MetaXpress Software (v6.2.3.733, Molecular Devices, LLC) and the quantification pipeline "DAPI - Cell Count". The generated data were exported to spreadsheet using the same software and further processed with GraphPad Prism (GraphPad Software, LLC, v8.3.0). For the IC50, glyoxal concentrations applied for this assay were transformed in Log10. The number of cells was normalized by averaging 8 technical replicates. Finally, IC50 was calculated using the Log10 of the concentration (inhibitor). Data from one experiment are shown (MEF (left) and HUVEC (right)). Red dots: 0.5 mM and 2 mM for MEF or 0.5 mM and 1 mM for HUVEC, respectively.

* Figure R2 and R3 are shown in this point-by-point response letter but not included in the revised manuscript.

Comment 9: In figure 3D KEGG pathways analysis for enriched proteins in glyoxal treated MEF cells displayed upregulation of proteins involved in oxidative phosphorylation, whereas glyoxal treatment reduced energy production and basal respiration via oxidative metabolism in HUVEC cells treated with glyoxal figure 4F. Did this suggest cell specific effect of glyoxal on mitochondrial bioenergetics?

We thank the reviewer for this comment. In both cell types, we found CML modification as well as changes in the abundance of mitochondrial proteins suggesting that mitochondrial energy production might be compromised in response to glyoxal. This was confirmed in HUVEC, although cellular ATP levels were not altered due to increased glycolysis and mitochondrial biogenesis. Endothelial cells generate energy mainly via aerobic glycolysis and may compensate decreased oxidative phosphorylation by increasing glucose flux. MEF may respond differently to inhibition of oxidative phosphorylation, i.e. by upregulating proteins involved in this process, which may confer some specificity to the response towards glyoxal.

We have now explored in more detail the effect of glyoxal on respiratory chain components (Fig. R4). We found that respiratory chain components are mainly increased in MEF upon 2 mM glyoxal treatment, however we have not further characterized the effect on energy metabolism in MEF.

Figure R4.*

Heatmap representing the abundance changes of protein involved in the category “Oxidative phosphorylation” induced in MEF by 0.5 mM and 2 mM GO after 24 h of treatment and in HUVEC by 0.5 mM and 1 mM after 48 h of treatment.

Comment 10: Fig S4 delete legend corresponding to γ H2AX (S139) immunofluorescence

We apologize for this negligence. We have now removed the respective sentence.

Comment 11: How long the cells were treated with glyoxal for senescence assays? After how many days of culturing the senescence assays (immunoblot analysis and SA-beta-gal assay) was performed? Include this detail in materials and methods.

We thank the reviewer for pointing this out. We observed premature senescence after treatment with 0.5 and 1 mM glyoxal for 48 h. This is now mentioned in the figure legend of Supplementary Fig. 6 and in the method section.

Page 51, line 2: *After glyoxal incubation of second-passage HUVEC for 48 h, cells were washed twice with 1 ml cold PBS and fixed with 1 ml of a 2 % formaldehyde/0.2 % glutaraldehyde solution for 3 min.*

Comment 12: Page 20, line 32 and 33; Does glyoxal treatment hamper the function of global PTM machinery to favor glycation? It seems that although K40 is not glycated its acetylation is reduced upon glyoxal treatment. Did authors find other sites with reduced PTM without CML modification?

We thank the reviewer for this comment. Unfortunately, we are not able to exclude the CML modification of tubulin K40 because the protein sequence surrounding this site lacks additional lysine or arginine residues. Therefore, in case of CML modification, the tryptic sites at K40 become resistant to proteolysis and the resulting peptide too long (>30 amino acids) to be detectable in our LC-MS workflow (Fig. R5A). In general, we did not observe a global alteration of the acetylome induced by glyoxal, as assessed by principal component analysis (PCA) (Fig. R5B). We therefore speculate that the potential interference between CML and other PTMs might be restricted to specific sites, rather than being a global effect.

* Figure R4 is shown in this point-by-point response letter but not included in the revised manuscript.

Figure R5.*

A. Primary sequence of tubulin alpha-1B chain with highlighted lysine and arginine residues cut by trypsin in yellow. In red, lysine 40 is highlighted. **B.** Principal component analysis of acetylated proteome data from HUVEC treated with 0, 0.5 mM and 1 mM glyoxal (GO) (n = 3). The smaller dots represent individual samples and the larger dots the centroids of each concentration-matched group. The percentage of variance explained by the first two principal components (PC) axes is reported in the axis titles.

Comment 12: Page 6 line 17, 18 authors claim that CML targets protein in organ specific manner.

We assume that the reviewer is referring to the overlap of CML sites identified in different organs. The new analysis of our dataset, which now also includes N-terminal modification of proteins, displayed higher overlap between organs, especially in terms of modified proteins (Fig. 2B). Therefore, we have revised our manuscript to highlight both common (Supplementary Fig. 2D), as well as organ-specific targets of protein carboxymethylation (Fig. 2C).

Page 6 lines 27: *Carboxymethylation appears to target proteins largely in an organ-specific manner (Fig. 2B). However, we identified a subset of proteins modified in all the organs tested. These included mainly mitochondrial and nuclear proteins, such as histones (Supplementary Fig. 2C,D). The affected proteins participate in biological processes related to oxidative phosphorylation in all organs, myofibril assembly in the heart, and detoxification in the liver (Fig. 2C).*

* Figure R5 is shown in this point-by-point response letter but not included in the revised manuscript.

Reviewer #3

Sanzo et al. generated CML modification maps using different cells and mouse tissues. CML is a widespread PTM, but its functions remain largely unknown. The result presented by this paper is probably the first large-scale dataset for CML, hence is a valuable resource for relevant studies. The experiments were well designed and performed, and the manuscript was written logically and clearly. I would recommend to accept this paper for publication after the authors address the following concerns.

We thank the reviewer for the positive comments.

Comment 1: The greatest drawback of this study is that the enrichment efficiency of the commercial CML antibody is bad. In many cases, only less than 1% of all captured peptides were CML modified. The low specificity strongly hampers the impact of the result. The immunogen is very critical in developing such antibodies. If the immunogen is a single CML modified peptide or protein, this antibody may not be suitable for enrichment purpose in large-scale studies. The authors should find out the immunogen information and state it in the revised manuscript. More, if the authors continue to be interested in this topic, I strongly recommend the authors to develop their own antibody for enrichment in the future studies.

We thank the reviewer for this comment. We fully agree that our strategy provides only a partial enrichment of CML modified peptides, likely due to the low abundance of the modified peptides. The antibody that we used was developed using carboxymethylated KLH as immunogen, and affinity purified with CML-agarose (https://www.immunechem.com/?app=product&act=look&type_id=8&id=125).

We have included the information about how the antibody was developed in the method session.

Page 27, line 12: The CML antibody (CN1040) used for enrichment was developed using carboxymethylated KLH as immunogen and affinity purified with CML-agarose. This antibody and the acetyl lysine antibody (agarose conjugate) were purchased from ImmuneChem Pharmaceuticals Inc. (Burnaby, Canada).

During the phase of method developed, we have compared other commercially available antibodies against CML and the one we chose provided the highest relative enrichment at the peptide level. We believe this is the currently commercially available best option, however future improvements might indeed require the development of new reagents. We have now included a statement regarding this point in the discussion paragraph of the revised manuscript.

Page 24, line 30: The presented strategy has limitations of sensitivity as indicated by the low percentage (1 - 4 %) of PSM deriving from CML-modified peptides even in samples from glyoxal-treated cells. It can be envisaged, however, that future development of antibodies with higher specificity for CML-modified peptides will enable the identification of carboxymethylation sites that occur at lower abundance.

Comment 2: The biological consequence reported by this paper was largely associated with glyoxal treatment. However, whether this is directly caused by the CML modification was unknown. Besides induction of CML, many other known and unknown effects exist for glyoxal. I would suggest the authors to elucidate at least one CML-directed mechanism. For example, the authors can incubate purified proteasome with glyoxal *in vitro*, and then examine the CML sites on all proteasome subunits, and corresponding proteasome activity. This simple system will allow to exclude any other indirect effects caused by glyoxal. A similar *in vitro* experiment can also be considered for tubulin.

We thank the reviewer for this constructive comment that we have addressed with two sets of experiments.

1. We have obtained purified tubulin, incubated it with 1 or 10 mM of glyoxal for 30 min on ice and subsequently assessed polymerization *in vitro*. We confirmed CML modification by immunoblot and mass spectrometry. Our data show that CML-modification of tubulin clearly decreased polymerization of tubulin induced by addition of cofactors and incubation at 37 °C thus linking glyoxal-induced carboxymethylation to functional alteration. These data are described in the text and in Fig. 6D, E and Supplementary Fig. 7B.

*Page 16, line 19: To understand whether CML modification of tubulin affected its polymerization behavior, we exposed purified tubulin to glyoxal. This triggered CML modification at sites that overlap with the ones identified in glyoxal-treated cells (Fig. 6D and Supplementary Fig. 7B) and led to a significantly lower polymerization *in vitro* (Fig. 6E). These data suggest changes of microtubule dynamics upon glyoxal treatment, which may contribute to the observed reduction of cell proliferation by impeding mitosis.*

Page 23, line 21: *In cells, glyoxal treatment triggered the formation of intensively stained, nocodazole-resistant tubulin filaments, while carboxymethylation of purified tubulin decreased the polymerization rate in vitro. Of note, CML sites of purified tubulin overlapped with the sites identified in cells and mouse organs. Although it remains to be clarified whether microtubule dynamics is also affected in vivo, and whether and how this is related to the increased staining intensity of tubulin filaments, our data suggest that CML modification of tubulin may be functionally relevant in cells and involved in the antiproliferative effects of glyoxal.*

Figure 6. Glyoxal (GO) affects post-translational modification and dynamics of tubulin.
D. Purified porcine tubulin incubated with GO for 30 min on ice was analyzed in immunoblots. n=5. **E.** Purified porcine tubulin was pre-incubated with GO (1 mM, 30 min, on ice) and polymerization induced by addition of cofactors at 37 °C was monitored in a fluorescence-based assay. Controls: taxol (3 μM) or nocodazole (20 μM). n=3-6, mean ± SEM. Statistical significance was analyzed using one-way repeated measurement ANOVA corrected via Holm-Šidák method for areas under the curves. * p<0.05 vs. control.

B Matched sites of tubulin glycation

ID Porcine	Sequence	ID Human	Gene	Protein name	CM site Human
Q2XVP4	DVNAAIATIK(1)TK(1)R	P68363	TUBA1B	Tubulin alpha-1B chain	K336; K338
Q2XVP4	FDLMYAK(1)R	P68363	TUBA1B	Tubulin alpha-1B chain	K401
Q2XVP4	GDVVPK(1)DVNAAIATIK(1)TK	P68363	TUBA1B	Tubulin alpha-1B chain	K336; K326
Q2XVP4	LDHK(1)FDLMYAK	P68363	TUBA1B	Tubulin alpha-1B chain	K394
Q2XVP4	LDHK(1)FDLMYAK(1)R	P68363	TUBA1B	Tubulin alpha-1B chain	K401; K394
Q2XVP4	LSVDY GK(1)K(1)SK	P68363	TUBA1B	Tubulin alpha-1B chain	K163; K164
Q767L7	ISVYYNEATG GK(1)YVPR	P07437	TUBB	Tubulin beta chain	K58
Q767L7	MAVTFIGNSTAIQELFK(1)R	P07437	TUBB	Tubulin beta chain	K379
A0A5G2R693	ISVYYNEASSHK(1)YVPR	Q13509	TUBB3	Tubulin beta-3 chain	K58
A0A287A275	INVYYNEATG GK(1)YVPR	P68371	TUBB4B	Tubulin beta-4B chain	K58
P02554	MSMK(1)EVDEQMLNVQNK	P68371	TUBB4B	Tubulin beta-4B chain	K324

Supplementary Figure 7.

B. Table of identified CML sites matched between purified porcine tubulin and endogenous tubulin in HUVEC.

- We have obtained purified 26S proteasome, incubated it with 1mM glyoxal for different amount of time, and assessed chymotrypsin-like activity in native gel activity assay. We confirmed CML modification by immunoblot. Surprisingly, we found that, under these experimental conditions, CML modification of proteasome is not sufficient to alter its proteolytic activity, suggesting that the effects that we observed in cells might be mediated by indirect mechanisms, e.g., modification of regulators of proteasome activity or excessive accumulation of glycated substrates. We have included this new data in Fig. 3 and the discussion paragraph.

Page 10, line 7: *To test whether carboxymethylation directly influences proteasome activity, we treated purified 26S proteasome with glyoxal, and assessed chymotrypsin-like activity, the central proteasomal peptidase activity³⁷, using a native gel assay. We confirmed induction of CML modification by immunoblot (Fig. 3H middle). However, we were not able to detect any change of proteasome activity induced by glyoxal treatment in vitro under the applied experimental conditions (Fig. 3H right).*

³⁷ Kisselev AF, Callard A, Goldberg AL. Importance of the different proteolytic sites of the proteasome and the efficacy of inhibitors varies with the protein substrate. *J Biol Chem* **281**, 8582-8590 (2006).

Page 22, line 24: Here, we provide evidence for an interference of glyoxal with the thermal stability of the 19S proteasome. A similar phenomenon was recently described in response to ATP deprivation in cells⁵³. However, in contrast with a previous report¹⁹, glyoxal was not sufficient to directly inhibit the chymotrypsin-like activity of the 26S proteasome *in vitro* in our study suggesting indirect mechanisms, e.g., modification of regulators of proteasome activity or excessive accumulation of glycated substrates. Indeed, we identified several enzymes involved in the ubiquitin cycle, including ubiquitin itself, to be direct targets of carboxymethylation both in cells and tissues. Given that proteasome activity is known to decrease during aging^{42,54,55,56}, the elevated levels of AGEs in old tissues might contribute to proteostasis impairment via alteration of the UPS.

Figure 3. Alteration of proteostasis induced by glyoxal (GO) in MEF.

H. Left, proteasome activity assay using native gel electrophoresis (top) after incubation of purified proteasome with GO (1 mM, 0, 1, 2 and 4 h, 37 °C). Immunoblot for CML (middle) and proteasome subunits α 1-7 (bottom). **Middle**, quantification of CML-modification. **Right**, proteasome chymotrypsin-like activity (CT-L) normalized to α 1-7 abundance. Positive control: purified proteasome incubated with MG132 (100 μ M, 4 h, 37 °C). n=4, mean + SD, * $p < 0.05$, one-way ANOVA.

Comment 3: Is there any sequence motif featured by CML modification? The authors claimed that CML competes with other lysine modifications. I'm curious if the CML and Kac peptides share a common sequence motif. Moreover, were there any higher structural features observed on these CML localizations. As the CML is a non-enzymatic modification, I would think lysine residues at protein surface may be prone to be modified.

We thank the reviewer for this suggestion. We have searched for enriched motifs using CML-modified peptides employing different strategies (15-mer peptide centered around the CML positions analysed by MEME^I, and alignment of 15-mer peptides with clustalomega^{II} but we were not able to retrieve significantly enriched motifs. We believe that this could be expected given the non-enzymatic nature of CML modification. However, as suggested by the reviewer, we found CML sites to be often exposed on the protein surface. We performed accessibility and secondary structure element analysis using DSSP^{III,IV}, considering only XRAY or NMR structures available in Protein Data Bank (PDB). For accessibility, we used the relative surface area (RSA) score. Using a widely accepted threshold of 25 % RSA to define

⁵³ Sridharan S, *et al.* Proteome-wide solubility and thermal stability profiling reveals distinct regulatory roles for ATP. *Nat Commun* **10**, 1155 (2019).

¹⁹ Queisser MA, *et al.* Hyperglycemia impairs proteasome function by methylglyoxal. *Diabetes* **59**, 670-678 (2010).

⁴² Hipp MS, Kasturi P, Hartl FU. The proteostasis network and its decline in ageing. *Nat Rev Mol Cell Biol* **20**, 421-435 (2019).

⁵⁴ Friguet B, Bulteau AL, Chondrogianni N, Conconi M, Petropoulos I. Protein degradation by the proteasome and its implications in aging. *Ann N Y Acad Sci* **908**, 143-154 (2000).

⁵⁵ Kelmer Sacramento E, *et al.* Reduced proteasome activity in the aging brain results in ribosome stoichiometry loss and aggregation. *Mol Syst Biol* **16**, e9596 (2020).

⁵⁶ Saez I, Vilchez D. The Mechanistic Links Between Proteasome Activity, Aging and Age-related Diseases. *Curr Genomics* **15**, 38-51 (2014).

^I Bailey TL, *et al.* MEME SUITE: tools for motif discovery and searching. *Nucleic Acids Res* **37**, W202-208 (2009). [not cited in the manuscript]

^{II} Sievers F, *et al.* Fast, scalable generation of high-quality protein multiple sequence alignments using Clustal Omega. *Mol Syst Biol* **7**, 539 (2011). [not cited in the manuscript]

^{III} Joosten RP, *et al.* A series of PDB related databases for everyday needs. *Nucleic Acids Res* **39**, D411-419 (2011). [not cited in the manuscript]

^{IV} Kabsch W, Sander C. Dictionary of protein secondary structure: pattern recognition of hydrogen-bonded and geometrical features. *Biopolymers* **22**, 2577-2637 (1983). [not cited in the manuscript]

residues as exposed, we found in each dataset (MEF, HUVEC and organs) that the majority of the identified CML sites are surface exposed. Additionally, analysis of secondary structures indicated that CML sites are often located in alpha-helices. We have included the results for organs in Fig. 2D of the revised manuscript, and we report here for the reviewer the complementary results for MEF and HUVEC (Fig. R6). We have also included the following statement in the revised result paragraph:

Page 6, line 34: *Analysis of CML sites on protein structures revealed that most of the modified residues are surface exposed and often located in alpha helices (Fig. 2D). Consistent with the non-enzymatic nature of these modifications, we did not find any sequence motif shared among CML-modified peptides. Finally, since CML modification occurs on lysines and these residues are targets for other posttranslational modifications (PTM), we assessed the overlap of CML sites and other known PTM. We found that approximately 80 % of the identified CML sites (260 out of 313) occur on residues that are known to be modified by other PTM, primarily ubiquitination and acetylation (Fig. 2E).*

Figure 2. The CM-modified proteome in mouse organs.

D. Left, percentage of surface exposed CML sites. A 25 % threshold for relative surface area (RSA) score was used to define sites as surface exposed. **Right**, percentage of CML sites located in different protein secondary structures.

Figure R6.*

A-B. Percentage of modified sites over the 25 % threshold for relative surface area (RSA) score (left) in samples from MEF (A) and HUVEC (B). Structural analysis of mapping modified sites (expressed as percentage) in different secondary structures in samples from MEF (A) and HUVEC (B).

Comment 4: Figure 2FG. GLO1 had a very mild decrease at expression level in aged mouse hearts, but had an unproportioned decrease in its activity. How should we understand the relationship between the protein abundance and activity? Similarly, Kidney had a greater decrease in abundance, but less effect on activity.

We thank the reviewer for this comment. Indeed, GLO1 activity does not depend only on the abundance of the enzyme, but also on the levels of glutathione (GSH). The activity of GLO1 is proportional to GSH levels and its activity decreases if cellular cytosolic GSH is diminished, e.g., upon oxidative stress. GSH levels are known to be reduced in old tissues,

* Figure R6 is shown in this point-by-point response letter but not included in the revised manuscript.

including the heart³⁹. We therefore speculate that the observed reduced GLO1 activity in old tissues is due to a combination of reduced GLO1 and GSH levels. We have more clearly discussed this important point in the revised manuscript:

Page 7, line 9: *In addition, we found a general trend for a reduced abundance of proteins involved in the GLO1 network in old mice in all organs tested (Fig. 2F). This was accompanied by a decrease of GLO1 activity in organ lysates from old mice (Fig. 2G). Notably, the decrease of enzyme activity was more pronounced than the reduction in protein levels, suggesting that additional mechanisms, e.g., limited availability of cofactors such as glutathione³³, might contribute to the observed decrease in GLO1 activity in old organs.*

Comment 5: Figure 3DE. Proteasome is the major component of UPS. Why did fig 3D show “ubiquitin-mediated proteolysis” was downregulated, but proteasome was increased in 3E upon glyoxal treatment?

We thank the reviewer for this comment and apologize for the lack of clarity in the original manuscript. We observed that indeed certain components of the UPS (i.e., ubiquitin ligases) are decreased in abundance following exposure to high concentration of glyoxal, while component of the proteasome are increased, presumably as a compensatory mechanism. The same trend can be observed in MEF as well as HUVEC, although to a lesser extent (Fig. R1). We have now clarified this apparent discrepancy in the revised manuscript.

Page 9, line 33: *At this time point, the abundance of CM-modified proteins was significantly increased (Fig. 3C), while KEGG pathways related to the UPS, mainly including ubiquitin-conjugating enzymes, were decreased (Fig. 3D and Supplementary Fig. 3C). Conversely, we found a significant increase of components of the 26S proteasome induced by glyoxal treatment in both MEF and HUVEC (Fig. 3E left and Supplementary Fig. 3D). This was accompanied by an increased proteasomal activity in MEF after 0.5 mM glyoxal but not after 2 mM (Fig. 3E right).*

Figure R1.*

Box plot representing the effect of glyoxal treatment on the expression of ubiquitin ligases and proteasome in MEF and HUVEC compared to untreated controls. Data were filtered for proteins being characterized as subunits of the proteasome (blue), and ubiquitin ligases: E1-E2 (red) and E3 (green).

Comment 6: The melting temperatures of proteasome subunits actually only changed a little with glyoxal treatment. Whether this could lead to an impairment on proteasome activity is uncertain. Do all of them (fig 3G) contain a CML modification? The melting curves of these proteins should be added in the supplementary files. More, the TPP results for non-proteasome proteins with CML modification should be added. Do they also have a decreased melting temperature with glyoxal?

³³ Hazelton GA, Lang CA. Glutathione contents of tissues in the aging mouse. *Biochem J* **188**, 25-30 (1980).

* Figure R1 is shown in this point-by-point response letter but not included in the revised manuscript.

We thank the reviewer for this suggestion. We have now included in Supplementary Fig. 4A all the melting curve plots for all the proteasome subunits displayed in Fig. 3G of the original manuscript and the list of CML-modified proteasome subunits (PSMC4), suggesting that the change of thermal stability might be an indirect effect, e.g., due to changes in proteasome assembly. Along these lines, we would also like to point out that all the affected subunits belong to the 19S regulatory particle. Interestingly, a similar effect (reduced thermal stability of 19S but not 20S) was also reported following ATP depletion⁵³. We have discussed these results in more detail in the revised manuscript:

Page 22, line 24: Here, we provide evidence for an interference of glyoxal with the thermal stability of the 19S proteasome. A similar phenomenon was recently described in response to ATP deprivation in cells⁵³. However, in contrast with a previous report¹⁹, glyoxal was not sufficient to directly inhibit the chymotrypsin-like activity of the 26S proteasome *in vitro* in our study suggesting indirect mechanisms, e.g., modification of regulators of proteasome activity or excessive accumulation of glycated substrates. Indeed, we identified several enzymes involved in the ubiquitin cycle, including ubiquitin itself, to be direct targets of carboxymethylation both in cells and tissues.

A

Supplementary Figure 4.

A. Melting curves of proteasome subunits represented in Fig. 3G. For each melting curve, the curve slope, the plateau and the R² are reported. Curves are colored based on the glyoxal (GO) treatment for 24 h: 0 mM (Ctrl,

⁵³ Sridharan S, *et al.* Proteome-wide solubility and thermal stability profiling reveals distinct regulatory roles for ATP. *Nat Commun* **10**, 1155 (2019).

¹⁹ Queisser MA, *et al.* Hyperglycemia impairs proteasome function by methylglyoxal. *Diabetes* **59**, 670-678 (2010).

green), 0.5 mM (purple) and 2 mM (yellow). The cross, in each plot, shows at which temperature the protein reaches its 50 % of fraction non-denaturated.

Comment 7: Figure S3E is inappropriate just because only a small subgroup of the total proteome could be modified by CML.

We apologize with the reviewer for the inappropriate figure representation in the original manuscript. We re-run the analysis grouping the melting temperature by condition and comparing the melting temperature distributions of not modified and modified proteins. The list of modified proteins was obtained from the CMLpepIP analysis done on MEF. We have included this new figure as Supplementary Fig. 4B,C.

Supplementary Figure 4.

B. Effect of GO on global protein thermal stability in MEF treated with GO for 24 h. Melting temperatures for 4367 (Ctrl), 4174 (GO 0.5 mM) and 4646 (GO 2 mM) protein groups were estimated using thermal proteome profiling (see Methods for details). **C.** Density plot representing the distribution of melting temperature of the entire dataset grouped by GO treatment (Ctrl, 0.5 mM and 2 mM). The distribution of non-modified proteins is reported in yellow, while the distribution of CM-modified proteins is shown in green. The list of modified proteins was obtained from CMLpepIP experiments performed in MEF. Dashed lines represent the group mean.

Comment 8: Can the authors estimate modification degree (stoichiometry) of CML? Taken K40 on tubulin as an example, how much was the acetylation and CML? This is again relevant to the competition mechanisms between CML and other lysine modification. Lysine acetylation frequently occurs at a very low level (<1%) on non-histone proteins, how CML competes with Kac is an unanswered question. Why doesn't CML just choose to modify the remaining free lysine? As far as this reviewer knows, competition is not a dominant mechanism in regulating different lysine modifications.

We thank the reviewer for this important comment. Unfortunately, we were not able to detect CML modification on K40 for technical reasons (see reviewer 2, comment 12), but we addressed this question by studying modifications on K58. We designed PRM assays for K58CML on TUBB4B and used spike-in reference peptides to estimate its absolute abundance in HUVEC treated with different concentration of glyoxal. Using this assay, we were able to quantify K58CML from not-enriched peptide samples, and estimate the fraction of modified tubulin β -4B using a non-modified proteotypic peptide for normalization. Across three independent samples (cells from different donors), we were able to estimate that K58CML occurs on tubulin β -4B at a stoichiometry of ~0.009 % following treatment with 1 mM glyoxal (Fig. 6G). We were unfortunately not able to detect K58CML neither acetylated K58 from untreated cells using PRM.

In a new set of experiments included in the revised version of the manuscript, we also explored the impact of prolonged exposure to lower concentrations glyoxal (0.1 mM for 14 days). Using the same analytical setup, we quantified the relative levels of K58CML following different treatment with glyoxal (in this case using CMLpepIP samples), and estimated a difference of about 60x between short (1 mM, 24 h) and prolonged (0.1 mM, 14 days) glyoxal treatment (Fig. 7G,H).

Although these experiments provide an estimate of the absolute levels of CML modification on tubulin, they do not prove a direct competition mechanism with acetylation. We have therefore decided to phrase the relevant passages more carefully in the revised manuscript to avoid any over-statement:

Page 16, line 30: CML modification of tubulin may alter microtubule dynamics directly or indirectly by interfering with other lysine modifications. Since tubulin is known to be regulated by acetylation of lysines³⁹ (Fig. 6A), we hypothesized that this may be influenced by CML modification. We were able to demonstrate opposite changes in CML modification and acetylation at lysine 58 (K58) of the tubulin β -4B chain in glyoxal-treated cells. While glyoxal induced CML modification of K58 at a stoichiometry of $\sim 0.009\%$ (Fig. 6G), K58 acetylation was concomitantly reduced (Fig. 6H). These data suggest that CML modification of tubulin is likely to have an impact on acetylation and may thereby affect its interactions with microtubule-binding proteins and regulatory enzymes.

Page 19, line 35: Using PRM assays on CMLpepIP samples, we were able to detect CML modification at K58 after long-treatment of cells with $100\ \mu\text{M}$ glyoxal, even though to a lower extent than after short-treatment with $1\ \text{mM}$ glyoxal (Fig 7G,H).

Figure 6. Glyoxal (GO) affects post-translational modification and dynamics of tubulin.

G. HUVEC were treated with GO (48 h). The absolute level of CML modification at K58 on TUBB4B was determined by PRM using the synthetic reference peptide `_INVYYNEATGGK[CML]YVPR`. The bar plot shows the fraction of CML-modified tubulin at K58 relative to total TUBB4B. Total TUBB4B levels were determined using the synthetic peptide `_INVYYNEATGGK`. n=3. HUVEC were treated with GO (24 h) and relative K58 acetylation on TUBB4B was quantified by label-free mass spectrometry using the endogenous peptide `_INVYYNEATGGK[Ac]YVPR`. n=3.

Figure 7. Long-term treatment with low doses of glyoxal (GO) impairs proliferation and affects tubulin.

HUVEC were treated with 1-100 μM GO for 14 d starting one day after seeding (day 1 - 15). **G.** Validation of CML-modified peptides by parallel reaction monitoring (PRM) using heavy spike-in peptides. **H.** Relative quantification of `_INVYYNEATGGK[CML]YVPR_` (from TUBB4B) in GO-treated HUVEC. Quantification was performed on elution samples after CMLpepIP.

Comment 9: Page 4, line 5. Were there any criteria in choosing peptides for PRM analysis? I wonder besides the two successful examples shown in figure 1E and F, how about other 14 validated peptides? Could the authors include those data in the supplementary files?

³⁹ Sadoul K, Khochbin S. The growing landscape of tubulin acetylation: lysine 40 and many more. *Biochem J* **473**, 1859-1868 (2016).

The peptides were selected among the most confident ones identified in CMLpepIP elutions. As indicated, we could validate 16 of them using PRM on CMLpepIP elutions. However, it was not possible to reproducibly detect all of them across multiple replicates in not-enriched samples used for quantification. In Fig. 1F,G (revised version), we opted for quantification in non-enriched samples to avoid potential technical biases due to variation of enrichment efficiency across replicates. We have included PRM traces for all the validated peptides in the Supplementary Fig. S2A of the revised manuscript.

Comment 10: The authors claimed that CML is likely to modify high abundant proteins, which might be an artifact. The antibody used in this study has low specificity and sensitivity, thus it just could not enrich those CML modified peptides from low abundance proteins.

We fully agree with the reviewer that the observed bias for high abundant protein could derive from the limit of detection of our approach. We have indicated this possibility in the discussion of our manuscript:

Page 21, line 20: *We found that carboxymethylation occurs more often in high abundant proteins that display slower turnover. This corresponds to previous studies showing that global AGEs accumulate particularly in tissues characterized by slow protein turnover, such as crystallin lens, cartilage and skin^{29,30,31}. Interestingly though, we detected carboxymethylation of proteins that spanned four orders of magnitude of protein abundance and, conversely, we did not identify modification for all the most abundant proteins. Keeping in mind potential biases due to the limited sensitivity of our approach and specific protein sequences that are not amenable to enzymatic digestion, our data suggest the existence of a subset of proteins that are more prone to undergo carboxymethylation than others*

However, we would like to point out that we have detected CML sites across a broad range of protein abundances (see rank plots in Supplementary Fig. 1E), and that, particularly in heart, turnover, rather than abundance *per se*, appears to discriminate CML-modified proteins (Supplementary Fig. 2E). In addition, the analysis of commonly modified proteins across tissues point to specific cellular compartments (mitochondrion) and molecular functions (detoxification enzymes) to be associated with CM modification. We speculate that exposure to higher local concentrations of metabolites, e.g., dicarbonyl compounds, could make these proteins more susceptible to modification. We have more clearly spelled out this hypothesis in the revised manuscript:

Page 21, line 32: *The modified proteins were enriched for mitochondrial proteins, histones, cytoskeletal proteins and enzymes involved in detoxification, which might, at least in part, derive from sub-cellular differences in local pH or concentration of metabolites that promote AGE formation⁴⁵.*

Comment 11: Page 6 Line 12. The observed lower CML level in brain is just against the result that CML tends to affect slow turnover proteins. Brain contains many more long-lived proteins than liver does. Could the authors provide any explanation?

We thank the reviewer for this comment. Indeed, proteins in the brain tend to have a slower turnover than in other organs, however the total CML levels appears to be lower. A potential explanation for this apparent inconsistency is that turnover is only one of the determinants of CML modification. We did not observe modification of all the proteins showing slow turnover. However, we identified classes of proteins with specific localization or molecular functions, e.g., mitochondrial proteins, that tend to be modified more often than others. As mentioned in the point above, we speculate that differences in local concentration of metabolites or pH could favor AGE formation. We hypothesize that similar differences can also occur between organs/cell types and, thereby, explain variation of CML levels independently of protein turnover. However, further studies are required to verify this hypothesis.

²⁹ Gkogkolou P, Bohm M. Advanced glycation end products: Key players in skin aging? *Dermatoendocrinol* **4**, 259-270 (2012).

³⁰ Smuda M, *et al.* Comprehensive analysis of maillard protein modifications in human lenses: effect of age and cataract. *Biochemistry* **54**, 2500-2507 (2015).

³¹ Verzijl N, *et al.* Age-related accumulation of Maillard reaction products in human articular cartilage collagen. *Biochem J* **350 Pt 2**, 381-387 (2000).

⁴⁵ Figlia G, Willnow P, Teleman AA. Metabolites Regulate Cell Signaling and Growth via Covalent Modification of Proteins. *Dev Cell* **54**, 156-170 (2020).

Comment 12: It's very interesting to know CML tends to occur on slow turnover proteins. However, as CML is a non-enzymatic modification, how was this preference achieved?

We speculate that the slow turnover enables more time for the modification reactions to occur.

Comment 13: The authors should carefully check if iodoacetamide-induced carbamidomethylation on lysine, though it mostly happens on cysteine, interferes the CML identification. The mass gains of these two modifications are 57 and 58, respectively. A mis-picking of monoisotopic peak may lead to misidentifications of CML. More, the exact delta masses used for modifications in database search should be added in the method.

We thank the reviewer for these important suggestions. We have designed our workflow to try to minimize artifacts induced by iodoacetamide (IAA). For instance, we perform reduction and alkylation prior to protein precipitation to ensure that IAA is not present during the long incubations for protein digestion. However, to exclude any potential artifacts, we have performed an additional control experiment where we analyzed the frequency of CML modified peptides in paired samples treated or not treated with IAA. These analyses showed no impact of IAA on the fraction of CML peptides detected (Fig. R7). We therefore exclude that IAA treatment in our workflow interferes with CML identification. We have also included a table in the method part of the revised manuscript stating the exact delta mass used (page 3, line 1).

Figure R7.*

Percentage of peptide spectrum matches (PSM) containing CML modification among control and 2 mM glyoxal. The sample preparation was carried out as described in the paragraph *Sample preparation for mass spectrometry-based proteomics* of material and methods with the testing the effect of iodoacetamide (IAA) (with or without 15 mM IAA) in the formation of CML modification. The analysis was performed on peptides from total cell lysate without CML enrichment. n=4, mean \pm SD.

Description	Composition	Position	Residue	Mass shift
CML	C(2) H(2) O(2)	Not-C-term	K	58.0054793084
CM-Nterm	C(2) H(2) O(2)	Protein N-term	Any	58.0054793084

Table 1. Mass shifts considered for carboxymethylation on lysine (CML) or at any protein N-terminus (CM-Nterm).

Comment 14: CML theoretically may not be easily cut by trypsin. The authors may want to check if all identified CML sites are non-C-terminal lysine. This will be a useful evidence to enhance the identification confidence.

We thank the reviewer for this comment. In our original analysis, we did not exclude modification by CML of peptide C-termini, and, indeed, we had a few identifications with low scores that contained this modification. We have now revised our analysis strategy to also include carboxymethyl modification of protein N-termini (see point below) and exclude peptide C-terminal localization of CML.

For transparency, we would like also to point out that in the process of updating our analysis we decided to use a different search engine for the search of CMLpepIP data. Instead of using MaxQuant as in the original manuscript, we opted for Spectromine (Biognosys AG). The motivation for this change was based on two reasons: (i) Spectromine natively enables to use a high confidence filter for PTMs using a strategy inspired by Sharma *et al.*⁸⁵. In this way, we now provide a single list of modifications with a defined FDR, instead of having to distinguish between low and high confidence identifications as before; (ii) we have empirically noted that Spectromine provides a better control of FDR especially when large sample set are analyzed, e.g., organ CMLpepIP.

* Figure R7 is shown in this point-by-point response letter but not included in the revised manuscript.

⁸⁵ Sharma K, *et al.* Ultra-deep human phosphoproteome reveals a distinct regulatory nature of Tyr and Ser/Thr-based signaling. *Cell Rep* 8, 1583-1594 (2014).

Comment 15: The CML modifies different target proteins in different samples. For example, the protein quality control related proteins were not significantly CML modified in mouse organs like they were in the MEF cells. Is there any data showing how CML on proteostasis related proteins in mouse tissues and mouse aging?

We thank the reviewer for this comment. In our updated analysis, several proteins involved in different stages of proteostasis were found to be carboxymethylated in organs, including subunits of T-complex protein ring complex (TRiC), heat shock proteins, small and large ribosomal subunits, peptidyl-prolyl *cis-trans* isomerase and proteins involved in autophagy (Table R8).

ID	Gene	Category	Protein name	CM site
Q8BGE6	ATG4B	AUTOPHAGY	Cysteine protease ATG4B	Heart, Kidney, Liver (ProtNterm)
Q9DB34	CHMP2A	AUTOPHAGY	Charged multivesicular body protein 2a	Liver (ProtNterm)
Q9D1L9	LAMTOR5	AUTOPHAGY	Ragulator complex protein LAMTOR5	Heart, Kidney, Liver (ProtNterm)
P68372	TUBB4B	AUTOPHAGY	Tubulin beta-4B chain	Heart, Kidney (K324)
P17742	PPIA	PPIs	Peptidyl-prolyl cis-trans isomerase A	Heart (ProtNterm)
P17742	PPIA	PPIs	Peptidyl-prolyl cis-trans isomerase A	Liver (K49)
Q9QZH3	PIIE	PPIs	Peptidyl-prolyl cis-trans isomerase E	Heart, Liver (K185)
Q9D1R9	RPL34	RIBOlarge	60S ribosomal protein L34	Liver (K36)
O55142	RPL35A	RIBOlarge	60S ribosomal protein L35a	Liver (K15)
P14131	RPS16	RIBOsmall	40S ribosomal protein S16	Liver (K60)
Q9CQR2	RPS21	RIBOsmall	40S ribosomal protein S21	Heart, Kidney, Liver (ProtNterm)
P97461	RPS5	RIBOsmall	40S ribosomal protein S5	Kidney (ProtNterm)
P62754	RPS6	RIBOsmall	40S ribosomal protein S6	Liver (K211)
P62082	RPS7	RIBOsmall	40S ribosomal protein S7	Kidney (ProtNterm)
P23927	CRYAB	sHSPs	Alpha-crystallin B chain	Heart, Kidney (ProtNterm)
Q5EBG6	HSPB6	sHSPs	Heat shock protein beta-6	Heart (ProtNterm)
P80318	CCT3	TRiC	T-complex protein 1 subunit gamma	Liver (ProtNterm)
P80313	CCT7	TRiC	T-complex protein 1 subunit eta	Heart, Kidney, Liver (ProtNterm)
P11983	TCP1	TRiC	T-complex protein 1 subunit alpha	Heart, Kidney, Liver (ProtNterm)

Table R8.*

Carboxymethylation sites occurring on proteostasis-related proteins identified in organs from young and old mice combined. The columns show from left to right, the protein UniProt ID, gene name, the category, the protein name and the tissue with the modified residue position where the modification has been identified. PPIs: peptidyl-prolyl *cis-trans* isomerase; RIBOlarge: 60S ribosome; RIBOsmall: 40S ribosome; sHSPs: small heat shock proteins; TRiC: T-complex protein ring complex.

Comment 16: Can glyoxal directly react with lysine and induce CML *in vitro*?

Yes, it does, as we have shown with the *in vitro* experiments on tubulin (Fig. 6D) and proteasome (Fig. 3H) that we included in the revised manuscript (see reply to Reviewer, 1 comment 1 and to comment 2 of this reviewer above).

Figure 6. Glyoxal (GO) affects post-translational modification and dynamics of tubulin.

D. Purified porcine tubulin incubated with GO for 30 min on ice was analyzed in immunoblots. n=5.

* Table R8 is shown in this point-by-point response letter but not included in the revised manuscript.

Figure 3. Alteration of proteostasis induced by glyoxal (GO) in MEF.

H. Left, proteasome activity assay using native gel electrophoresis (top) after incubation of purified proteasome with GO (1 mM, 0, 1, 2 and 4 h, 37 °C). Immunoblot for CML (middle) and proteasome subunits α 1-7 (bottom). **Middle**, quantification of CML-modification. **Right**, proteasome chymotrypsin-like activity (CT-L) normalized to α 1-7 abundance. Positive control: purified proteasome incubated with MG132 (100 μ M, 4 h, 37 °C). n=4, mean + SD, * p<0.05, one-way ANOVA.

Comment 17: More information, such as catalog number and immunogen, about the antibody used for enriching CML-containing peptides should be provided.

We apologize for not including this information in the original manuscript. We have now included the requested information in the method section of the revised manuscript.

Page 27, line 12: *The CML antibody (CN1040) used for enrichment was developed using carboxymethylated KLH as immunogen and affinity affinity-purified with CML-agarose. This antibody and the acetyl lysine antibody (agarose conjugate) were purchased from ImmuneChem Pharmaceuticals Inc. (Burnaby, Canada).*

Comment 18: Figure 4A. 38.6% of CML proteins are nucleus proteins (upper). Why isn't "nucleus" an enriched term in GO (lower)?

We thank the reviewer for pointing out this inconsistency. We have included an updated version of the GO enrichment based on the new analysis of CMLpepIP data (see reply above) in the revised manuscript. In this new analysis, GO terms related to nucleus appear among enriched categories in modified proteins.

Figure 4. Glyoxal (GO) impairs the proliferation of HUVEC.

A. Left, CM-modified proteins annotated to different cellular compartments according to Gene Ontology annotation after GO treatment (1 mM, 48 h). **Right**, Gene Ontology cellular component terms enriched in CM-modified proteins. False Discovery Rate (FDR)<0.05. **B. Left**, Volcano plot depicting proteins that significantly increase (red) or decrease (blue) abundance upon GO treatment (1 mM, 48 h) or remain unchanged (gray). Horizontal dashed line indicates a significance cut-off of $q < 0.05$ and vertical dashed lines an absolute fold change (\log_2)>0.58. $n=4$. **Right**, Gene set enrichment analysis for Gene Ontology biological process terms based on protein fold changes. Terms enriched among increased (red) or decreased (blue) proteins are shown. FDR<0.05; NES: normalized enrichment score.

Comment 19: Page 3, Line 29. The authors stated the CMLpepIP was “reproducible”. However, the data in Figure S1a and S1b were not about reproducibility. The data in figure S1c actually showed the CMLpepIP was not ideal in reproducibility, as only a very small fraction of sites could be detected in all three replicates. For example, the 3rd replicate of MEF control group only had 19 CML sites, whereas the 2nd replicate got 155 sites. I can imagine that the quantification will bear even larger variations between replicates. The authors should rephrase their statement and deliver the right message.

We thank the reviewer for this comment. We have updated the indicated figure panels to the revised version of the CMLpepIP analysis. We believe that part of the reason of the limited overlap between replicates lies in the stochastic nature of Data Dependent Acquisition that we used to acquire data. In this context, we would like to point out that we did not use tools such as “Match Between Runs” to transfer identifications between replicates. We have rephrased the statement according to the reviewer’s suggestion:

Page 3, line 27: *With a lower concentration of glyoxal, the gain of modified peptides was reduced and, in any case, the fraction of PSM assigned to CML peptides never exceeded 5 % (Fig. 1E). This indicates that CMLpepIP enables only partial enrichment of CML-modified peptides, especially in high complexity samples where CML peptides occur at low concentration.*

D

Unique modified sites among MEF experiments = 1113

Unique modified sites among HUVEC experiments = 307

Supplementary Figure 1.

D. Venn diagram representing CM sites identification across independent experiments.

Comment 20: Supplementary tables should be organized and named better. There is no table number in the excel files.

We thank the reviewer for pointing this out. We have revised the organization of the supplementary tables and clearly indicated the number and the keywords of the respective table in the excel file.

Comment 21: Though the authors used antibody against CML, I'm curious if the authors got any carboxymethylation at other positions, such as N-terminus and cysteine.

We thank the reviewer for highlighting this point that we overlooked in our original analysis. We have now re-analyzed our dataset using carboxymethylation of protein N-termini as a variable modification (in addition to exclude CML at peptide C termini). Indeed, we were able to identify a significant number of N-terminal carboxymethylation especially in organs. We have now included these sites in our analysis and reported them separately in Supplementary Table 2. We have also modified accordingly the title our manuscript and modified the result paragraph:

Page 4, line1: *Using CMLpepIP, we identified 1113 unique carboxymethylation sites in MEF and 307 in HUVEC (Supplementary Table 1), of which 1090 (MEF) and 300 (HUVEC) were CML sites and 23 (MEF) and 7 (HUVEC) were CM-Nterm sites.*

Page 6, line 24: *By combining data from young and old mice, we identified 198, 71 and 105 unique CML sites and 81, 116, and 88 CM-Nterm sites in heart, kidney, and liver, respectively (Supplementary Table 2).*

Figure R9.*

Number of peptide spectrum matches containing CML and CM-Nterm modifications. Numbers are the sum of PSM from CML or CM-Nterm-containing peptides from all the runs grouped as MEF, HUVEC and organs.

* Figure R9 is shown in this point-by-point response letter but not included in the revised manuscript.

REVIEWERS' COMMENTS

Reviewer #1 (Remarks to the Author):

The manuscript is much improved.

However, this referee have two comments that must be corrected before it can be published.

- Authors claimed that carboxymethyllysine (CML) is one of t protein modification derived from advanced glycation end product (AGE). Authors cited two papers to valid this claim. What if the cited papers are wrong. Should authors want to include the statement, they should compare this modification with other popular modifications such as phosphorylation, acetylation, etc. otherwise, this sentence should be removed.

authors should include a structure in Figure 1 with this modification so that biologists can have an understanding of this structure.

Reviewer #3 (Remarks to the Author):

I thank the authors for their thoughtful explanations and efforts to address almost all my concerns. I'd like to recommend a minor revision on the following items which will improve this manuscript further.

1. Figure 6E. Was taxol or nocodazole added after the GO treatment? The chart legend should include GO in taxol or nocodazole groups as well.

2. The Y-axis labeling of the rank plots in figure S1E seems incorrect (see other reasonable numbers in figure 1J and K).

3. Figure S4. There is no difference on melting temperature distribution between CM-modified and unmodified proteins, suggesting CM modification is not the reason for the impacted thermal stability on proteasome. The authors should rephrase their statement on this data.

4. Previous comment 21. It's very interesting to see many CM-Nterm sites in the mouse tissues, but not in the cultured cells. Can the authors provide any possible explanation and discussion, which should be included in the main manuscript? Is this because tissues mostly contain post-mitotic cells, thus proteins are more slowly replenished than proteins in the cultured cells? This observation may be in line with the result that slower turnover was more likely to be affected by carboxymethylation. More, the authors haven't answer if carboxymethylation can also occur on cysteine.

Mapping protein carboxymethylation sites provides insights into their role in proteostasis and cell proliferation

Point-to-point reply to the reviewer's comments

We would like to thank the reviewers for the positive evaluation of our manuscript and the additional comments. Below we address the reviewers' comments in a point-to-point reply. New or altered text passages in the manuscript are visible using the track changes mode.

Reviewer's comments

Reviewer #1

Comment 1: The manuscript is much improved.

We thank the reviewer for this positive judgment of our work.

Comment 2: Authors claimed that carboxymethyllysine (CML) is one of the protein modification derived from advanced glycation end product (AGE). Authors cited two papers to valid this claim. What if the cited papers are wrong. Should authors want to include the statement, they should compare this modification with other popular modifications such as phosphorylation, acetylation, etc. otherwise, this sentence should be removed.

We thank the reviewer for this comment. We agree that two references (5,6) would not justify our statement "One of the most abundant AGEs *in vivo* is N(6)-carboxymethyllysine (CML) (page 2, line 12)". However, in general there is strong evidence in the literature confirming that CML belongs to the most abundant AGE modification in tissues (see references I - VIII below). To highlight this, we have now cited two reviews (reference 7,8), which support our statement.

Page 2, line 12: *One of the most abundant AGEs in vivo is N(6)-carboxymethyllysine (CML)*^{5, 6, 7, 8}.

We would like to emphasize that the above-cited sentence refers to CML modification in comparison to other AGE modifications such as methylglyoxal-derived hydroimidazolone 1 or 3-deoxyglucosone-hydroimidazolone 1. The fact that CML is one of the most abundant AGE modifications *in vivo* with potential pathophysiological importance was the rationale to investigate it in more detail in our study. We did not compare the abundance of CML modification with other posttranslational protein modifications such as phosphorylation or acetylation.

References:

5. Hohmann C, *et al.* Detection of Free Advanced Glycation End Products in Vivo during Hemodialysis. *J Agric Food Chem* **65**, 930-937 (2017).
6. Baldensperger T, Eggen M, Kappen J, Winterhalter PR, Pfirmann T, Glomb MA. Comprehensive analysis of posttranslational protein modifications in aging of subcellular compartments. *Sci Rep* **10**, 7596 (2020).
7. Brings S, Fleming T, Freichel M, Muckenthaler MU, Herzig S, Nawroth PP. Dicarbonyls and Advanced Glycation End-Products in the Development of Diabetic Complications and Targets for Intervention. *Int J Mol Sci* **18**, (2017).
8. Delgado-Andrade C. Carboxymethyl-lysine: thirty years of investigation in the field of AGE formation. *Food Funct* **7**, 46-57 (2016).
- I. Kawabata K, *et al.* The presence of N(epsilon)-(Carboxymethyl) lysine in the human epidermis. *Biochim Biophys Acta* **1814**, 1246-1252 (2011). [not cited in the manuscript]
- II. Bartakova V, Kollarova R, Kuricova K, Sebekova K, Belobradkova J, Kankova K. Serum carboxymethyl-lysine, a dominant advanced glycation end product, is increased in women with gestational diabetes mellitus. *Biomed Pap Med Fac Univ Palacky Olomouc Czech Repub* **160**, 70-75 (2016). [not cited in the manuscript]
- III. Gaens KH, *et al.* Protein-Bound Plasma Nepsilon-(Carboxymethyl)lysine Is Inversely Associated With Central Obesity and Inflammation and Significantly Explain a Part of the Central Obesity-Related Increase in Inflammation: The Hoorn and CODAM Studies. *Arterioscler Thromb Vasc Biol* **35**, 2707-2713 (2015). [not cited in the manuscript]
- IV. Gaens KH, *et al.* Endogenous formation of Nepsilon-(carboxymethyl)lysine is increased in fatty livers and induces inflammatory markers in an in vitro model of hepatic steatosis. *J Hepatol* **56**, 647-655 (2012). [not cited in the manuscript]

- V. Duran-Jimenez B, *et al.* Advanced glycation end products in extracellular matrix proteins contribute to the failure of sensory nerve regeneration in diabetes. *Diabetes* **58**, 2893-2903 (2009). [not cited in the manuscript]
- VI. Thornalley PJ, *et al.* Quantitative screening of advanced glycation endproducts in cellular and extracellular proteins by tandem mass spectrometry. *Biochem J* **375**, 581-592 (2003). [not cited in the manuscript]
- VII. Smuda M, *et al.* Comprehensive analysis of maillard protein modifications in human lenses: effect of age and cataract. *Biochemistry* **54**, 2500-2507 (2015). [not cited in the manuscript]
- VIII. Ni J, *et al.* Plasma protein pentosidine and carboxymethyllysine, biomarkers for age-related macular degeneration. *Mol Cell Proteomics* **8**, 1921-1933 (2009). [not cited in the manuscript]

Comment 3: Authors should include a structure in Figure 1 with this modification so that biologists can have an understanding of this structure.

We thank the reviewer for this comment. We have now included the CML structure in Figure 1 (panel 1a).

Figure 1. Antibody-based enrichment of CML-modified peptides.
 a. Workflow for the identification of CML-modified peptides. Ab: antibody

Reviewer #3:

Comment 1: I thank the authors for their thoughtful explanations and efforts to address almost all my concerns. I'd like to recommend a minor revision on the following items which will improve this manuscript further.

We thank the reviewer for this positive comment.

Comment 2: Figure 6E. Was taxol or nocodazole added after the GO treatment? The chart legend should include GO in taxol or nocodazole groups as well.

We apologize for the lack of clarity in the legend to Figure 6e. We have now pointed out that taxol and nocodazole were added to vehicle-treated tubulin at the start of polymerization.

Legend Figure 6

6e. Purified porcine tubulin was pre-incubated with GO (1 mM, 30 min, on ice), or vehicle and polymerization induced by addition of cofactors at 37 °C was monitored in a fluorescence-based assay. Controls: taxol (3 μM) or nocodazole (20 μM) added to vehicle-treated tubulin at the start of polymerization. n=3 (taxol, nocodazole), n=6 (control (Ctrl), GO), mean ± SEM. Statistical significance was analyzed using one-way repeated measurement ANOVA corrected via Holm-Šidák method for areas under the curves. * p<0.05 vs. control.

Comment 3: The Y-axis labeling of the rank plots in figure S1E seems incorrect (see other reasonable numbers in figure 1J and K).

We thank the reviewer for spotting this inconsistency. The inconsistency was due to an erroneous double logarithmic transformation in Figure S1e. We have now corrected this mistake and updated Figure S1e accordingly.

Supplementary Figure 1.

e. Rank plot showing the abundance distribution of proteins identified as targets of CM.

Comment 4: Figure S4. There is no difference on melting temperature distribution between CM-modified and unmodified proteins, suggesting CM modification is not the reason for the impacted thermal stability on proteasome. The authors should rephrase their statement on this data.

We thank the reviewer for this comment. Figure S4b shows that CM-modified proteins tend to do not display significant changes in melting temperature. In the case of the proteasome, especially 19S proteins, we observed mild but significant changes in thermal stability. However, with the current data, we cannot link these changes of stability directly to CM modification. We have therefore re-phrased the relevant statement as follows:

Page 12, line 6: Here, we provide evidence for an interference of glyoxal with the thermal stability of the 19S proteasome. However, with the current data, we cannot establish whether the change of thermal stability of the proteasome is due to CM modification or other mechanisms mediated by glyoxal. For example, a similar phenomenon was recently described in response to ATP deprivation in cells⁵¹.

Comment 5: Previous comment 21. It's very interesting to see many CM-Nterm sites in the mouse tissues, but not in the cultured cells. Can the authors provide any possible explanation and discussion, which should be included in the main manuscript? Is this because tissues mostly contain post-mitotic cells, thus proteins are more slowly replenished than proteins in the cultured cells? This observation may be in line with the result that slower turnover was more likely to be affected by carboxymethylation. More, the authors haven't answer if carboxymethylation can also occur on cysteine.

We thank the reviewer for this positive comment. Indeed, a possible explanation for the higher number of CM-Nterm sites identified in tissues is the slower turnover of proteins *in vivo*, as compared to cell culture systems. We have indicated this possibility in the revised manuscript:

Page 11, line 1: In addition to CML, we also detected carboxymethylation of protein N-termini, especially in mouse organs. We speculate that the higher number of CM-Nterm sites identified in organs might derive from the slower turnover of proteins in vivo as compared to cultured cells. This observation is in line with the results suggesting that slow protein turnover is associated with an increased likelihood of carboxymethylation.

Regarding carboxymethylation of cysteines, we have investigated the impact of iodoacetamide (IAA) treatment (part of our sample preparation procedure) on the formation of this modification, similarly to what we did for CML in the previous rebuttal letter. Differently from CML, carboxymethylation of cysteines was influenced by IAA treatment, via a known process that occurs at low pH^{IX}.^X Because of this, we have decided not to investigate further carboxymethylation of cysteines in our study (Figure R1).

⁵¹ Sridharan S, et al. Proteome-wide solubility and thermal stability profiling reveals distinct regulatory roles for ATP. *Nat Commun* **10**, 1155 (2019).

^{IX} Aitken A., Learmonth M. Carboxymethylation of Cysteine Using Iodoacetamide/ Iodoacetic Acid. *Walker J.M. (eds), The Protein Protocols Handbook. Springer Protocols Handbooks. Humana Press* (2002). <https://doi.org/10.1385/1-59259-169-8:455>. [not cited in the manuscript]

^X Simpson R.J. Reduction and s-carboxymethylation of proteins: large-scale method. *CSH Protocols* (2007). <https://doi.org/10.1101/pdb.prot4569>. [not cited in the manuscript]

Figure R1.*

Percentage of peptide spectrum matches (PSM) carrying the indicated modifications in samples obtained from control and 2 mM glyoxal (GO)-treated cells. The sample preparation was carried out as described in the paragraph "Sample preparation for mass spectrometry-based proteomics" of the methods part. In order to directly test the effect of iodoacetamide (IAA) on the formation of carboxymethyl lysine (CML), carboxymethyl cysteine (CMC) and carbamidomethyl cysteine, we omitted the alkylation step in one sample group (-IAA). The analysis was performed on peptides from total cell lysate without CML enrichment. n=4, mean \pm SD.

* Figure R1 is shown in this point-by-point response letter but not included in the revised manuscript